# Data-Efficient Training of CNNs and Transformers with Coresets: A Stability Perspective

## Abstract

Coreset selection is among the most effective ways to reduce the training time of CNNs, however, only limited is known on how the resultant models will behave under variations of the coreset size, and choice of datasets and models. Moreover, given the recent paradigm shift towards transformer-based models, it is still an open question how coreset selection would impact their performance. There are several similar intriguing questions that need to be answered for a wide acceptance of coreset selection methods, and this paper attempts to answer some of these. We present a systematic benchmarking setup and perform a rigorous comparison of different coreset selection methods on CNNs and transformers. Our investigation reveals that under certain circumstances, random selection of subsets is more robust and stable when compared with the SOTA selection methods. We demonstrate that the conventional concept of uniform subset sampling across the various classes of data is not the appropriate choice, highlighting the need for investigation towards adaptive sampling based on the complexity of each class's data distribution. Transformers are generally pretrained on large datasets, and we show that for certain target datasets, it helps to keep their performance stable at even very small coreset sizes. We further show that when no pretraining is done or when the pretrained transformer models are used with non-natural images (*e.g.* medical data), CNNs tend to generalize better than transformers at even very small coreset sizes. Lastly, we demonstrate that in the absence of the right pretraining, CNNs are better at learning the semantic coherence between spatially distant objects within an image, and these tend to outperform transformers at almost all choices of the coreset size.

## 1 Introduction

One of the primary reasons for the success of deep learning models is the availability of a large amount of data to train them. However, this success is also accompanied by the requirement of a large amount of computing power, primarily GPU computing, and can lead to significant financial burden as well as energy usage (Strubell et al., 2019). For example, training such models on cloud-based GPU servers is often very expensive, and companies would often want to cut down these costs by possibly reducing the training time. Clearly, there is a need to train these deep models in a more efficient manner without compromising on their performance (Asi & Duchi, 2019).

Training large datasets is challenging in terms of required memory, and methods such as stochastic gradient descent or similar are used to gradually optimize the models over small subsets of the training data (Hofmann et al., 2015). While such a strategy provides an unbiased estimate of the full-gradient over a series of update steps, it still introduces variance and the convergence is generally very slow for large datasets. The significant increase in the associated training time then becomes a bottleneck for cases where a limited amount of computational time is available. This issue is even more prevalent in adversarial training of deep networks (Dolatabadi et al., 2022). It is very effective for training robust models against adversarial attacks, however, this comes at the cost of the training process being very slow since it needs to construct adversarial examples for the entire training data at every iteration.

Coreset selection is an effective way to reduce the model training time. In simple terms, coreset selection aims at optimally selecting a subset $\mathcal{S} \subset \mathcal{D}$ of the full training data $\mathcal{D}$, such that when trained on only $\mathcal{S}$, the model converges approximately to the same solution as $\mathcal{D}$. If this subset $\mathcal{S}$ can be easily found with almost no overhead time, then we see a straightforward speed up of $|\mathcal{D}|/|\mathcal{S}|$ in the overall training time. However, there are several challenges associated with coreset selection, such as identifying the right principle for selecting the training samples, tackling the additional time associated with this process, and appropriately selecting the training hyperparameters based on the selected subset, among others. There exist several recent works that attempt to address the issues outlined above. Some examples include CRAIG (Mirzasoleiman et al., 2020a), GradMatch (Killamsetty et al., 2021a), GLISTER (Killamsetty et al., 2021b), among others. CRAIG and GradMatch are gradient-based methods that identify the optimal coreset by converting the original gradient matching problem to a problem of monotone submodular function optimization, and solve it within a pre-defined error bound. GradMatch additionally uses L2 regularization to reduce the level of dependency on any specific data sample. GLISTER poses a bi-level optimization that solves the selection of subsets as an outer objective and the optimization of model parameters as an inner objective of the optimization problem. Each of these methods has certain advantages, however, it is unclear at this point as to when to use which method. Or in other words, a rigorous comparative study on model performance vs. training time across different coreset selection methods for recent state-of-the-art (SOTA) models is still missing.

Further, given the recent paradigm shift to transformers, it is still an open question on how coreset selection would impact the performance of transformer networks. It is well known that transformers are data-hungry, and without heavy pretraining, these are known to underperform at downstream tasks. Clearly, with small coresets, this should worsen even more, and this would imply that in the absence of suitable pretraining, coreset selection should not be used at all. However, a clear consensus on this is still missing in the existing literature. It is not yet known which coreset selection method should be preferred for transformers and what should be the coreset sampling frequency. Further, when using pretrained models, it is still unclear whether transformers would be as stable as CNN models when small coresets are chosen. Transformers are known, in general, to work well for tasks related to natural images, and this is especially due to the detailed learning on Imagenet21K (Ridnik et al., 2021), which has similar images. However, there is no clear evidence that the knowledge distilled from Imagenet21K is sufficient enough to keep their performance stable when using small coresets of the training data. Thus, beyond understanding how transformers perform on non-natural images, we also study how the performance is affected when the coreset size is reduced.

The practical applications of deep learning span far beyond image classification on natural images. We consider here two specific cases. First is the application on medical datasets, comprising images different from the standard natural scenes (referred to as non-natural images). For such cases, it is of interest to study whether the conventional pretraining of the models is good enough to keep the performance of transformers stable and at par with a CNN model of equivalent training time. And if this is true, we are further interested in exploring how the performance dips when the coreset size is reduced. Our second application is to seek the goodness of spatial correspondence in non-natural images when working with transformers. Transformers are known to capture the global context very well, however, when seeking strong semantic coherence between spatially distant objects (*e.g.* UltraMNIST classification), it is of interest to see how their pretraining, as well as coreset selection, affect the learning process. Conventionally, coreset selection methods assume an equal number of samples per class for the chosen coreset, however, the complexity of the distribution is different for each class of UltraMNIST, and we are interested to understand how the conventional coreset selection scheme affects the overall performance of CNNs and transformers and what adaptation can be done to improve them.

Finally, we summarize the outcome and novel insights obtained from this study below.

- We present a systematic benchmarking setup and perform a rigorous comparison of different coreset selection methods on CNNs and transformers. Our investigation reveals that under certain circumstances, random selection of subsets is more robust and stable when compared with the SOTA selection methods.

- We demonstrate that uniform subset sampling across different data classes is not a suitable approach, highlighting the need for adaptive sampling based on the complexity of each class's data distribution.

- Transformers are generally pretrained on large datasets, and we show that for certain target datasets, it helps to keep their performance stable at even very small coreset sizes. Thus, these tend to outperform time-equivalent CNN models by large margins.

- We further show that when no pretraining is done or when the pretrained transformer models are used with non-natural images (*e.g.* medical data), CNNs tend to generalize better than transformers at even very small coreset sizes.

- Lastly, we demonstrate that in the absence of the right pretraining, CNNs are better at learning the semantic coherence between spatially distant objects within an image, and these tend to outperform transformers at almost all choices of the coreset size.

## 2 Related Work

Most of the recent deep learning models are computationally expensive and require training on very large datasets. This leads to huge energy-related costs, and there is a strong desire to make the training energy-efficient (Strubell et al., 2019). This can be achieved in many ways, such as training the model on lower bit representation or using lower resolution images (Banner et al., 2018; Chin et al., 2019), designing training schemes adapted to the hardware design, (Wu et al., 2019; Hoffer et al., 2018), and using only the important data samples for training (Mirzasoleiman et al., 2020a; Killamsetty et al., 2021b;a), among others. This paper focuses on making the training efficient through the selective sampling of the training data, also referred to as coreset selection in the existing literature.

Coreset selection methods have been studied for over two decades (Feldman, 2020), designed originally for the applications in computational geometry (Agarwal et al., 2005), but quickly adopted later by the machine learning community and applied to classical machine learning based problems. As described in Mirzasoleiman et al. (2020a), initially, these problems varied from, finding a high likelihood solution for $k$-means and $k$-median clustering (Har-Peled & Mazumdar, 2004), SVM (Clarkson, 2010), graphical model training (Molina et al., 2018), logistic regression (Huggins et al., 2016), naive bayes (Wei et al., 2015) and nsytrom methods (Musco & Musco, 2017).

With the emergence of data-driven approaches, one of the natural applications of coreset selection is deep learning, as data is a critical component of all deep-learning-based algorithms. However, this application is not straightforward due to the innate nature of coreset algorithms, which makes them extremely task-specific. This intrinsic property of the coreset algorithm poses a challenge in their utility for deep-learning-based methodologies (Killamsetty et al., 2021a). Nonetheless, recent works using coresets in deep learning have shown promise. Starting from one of the pioneering works (Mirzasoleiman et al., 2020a), where authors proposed to select a subset of training data, that can capture the approximation of the full gradient.

Some recent works have used coreset algorithms, not only for data sub-selection but also for data cleaning and learning from noisy labels. For instance, Mirzasoleiman et al. (Mirzasoleiman et al., 2020b), proposed a technique based on coresets, which would select clean data points and provide an approximation of low-rank jacobian matrix for a more generic representation learning. Along with noisy data, class imbalance in training data is often the cause of over-fitting and poor model performance. Therefore, coreset also gained attention in this domain, and a few recent works, such as Killamsetty et al. (2021b;a), have successfully deployed coreset algorithms to tackle the class-imbalance problem, which in turn leads to more generalized models. Besides data, other important components of deep-learning-based algorithms are the model size and number of parameters. One of the first works is Baykal et al. (2018), which proposed to use a coreset, not for data sub-selection but for model parameter reduction. This seminal work laid the foundation for several other works, such as Mussay et al. (2021; 2019); Tukan et al. (2022), using coresets for neural network pruning. Sorscher et al. (2022) studied the behaviour of different data pruning methods and how it can be used to overcome the power law scaling behaviour observed in large neural networks. They showed that by selecting which samples to retain and which to discard, it is possible to achieve similar or even better performance with smaller network sizes. This work differs from ours as coreset selection methods aim to construct a smaller subset of the training dataset that represents the overall data distribution. In contrast, data pruning focuses on removing specific samples based on their individual contribution. Data pruning is more concerned with reducing redundancy and noise, while coreset selection is more concerned with preserving the essential

characteristics of the data. However, all of the aforementioned works either limited their study to a specific dataset or studied how coresets can be utilized for neural network pruning, given a specific architecture (CNNs, for instance).

However, in this work, we study coresets from a data perspective. Several recent works have attempted to study coreset selection behaviour using different settings of the data. Mirzasoleiman et al. (2020b) used 50% symmetric noise versions of the CIFAR10 and CIFAR100 and studied the effect of size of coreset on the final accuracy. Killamsetty et al. (2021b) showed the effectiveness of their method by using either shallow network architectures (a two-layer fully-connected neural network with 100 hidden nodes) or deep learning models such as Resnet18 on small-scale datasets like MNIST and CIFAR10. These and most other methods are limited to small-scale datasets which share a distribution similar to ImageNet21k Russakovsky et al. (2015a). These works have reported full training time, but none of the methods studies the coreset selection time of different coreset selection methods. We show in this study that although we generally assume the coreset selection time to be negligible, it is not. Our work presented in this paper covers a study on large-scale models like Vision Transformers and is experimented on diverse datasets, including those from a different domain such as medical datasets. We also study the influence of the individual coreset selection time taken by each method for different coreset sizes.

While the various coreset selection methods that exist in the literature have been successful in significantly reducing the associated computational time, only limited research has been conducted to perform comparative analysis of these methods. To our knowledge, only Killamsetty et al. (2021a) have compared the existing coreset selection methods in terms of performance under a unified setting. However, the scope of this study is still limited in various aspects. First and foremost, the study is restricted to CNN architectures and no experiments with Transformer models were included. With the recent popularity gained by Transformers, it is of interest to study and compare coreset selection methods on them. Further, the previous work has only focused on studying coreset selection methods in the context of natural images similar to the ImageNet21K dataset, however, we extend our study to also include non-natural images such as medical images. Lastly, coreset selection methods have been tested only for scenarios where the complexity of the distribution across different classes is balanced. In this paper, we also consider a scenario where different classes of a classification problem have different levels of complexity, and study the impact of coreset selection on the model performance for such as dataset.

Overall, the study of coreset selection methods in the context of data-efficient training has been very limited, and none of the existing works have thoroughly analyzed the influence of the coreset selection time as well. For the first time, we present a thorough quantitative and objective perspective to coreset selection algorithms on different benchmarks. Additionally, we also study how different architectures (CNNs and Transformers) respond to subtle differences in different coreset selection algorithms to provide insights for future research.

## 3 Coreset Selection

### 3.1 General description

Coreset selection refers to identifying a subset of the training data that, when used to train the model, can approximately reach the same optimal solution as when trained with the full data. Fig. 1 presents an overview of the process. Generally, a coreset budget is chosen, and an initial random subset is used to choose the first coreset.

This coreset is then used to train the model for a certain number of epochs $T_1$. At this stage, another coreset selection is done, but with more sophisticated approaches, such as GradMatch, and GLISTER, among others. The model is trained again till epoch $T_2$ and another coreset selection happens. This process continues till the model is fully trained. Coreset selection happens after pre-defined intervals (referred to as subset selection internal (SSI) in this paper).

To further explain the concept of coreset selection, we formulate the mathematical optimization problem solved as part of training a neural network model. Let $f : \mathbb{R}^d \to \mathbb{R}^k$ denote a neural network with weights

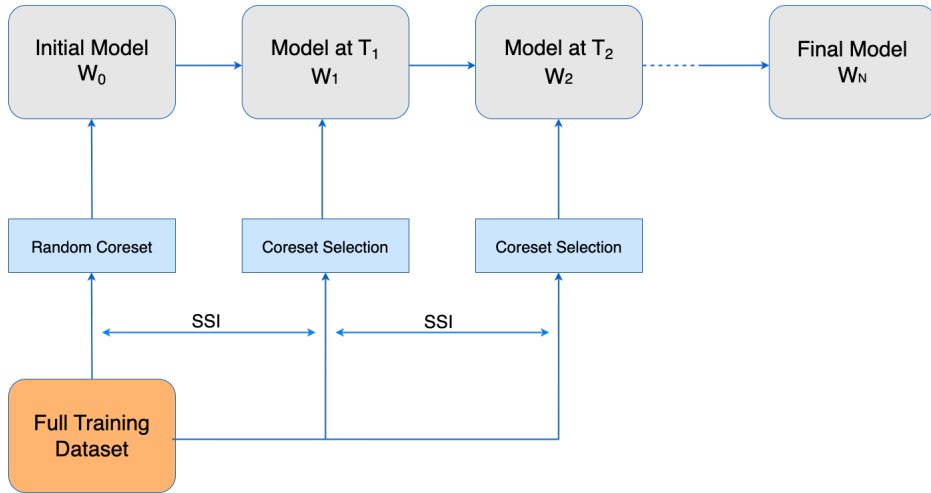

Figure 1: Schematic representation of coreset selection for model training. Here SSI denotes the subset selection interval.

$\mathbf{w}$. During training, the parameter set $\mathbf{w}$ is optimized with respect to a certain objective function, and the associated optimization problem is to find the optimal weights $\mathbf{w}^*$ as follows.

$$\mathbf{w}^* = \underset{\mathbf{w} \in \mathcal{W}}{\operatorname{argmin}} \ \mathcal{L}(f(\mathbf{w}; \mathbf{x}), \mathbf{y}) \tag{1}$$

where $\mathcal{D} = \{\mathbf{x}, \mathbf{y}\}$ denotes the input-output pairs in the training data, and $\mathcal{W}$ denotes the parameter search space. Further, $\mathcal{L}^*$ denotes the optimized objective value above.

The simple idea of coreset selection is to find the data subset $\mathcal{S} \subset \mathcal{D}$ where $\mathcal{S} = \{\tilde{\mathbf{x}}, \tilde{\mathbf{y}}\}$ such that $\tilde{\mathcal{L}}^* \approx \mathcal{L}$, where

$$\tilde{\mathcal{L}}^* = \underset{\mathbf{w} \in \mathcal{W}}{\min} \ \mathcal{L}(f(\mathbf{w}; \tilde{\mathbf{x}}), \tilde{\mathbf{y}}) \tag{2}$$

Assuming that the total time spent on coreset selection is very small compared to solving the optimization problem stated in Eq. 2, it is evident that a straightforward speedup of $|\mathcal{D}/\mathcal{S}|$ fold can be obtained. It is important to note that the above outlined assumption is an ideal scenario that still needs to be achieved and a perfect coreset selection method that complies with it is still be to identified. To take this into account, we consistently consider the coreset selection time as well in all the experiments reported in this paper.

However, there are several challenges that make the selection of $\mathcal{S}$ hard. The first is to identify the right guiding principle for coreset selection. For example, one could choose more points around the decision boundary or alternatively prefer to go for a more diverse distribution. Depending on the choice, the resultant model is expected to behave differently. Secondly, this paper focuses on minimizing the desired training time, and it is important that the time spent on selecting the coresets is not very significant. Another equally important challenge is appropriately adapting the model's hyperparameters to the obtained coreset. Most SOTA coreset selection methods that exist so far have focused on eliminating the challenges posed above, however, each of these has its own benefits as well as limitations, and we will study this aspect in the paper.

### 3.2 Choice of coreset selection methods

In this paper, we choose 3 recent coreset selection methods, and in addition, we also include the random coreset selection approach. We review in brief these coreset selection algorithms along with their formal definitions [1].

---

[1]For more details on the subselection method, please refer to Killamsetty et al. (2022)

**CRAIG.** CRAIG (Mirzasoleiman et al., 2020a) is a method that focuses on selecting a smaller subset (coreset) of data points to accelerate incremental gradient descent (IG) by approximating the full gradient update while reducing computational cost. It aims to minimize the average maximum deviation of gradients between the full set and the coreset. By iteratively making steps based on the gradient of each function, CRAIG enables fast convergence to a near-optimal solution in the incremental setting. The optimization problem of finding the optimal subset involves selecting the smallest subset $\mathcal{S}$ and corresponding per-element stepsizes $\gamma_j$ that approximate the full gradient with an error no greater than $\mathcal{M}$. CRAIG's ability to select data subsets at each epoch offers flexibility in applications. For more detailed information on how CRAIG works, please refer to the appendix section of the paper.

**GradMatch.** GradMatch (Killamsetty et al., 2021a) is a Gradient Matching method that aims to identify subsets of data resembling the gradients of the training or validation set. Drawing inspiration from adaptive data selection strategies' convergence analysis, GradMatch optimizes an error term to measure the resemblance between a weighted data subset and either the full gradient or validation set gradients. This optimization is carried out using an orthogonal matching pursuit (OMP) algorithm. By selecting representative subsets, GradMatch enhances the efficiency and effectiveness of deep learning model training. It offers a principled approach for data subset selection, improving the training process's efficiency while maintaining the quality of gradient approximation. For a more detailed understanding of how GradMatch operates, refer to the appendix section.

**GLISTER.** GLISTER (Killamsetty et al., 2021b) is a method that aims to identify a small subset of training data that effectively represents the entire dataset while maintaining high model performance and generalization. It employs a two-stage strategy consisting of Exploration and Exploitation steps. In the Exploration step, GLISTER scans the dataset to identify representative data points for multiple subsets. The representation ability of each subset is evaluated based on various factors, including distance-based dissimilarity, model confidence, and model sensitivity. Once a diverse and relevant pool of subsets is generated, the Exploitation step evaluates the model performance on each subset and selects the one that achieves the best trade-off between performance and subset size. In essence, GLISTER addresses the data sub-selection task by solving a bi-level optimization problem, with the objective of maximizing the log-likelihood on a held-out validation set. For a more detailed knowledge of GLISTER's methodology, refer to the appendix section.

**Random selection.** It involves selecting a random coreset from the full training data, without any heuristics and learning. In this work, random selection serves as a baseline.

### 3.3 Benchmarking strategy

For a fair comparison of different coreset selection methods across the different CNN and transformer methods, we define here a benchmarking setup. Note that this setup is designed with respect to making the training process more efficient. In this regard, we also introduce some parameters that will be used consistently across all experiments of this paper to compare the results.

**Definitions.** There are several parameters that are important for the setup, and we define them here. We also provide the motivation behind the relevance of each parameter in the context of comparing various coreset methods for CNNs and transformers across the different coreset sizes.

*Effective data per epoch (EDPE).* It refers to the fraction of the trainset used as a coreset for training the model per epoch. EDPE gives an approximate estimate of the reduction in the training time of the model and excludes the time spent on selecting the coresets.

*Subset selection interval (SSI).* It refers to the number of epochs that need to be run between two consecutive coreset selections. Thus, for a setup with a fixed number of epochs, a higher value of SSI would imply a lesser number of coreset selection cycles. In general, the influence of SSI on the overall performance of the trained model is not known well, and we experiment with different values of SSI to find the right number of coreset selections that maximize the performance of the model.

*Total selection time.* It refers to the amount of time spent on selecting the optimal subset of the training data using the chosen coreset selection method. Although it is assumed that the cost associated with coreset

selection is negligible, we have seen that this time plays a crucial role in evaluating the performance of any given coreset selection method. Details on the total selection time for all the experiments are reported in the supplementary section of this paper.

*Total time.* It corresponds to the total time spent on training the model as well as coreset selection. In terms of computing the efficiency of a training process, we consider total time as the right metric and report it frequently in our results presented later in this paper.

*Epochs.* Similar to any training procedure in deep learning, epoch refers to the number of passes through the entire dataset (coreset here). It can also be interpreted as the iterative effort invested to identify the best model and has a significant impact on the training cost. Note that best models for different methods might be obtained at a different epoch value. However, when calculating the training cost, we consider the total number of epochs that are planned beforehand. The reason is that from the implementation perspective, the best model cannot be known till the training process has been completed.

**Setup.** Among the defined control parameters, we fix the total number of epochs to be constant across all the experiments of this paper, and it is denoted as $N$. Further, we experiment with different values of SSI to identify the right balance between the subset sampling frequency and the performance of the trained model. No constraint is imposed on the other three parameters, and these are measured for different experiments. Among the total time and the total selection time, total time is a more practical choice for study. However, since the coreset selection can significantly vary across CPUs depending on the availability of threads, we also study training time for each model.

## 4 Experiments

### 4.1 Implementation details

**Models.** In this work, we experimented with four baselines, two CNN-based and two transformer-based backbones. We adopted a ResNet-50 model, along with an application-oriented, embedded vision-based model MobileNetV3-Large (referred to as MobileNetV3). For ResNet, we used the recent ResNetV2-50x1 model (referring further as ResNet50), where x1 denotes the factor by which every hidden layer is widened. Saved checkpoint models for ResNet50 and MobilenetV3 were obtained from Kolesnikov et al. (2020) and Wightman (2019), respectively. For transformers, we used two SOTA models, namely ViT (B16) and Swin-Transformer, the base variant. These choices were dictated by the fact that these models have achieved SOTA results on several visual perception tasks (classification, object detection, etc.). We used the standard ImageNet21k weights publicly available for initializing our models.

**Datasets.** For benchmarking the coreset selection methods, we experiment on four popular datasets.

*CIFAR10.* CIFAR10 (Krizhevsky et al., 2009) dataset contains 50000 images with 10 classes. We used $224 \times 224$ as the input to the models in our experiments. We chose this dataset due to its small size and to see the performance of Transformers when trained with small coreset sizes.

*TinyImageNet.* TinyImageNet (Wu et al., 2017) is a subset of ImageNet21k containing a total of 100k images with 200 classes. This dataset has been created as a smaller, but very good representative of the original Imagenet Russakovsky et al. (2015b). Thus, we use it to show a comparison study of coresets with different backbones close to the ImageNet.

*Oxford-IIIT Pet.* The Oxford-IIIT Pet dataset (Parkhi et al., 2012) comprises images showcasing different breeds of cats and dogs. The dataset includes 37 categories, with 200 samples available for each category. Notably, this dataset is distinct from the ImageNet21K subset, but it does contain images resembling those found in ImageNet21K. Consequently, it proves valuable in exploring the impact of pretrained and randomly initialized models using various coreset methods on natural datasets.

*APTOS-2019.* APTOS 2019 (Karthik, 2019), is medical dataset related to blindness detection. The dataset contains 5 classes, with classification labels ranging from 0 to 4, showing the severity level. In total, APTOS (Karthik, 2019), contains 3662 fundus images. This dataset is a good example of non-natural images, and we expect that strong priors learnt on ImageNet21k might not adapt well to this domain.

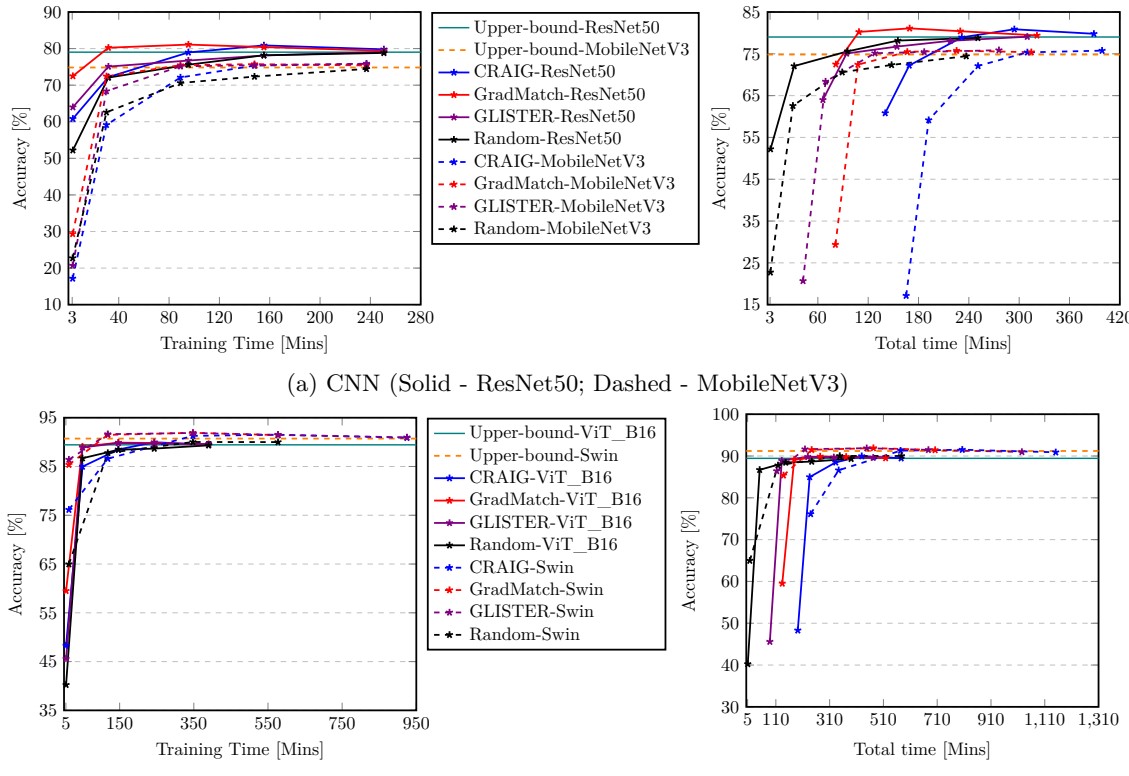

(a) CNN (Solid - ResNet50; Dashed - MobileNetV3)

(b) Transformer (Solid - ViT_B16; Dashed - Swin)

Figure 2: Performance scores for various coreset selection methods for different choices of training time as well as total time. Models are pretrained on ImageNet21k and further trained for TinyImageNet classification. Each data point in the above figures has different EDPE values. EDPE values are in increasing order of the x-axis as 1%, 10%, 30%, 50%, and 80%.

*UltraMNIST.* This dataset (Gupta et al., 2022) is designed primarily to assess the capability of models to capture semantic coherency in large images. We use $512 \times 512$ variants of the original dataset. Each sample of UltraMNIST comprises 3-5 MNIST digits of varying scales, and the sum of these digits can vary from 0 to 27, also defining the class labels 0-27. Note that each label in the dataset can be formed with different combinations of the digits, with some classes having very less combinations, while some can be formed by a large number of combinations. Thus, the complexity of the distribution of every class is different, and this makes UltraMNIST an interesting dataset for coreset selection study. For more details related to the dataset, see Gupta et al. (2022).

**Configuration of hyperparameters.** We used the hyperparameters taken from the extensive study by Steiner et al. (Steiner et al., 2021): SGD momentum optimizer, cosine decay schedule, no weight decay, gradient clipping at global norm 1 and 224×224 image resolution for CIFAR10, Tiny-Imagenet, APTOS-2019, and 512x512 for UltraMNIST. For the sake of fair comparison, all other hyperparameters are similar across all the backbones and datasets used in this study.

### 4.2 Results

We discuss below the results related to the various benchmarking experiments conducted in this paper.

**Comparison of coreset selection methods.** To the best of our knowledge, a clear comparison of the recent coreset selection methods in terms of training time does not exist, especially for transformers. We begin with benchmarking our selected models, using 4 coreset selection methods as described above. Fig. 2, illustrates the performance comparison in terms of training time along with the total time (training + coreset selection). From a practical perspective, the total time needs to be reduced. However, it can be argued that

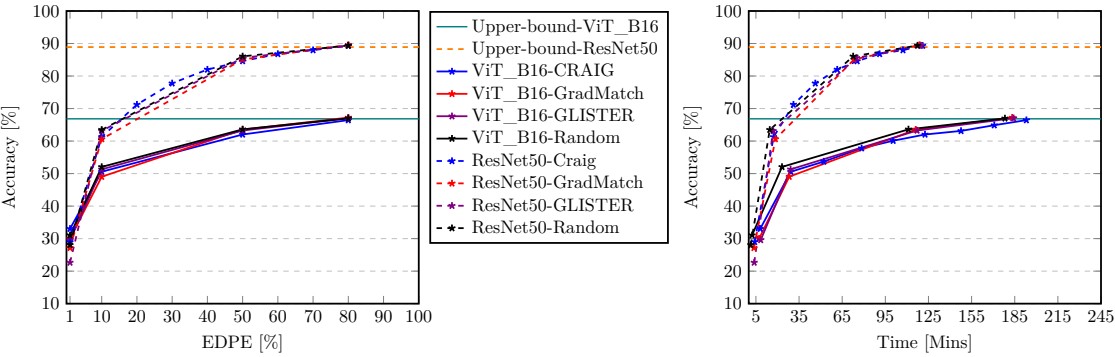

Figure 3: Performance comparison between Randomly Initialised ViT_B16 and ResNet50 for time, EDPE and performance on CIFAR10, SSI=50.

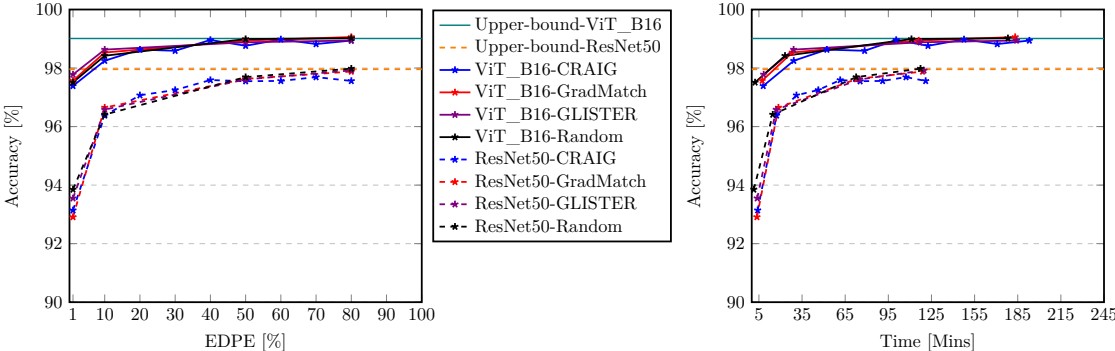

Figure 4: Performance scores for ResNet50 and ViT_B16 for various EDPE values and total time budgets for time, EDPE and performance on CIFAR10. Models are pretrained on ImageNet21k and further trained for CIFAR10 classification.

coreset selection can be scaled across multiple CPU threads, thereby making the coreset selection time very small.

For ResNet and MobileNet, except for random selection, GradMatch consistently outperforms other methods. Whereas, for longer training time, CRAIG is only marginally better. For very small coresets, GLISTER is outperformed by even the random selection method. As for the total time, except for GradMatch, all methods get outperformed by random selection (especially, small time budgets), which is counterintuitive and implies that the coreset selection methods are not stable for CNNs at low training time budgets.

Similar observations are made for transformer models (all training time). For small time budgets, we see that for ViT_B16 as well as the Swin transformer, random coreset selection performs better than other methods, clearly indicating that a generalizable solution to make models time-efficient through coreset selection is still missing.

**Data subselection under no pretraining.** In this study, we performed experiments on the CIFAR10 and Oxford-IIIT Pet datasets, utilizing ResNet50, MobileNetV3, VIT_B16, and Swin models with CRAIG, GradMatch, GLISTER, and Random methods. The corresponding results for CIFAR10 and Oxford-IIIT dataset are presented in Figs. 3 and Fig. 6, respectively. It is evident that CNN architectures outperform transformers significantly when not pretrained. We contend that this discrepancy arises from transformers' high data requirements, making them particularly vulnerable in the absence of pretraining. Moreover, this finding highlights the potential irrelevance of coreset selection for transformers without appropriate pretraining.

**Data subselection under pretraining.** In this section, we explore the influence of pretraining (specifically, on ImageNet21k) on model performance when employing coreset selection. The performance scores are depicted in Fig. 4, where CRAIG is utilized as the coreset selection method. Notably, both models exhibit

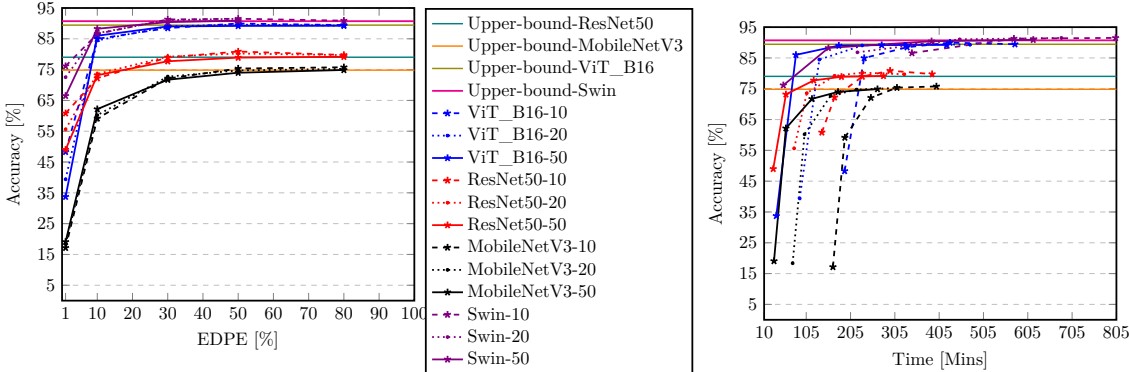

Figure 5: Performance scores on TinyImagenet at SSI values of 10, 20 and 50 for pretrained CNN and transformer models (Method=CRAIG).

consistent and stable performance, even when the EDPE reaches as low as 10%. Additionally, ViT_B16 consistently outperforms the ResNet50 model. This performance trend remains consistent across various $SSI$ values, including 10, 20, and 50. However, a significant decline in the performance of CNN models is observed for very low EDPE values, as illustrated in Fig. 6.

Overall, the stable performance of both architectures, even at minimal coreset sizes and time budgets, demonstrates the strong influence of pretraining on ImageNet21k. It indicates that a small number of CIFAR10 samples are sufficient to enhance the discriminative power of these models, thanks to the robust prior knowledge obtained through pretraining. These findings hold true for the Oxford-IIIT Pet dataset as well, as depicted in Fig. 6.

Results on TinyImageNet classification are shown in Fig. 5. We use all 4 architectures (CNN and transformer) and present scores for $SSI$ values of 10, 20 and 50. It can be seen that $SSI$, in general, does not impact model performance. Among the used models, the Swin transformer seems the most robust even at an EDPE value of 1%, whereas others deteriorate significantly. However, Swin Transformer is computationally expensive. Moreover, ViT_16 outperforms the CNN models at all coreset sizes. When compared with ResNet50, we see that at extremely low subsets, ResNet50 performs better than ViT_B16, thus being more stable than transformers at this low extent of subset selection. Finally, we observe, when pretrained, transformer models are stable in performance across different coreset sizes.

**Performance on OXFORD-PETS dataset** We further extended our experiments for the natural images by using the Oxford-IIIT Pet dataset which is not a subset of the ImageNet21k but contains matching images similar to ImageNet21K. From Fig. 6, we can see that when transformers are initialized with ImageNet21k they outperform CNNs across all the EDPE values and coreset methods. When the models are randomly initialized, it can be seen that transformers is suffering and CNNs are outperforming them. For more detailed results on the dataset, please check the tables provided in Section A.4.

**Performance on the medical dataset.** We extend our experiments to non-natural images. Fig. 7, illustrates results on the APTOS-2019 blindness detection dataset for ResNet50 and ViT_B16. It can be seen that the ResNet50 performs better than ViT_B16 at all coreset sizes, thereby confirming that the heavy pretraining in this case is not helping the transformers. We see that for CNN, random coreset selection is better or almost at par with GradMatch which itself performs better than the other coreset methods. For low coreset size (EDPE), random selection seems to be the most stable solution. Interestingly, when we look at the computational time, including the coreset selection time, random selection outperforms the other methods. For larger time investments, GradMatch is better than the random choice, however, for lower time budgets, random selection is the preferred choice.

**Learning semantic coherency in non-natural images.**

We analyze here how well transformers can learn the semantic coherency of the dataset, given that the data comprises non-natural images (very different from those used in pretraining).

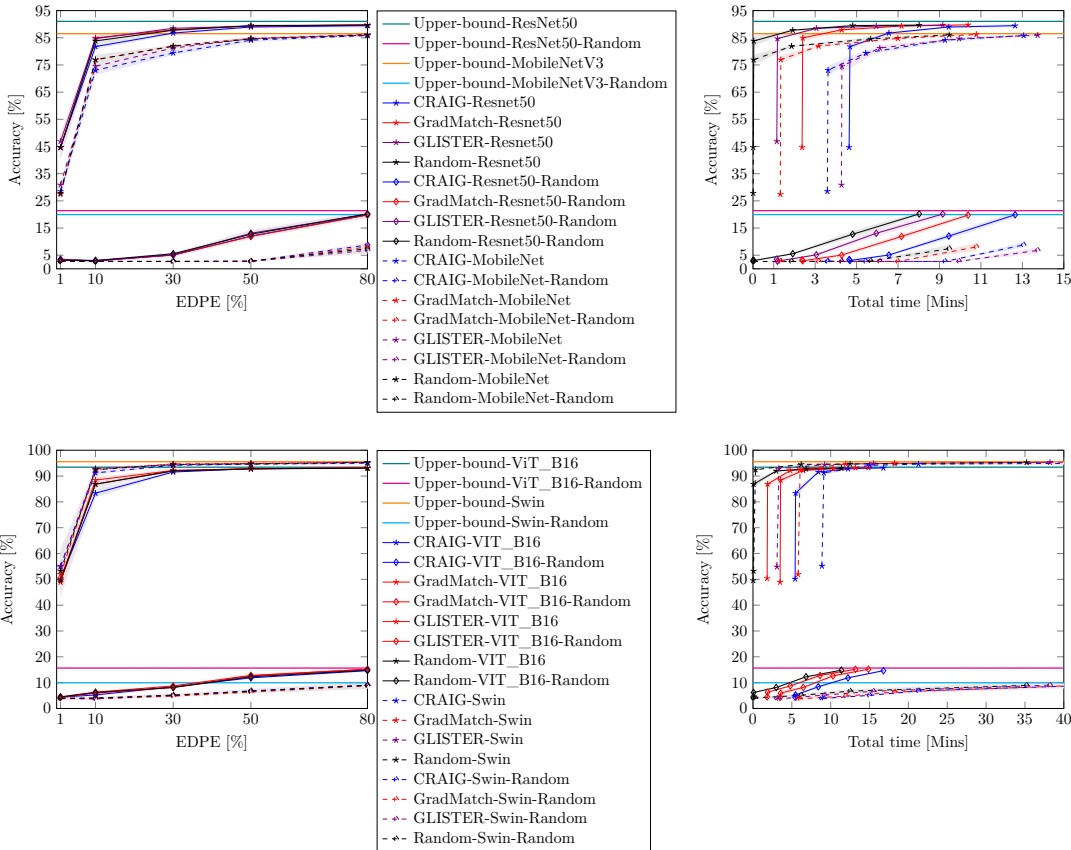

Figure 6: Performance scores of different networks with pretrained weights on ImageNet21k initialized and randomly initialized for various EDPE values and total time budgets for time, EDPE and performance on Oxford-IIIT Pet.

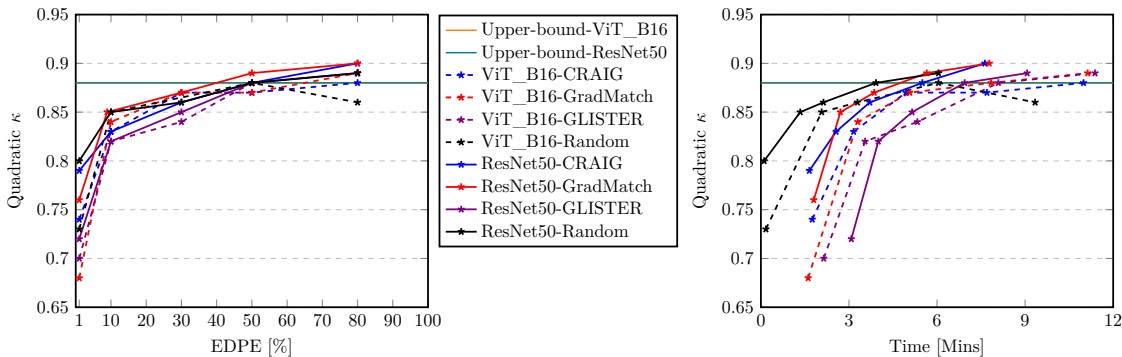

Figure 7: Performance on APTOS-2019 classification task as a function of EDPE and total time for pretrained ResNet50 and ViT_B16 for the four coreset selection methods (SSI=20).

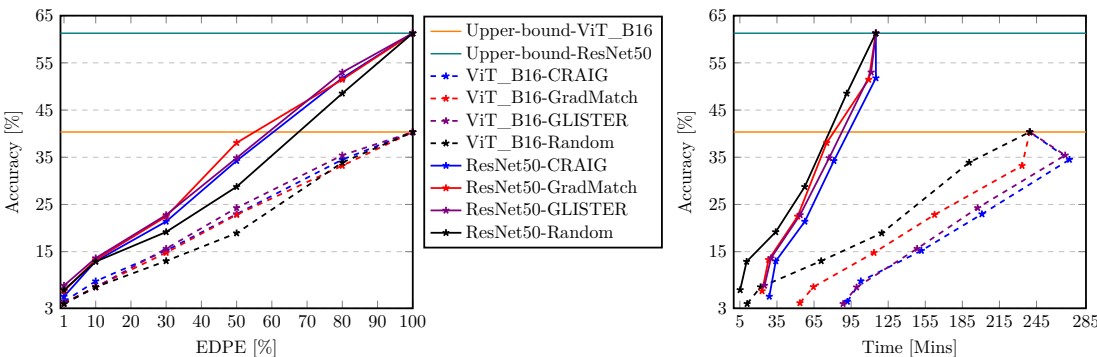

Figure 8: Performance comparison between ImageNet-21k weight initialized ViT_B16 and ResNet50 for time, EDPE and performance on UltraMNIST with different coreset methods, SSI=20.

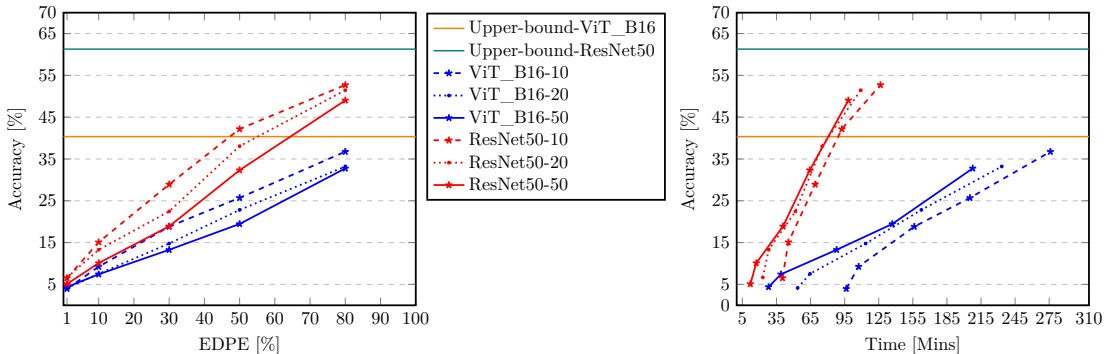

Figure 9: Performance comparison between pretrained ViT_B16 and ResNet50 for time, EDPE on UltraMNIST dataset (Method=GradMatch).

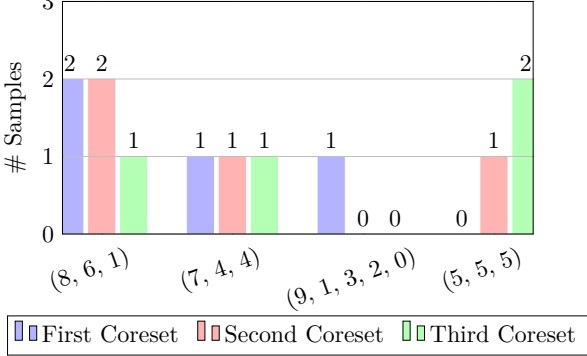

Figure 10: Number of samples present in different combinations of class 15 of UltraMNIST dataset in each coreset after coreset selection is done using GradMatch method.

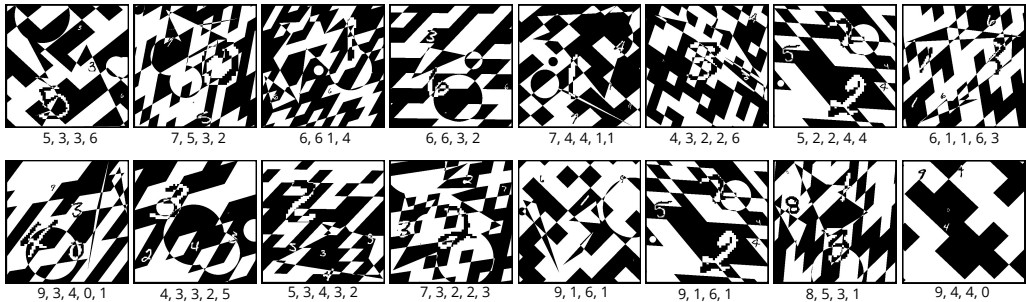

Figure 11: UltraMNIST samples for class label 17 from the two different coresets (top and bottom) for EDPE=1% obtained with GradMatch. Note that there is no overlap between the two coresets in terms of the combination of the digits in each image.

First, we study how the performance of the models is affected using different coreset selection methods. Fig. 8 presents the results for ViT_B16 and ResNet50. We see that independent of the coreset method, the CNN model is always superior to the transformer model across all coreset sizes. This observation is consistent across the various values of SSI (Fig. 9). UltraMNIST is a hard dataset to learn, and it is possible that with no similar pretraining, it is hard for transformers to learn well the desired semantic context.

In terms of the coreset size (EDPE), GradMatch outperforms the other methods while the random selection of coresets seems to perform the worst. However, in terms of the total time spent, random subset selection outperforms the other methods. This clearly indicates that although GradMatch might be a better selector of useful samples, the time spent scrutinizing the samples is not worth spending when compared to a random selection approach. For the SSI values shown in Fig. 9, we see that when evaluating performance with respect to EDPE, more number of coreset selection intervals are favoured, however, when evaluating in terms of the time, lesser selection cycles are preferred. This is because the coreset selection time is significant for most methods, however, EDPE does not take that into account.

We also look at why the performance of CNNs and transformer models is extremely low when a coreset size of as low as 1% is chosen. Our investigation reveals that this bottleneck is imposed by the fact that we conventionally choose an equal number of samples per class of the data. However, the complexity of the distribution of each class in UltraMNIST is very different. For example, class label 0 can only be formed by the combination of 3-5 occurrences of 0. However, 15 can be formed by many combinations of the digits. When selecting a coreset for this class, we tend to miss some of the combinations, thereby limiting the discriminative power of the trained model for such instances. This is also evident from the distributions shown in Fig. 10. There are certain combinations of 15 that miss from one coreset to another. Clearly, the model will catastrophically forget them as soon as they are completely out of the coresets. Interestingly, this is not the worst. If the distributions of the classes are not similar, uniform sampling could lead to a situation, where the distribution of the hard class could change completely between two coresets. Fig. 11 shows the training samples of class 17 from two different coresets of EDPE = 1%. As can be seen, the two sets are completely different which implies that it would be hard for the model to generalize for this class for very small coreset sizes. Overall, the takeaway is that the complexity of the distribution per class can be a valuable consideration for future research when constructing coresets.

### 4.3 Discussion

In this section, we summarize the insights from experiments and present answers to some intriguing questions.

*Should random coreset selection be a preferred choice?* Our results reveal that when comparing performance in terms of time budget, the random selection of the coresets is a more time-efficient choice than the other coreset selection methods. Clearly, one of the factors in favour of random selection is that the coreset selection time is almost zero. From a practical point of view, the impact of the small gains observed by other coreset selection methods would be diminished if the associated subset selection time is significant. Thus, unless

the coreset method is very efficient or parallelizable to the extent that the selection time can be made very small, random coreset selection should be a preferred choice for training models in a data-efficient manner.

*Choosing the right subset selection interval (SSI).* When selecting the appropriate Subset Selection Interval (SSI), it is important to note that there is no universally correct value, but optimization based on available resources can lead to better outcomes. Evaluating coreset selection methods for different SSI values reveals performance variations across different SSI values. Therefore optimizing the SSI value can improve performance. However, given the significant computational time associated with coreset selection, it is essential to ensure that the overall time remains below the desired threshold when choosing the SSI.

*Should coresets be chosen uniformly across the classes?* In constructing coresets, an alternative approach could involve emphasizing the approximation of class-specific distributions. This instructs sampling more instances from complex classes while sparse sampling from easier ones, enabling more efficient utilization of a limited coreset budget. Although challenges like class imbalance may arise, we support a non-uniform selection of coresets across classes, by adopting an adaptive strategy that deserves further investigation.

## 5 Conclusion

We have presented a systematic benchmarking scheme that facilitates a fair comparison of different coreset selection methods for data-efficient training. A thorough comparison of the various coreset methods on model performance at different coreset sizes is discussed. We demonstrated that the conventional concept of uniform subset sampling across the various classes of data is not the appropriate choice, suggesting the need to explore the benefits of adaptive sampling based on the complexity of each class's data distribution. This alternative solution requires further research and exploration. We further performed a comparative study between CNNs and transformers and demonstrated coreset selection scenarios where transformers outperform CNNs and vice versa. In our experiments, we discovered multiple cases where the random coreset selection method consistently outperformed other methods. Consequently, it would be unjust to claim that coreset methods universally outperform random selection. Instead, it is more accurate to state that random selection is generally sufficient for most scenarios, with only marginal performance differences observed in cases where it falls short compared to other methods. Overall, we hope that the insights presented in this paper provide a better understanding of the usage of various coreset selection methods with CNNs and transformers, thereby helping to contribute better towards developing energy-efficient systems for model training.

# A  Appendix

## A.1  Introduction

We have also provided quantitative results, further illustrating the observations and analysis presented in the main manuscript.

An improved version of the code will be made publicly available with the final version of the paper.

## A.2  Exploring Coreset selection Methods in Detail

**CRAIG.** CRAIG aims to select an optimal subset $\mathcal{S}^*$ by solving the following equation:

$$\mathcal{S}^* = \arg\min_{\mathcal{S} \subseteq \mathcal{D}, \gamma_j \geq 0 \ \forall_j} |\mathcal{S}|, \ \text{s.t.}$$
$$\max_{\mathbf{w} \in \mathcal{W}} \| \sum_{i \in \mathcal{D}} \nabla f_i(\mathbf{w}) - \sum_{j \in \mathcal{S}} \gamma_j \nabla f_j(\mathbf{w}) \| \leq \mathcal{M}. \tag{3}$$

In the equation above, $\mathcal{D}$ denotes the training dataset, and $\mathcal{S}$ represents the selected subset with a budget $\mathcal{K}$. The optimization problem aims to find the smallest subset $\mathcal{S} \subseteq \mathcal{D}$ along with per-element stepsizes $\gamma_j > 0$ that approximate the full gradient with an error no greater than $\mathcal{M}$ for all possible optimization parameters $\mathbf{w} \in \mathcal{W}$. It is important to note that the CRAIG method can be utilized to select data subsets at different epochs.

**GradMatch.** We further explain GradMatch in a detailed step-by-step breakdown of the optimization problem formulation, the utilization of the Orthogonal Matching Pursuit (OMP) algorithm, and additional details regarding the computation of the error term used in GradMatch. Let $\mathcal{S}^t$ represent subsets for $t = 1, \cdots, T$, with $p$ denoting the data points from the training or validation set. The loss function is defined as $\mathcal{L}(\mathbf{w}) = \sum_{i \in \mathcal{W}} \mathcal{L}(\mathbf{x_i}, \mathbf{y_i}, \mathbf{w})$, where $\mathcal{L} = \mathcal{L}_{\mathcal{T}}$ when $\mathcal{W} = \mathcal{D}$ (training set) and $\mathcal{L} = \mathcal{L}_{\mathcal{V}}$ when $\mathcal{W} = \mathcal{V}$ (validation set).

$$\text{Err}(\boldsymbol{\phi}^t, \mathcal{S}^t, \mathcal{L}, \mathcal{L}_{\mathcal{T}}, \mathbf{w_t}) = \left\| \sum_{i \in \mathcal{S}^t} \phi \nabla_{\mathbf{w}} \mathcal{L}_{\mathcal{T}}{}^i(\mathbf{w_t}) - \nabla_{\mathbf{w}} \mathcal{L}(\mathbf{w_t}) \right\| \tag{4}$$

The error term $\text{Err}(\boldsymbol{\phi}^t, \mathcal{S}^t, \mathcal{L}, \mathcal{L}_{\mathcal{T}}, \mathbf{w_t})$ is computed using Equation (4), where $\phi$ is a weight vector containing the weights for each data instance, $\mathcal{K}$ represents the budget for the subset, and the L2 norm is applied. The objective is to minimize this error by solving the optimization problem using the Orthogonal Matching Pursuit (OMP) algorithm.

**GLISTER.** It is an approach that tackles the challenge of selecting a representative subset from the training data while optimizing model performance and efficiency. In GLISTER, the selection process involves maximizing a specific objective function, taking into account the log-likelihood on the validation set and utilizing gradient information for subset evaluation. GLISTER follows below approach to solve the optimization problem:

$$\mathcal{S}^{t+1} = \underset{\mathcal{S} \subseteq \mathcal{D}, |\mathcal{S}| \leq \mathcal{K}}{\text{argmax}} \ G_{\mathbf{w}^t}(\mathcal{S}), \text{where} \tag{5}$$
$$G_{\mathbf{w}}(\mathcal{S}) = \mathcal{LL}_{\mathcal{V}}(\mathbf{w} + \eta \nabla_{\mathbf{w}} \mathcal{LL}_{\mathcal{D}}(\mathbf{w}, \mathcal{S}), \mathcal{V}),$$

In the above equations, $\mathcal{D}$ represents the training set, $\mathcal{V}$ refers to the validation set, and $\mathcal{LL}_{\mathcal{V}}$ represents the log-likelihood on the validation set. The step size is denoted by $\eta$, and $\mathcal{K}$ represents the budget for the subset. GLISTER solves a bi-level optimization problem, where the first problem is addressed by employing an online one-step meta approximation through a single gradient step. The second approximation is achieved using the Taylor-Series approximation, implemented via the greedy search algorithm.

| EDPE (%) | Epoch | Total Time (Mins) | | | Total Selection Time (Mins) | | | Accuracy | | |
|---|---|---|---|---|---|---|---|---|---|---|
| | 105 | | 147.46 | | | - | | | 99.01 | |
| 100 | 89 | | 125.88 | | | - | | | 98.61 | |
| | 79 | | 112.06 | | | - | | | 98.86 | |
| | | SSI=10 | SSI=20 | SSI=50 | SSI=10 | SSI=20 | SSI=50 | SSI=10 | SSI=20 | SSI=50 |
| | 105 | 240.75 | 227.10 | 217.38 | 31.20 | 15.41 | 6.24 | 98.89 | 98.94 | 98.95 |
| 90 | 89 | 203.89 | 193.08 | 183.40 | 24.96 | 12.33 | 3.12 | 98.86 | 98.89 | 98.87 |
| | 79 | 180.12 | 169.14 | 162.60 | 21.84 | 9.25 | 3.12 | 98.73 | 98.78 | 98.78 |
| | 105 | 217.94 | 201.95 | 193.10 | 31.20 | 15.75 | 6.25 | 98.86 | 98.98 | 98.94 |
| 80 | 89 | 184.33 | 171.51 | 162.58 | 24.96 | 12.60 | 3.12 | 98.79 | 98.94 | 98.87 |
| | 79 | 162.73 | 149.93 | 144.09 | 21.84 | 9.45 | 3.12 | 98.80 | 98.84 | 98.80 |
| | 105 | 196.18 | 181.35 | 170.78 | 30.94 | 15.75 | 6.35 | 98.94 | 98.82 | 98.82 |
| 70 | 89 | 165.93 | 154.08 | 143.66 | 24.75 | 12.60 | 3.17 | 98.89 | 98.72 | 98.79 |
| | 79 | 146.63 | 134.69 | 127.53 | 21.66 | 9.45 | 3.17 | 98.92 | 98.78 | 98.69 |
| | 105 | 171.45 | 156.75 | 147.65 | 30.96 | 15.81 | 6.30 | 98.99 | 98.94 | 98.98 |
| 60 | 89 | 144.73 | 132.99 | 123.84 | 24.76 | 12.65 | 3.15 | 98.96 | 98.93 | 98.92 |
| | 79 | 127.79 | 115.94 | 109.91 | 21.67 | 9.48 | 3.15 | 98.89 | 98.80 | 98.84 |
| | 105 | 148.45 | 132.09 | 122.60 | 31.58 | 15.67 | 6.28 | 98.92 | 98.97 | 98.76 |
| 50 | 89 | 124.98 | 111.86 | 102.38 | 25.26 | 12.53 | 3.14 | 98.92 | 98.90 | 98.67 |
| | 79 | 110.22 | 97.18 | 90.84 | 22.10 | 9.40 | 3.14 | 98.81 | 98.86 | 98.71 |
| | 105 | 125.13 | 109.69 | 100.47 | 30.80 | 15.53 | 6.33 | 98.87 | 98.73 | 98.96 |
| 40 | 89 | 105.26 | 92.90 | 83.63 | 24.64 | 12.42 | 3.16 | 98.85 | 98.65 | 98.92 |
| | 79 | 92.96 | 80.60 | 74.43 | 21.56 | 9.32 | 3.16 | 98.78 | 98.59 | 98.82 |
| | 105 | 99.90 | 87.05 | 78.51 | 29.97 | 16.96 | 8.02 | 98.69 | 98.67 | 98.59 |
| 30 | 89 | 83.68 | 73.41 | 64.20 | 23.97 | 13.57 | 4.01 | 98.65 | 98.56 | 98.49 |
| | 79 | 73.80 | 63.11 | 57.25 | 20.97 | 10.17 | 4.01 | 98.57 | 98.58 | 98.46 |
| | 105 | 75.67 | 60.52 | 52.25 | 30.13 | 15.07 | 6.05 | 98.64 | 98.58 | 98.63 |
| 20 | 89 | 62.91 | 50.78 | 42.39 | 24.10 | 12.05 | 3.02 | 98.52 | 98.58 | 98.55 |
| | 79 | 55.33 | 43.21 | 37.76 | 21.09 | 9.04 | 3.02 | 98.60 | 98.38 | 98.57 |
| | 105 | 54.29 | 39.01 | 29.24 | 30.07 | 14.95 | 5.57 | 98.25 | 98.28 | 98.25 |
| 10 | 89 | 44.81 | 32.59 | 23.07 | 24.06 | 11.96 | 2.78 | 98.22 | 98.21 | 98.18 |
| | 79 | 39.50 | 27.30 | 20.82 | 21.05 | 8.97 | 2.78 | 98.25 | 98.28 | 98.25 |
| | 105 | 30.38 | 16.49 | 8.02 | 27.98 | 14.09 | 5.62 | 97.49 | 97.59 | 97.39 |
| 1 | 89 | 24.44 | 13.32 | 4.86 | 22.39 | 11.27 | 2.81 | 97.49 | 97.59 | 97.39 |
| | 79 | 21.42 | 10.28 | 4.64 | 19.59 | 8.45 | 2.81 | 97.49 | 97.59 | 97.39 |

Table A1: Performance scores for ViT-B_16 with ImageNet 21K weights on CIFAR10 and its Effective data per epoch (EDPE), method=CRAIG

### A.3   Results

### A.3.1   CRAIG Method Results on CIFAR10

We initially started our experiments on the CIFAR10 dataset. Tables [A1, A2, A3, A4] gives a comparative study for these models. We used Fig.3 and Fig. 4 to demonstrate that CNN underperforms across all coreset sizes than transformers under the pretraining setting. We can also see that at tiny coreset sizes, transformers are more stable than CNNs for the CIFAR10 dataset.

### A.3.2   CRAIG, GradMatch, GLISTER and Random Results on Tiny ImageNet

This section shows results with different subsets methods on Tiny Imagenet with ViT_B16, ResNet50, and MobileNet.

### A.4   CRAIG, GradMatch, GLISTER and Random Results on Oxford-IIIT Pet

To further study the effect on ImageNet21k pretaining and random initialized models with the coreset methods. We further conducted experiments on the Oxford-IIIT Pet dataset. Each experiment in the tables

| EDPE (%) | Epoch | Total Time (Mins) | | | Total Selection Time (Mins) | | | Accuracy | | |
|---|---|---|---|---|---|---|---|---|---|---|
| 100 | 105 | 147.46 | | | - | | | 66.84 | | |
| | 89 | 125.88 | | | - | | | 62.21 | | |
| | 79 | 112.06 | | | - | | | 64.15 | | |
| | | *SSI=10* | *SSI=20* | *SSI=50* | *SSI=10* | *SSI=20* | *SSI=50* | *SSI=10* | *SSI=20* | *SSI=50* |
| 90 | 105 | 240.75 | 227.10 | 217.38 | 31.20 | 15.41 | 6.24 | 66.66 | 66.76 | 67.03 |
| | 89 | 203.89 | 193.08 | 183.40 | 24.96 | 12.33 | 3.12 | 66.42 | 66.46 | 66.85 |
| | 79 | 180.12 | 169.14 | 162.60 | 21.84 | 9.25 | 3.12 | 65.47 | 66.16 | 66.29 |
| 80 | 105 | 217.94 | 201.95 | 193.10 | 31.20 | 15.75 | 6.25 | 65.42 | 65.82 | 66.44 |
| | 89 | 184.33 | 171.51 | 162.58 | 24.96 | 12.60 | 3.12 | 65.31 | 65.16 | 66.08 |
| | 79 | 162.73 | 149.93 | 144.09 | 21.84 | 9.45 | 3.12 | 64.02 | 64.12 | 65.34 |
| 70 | 105 | 196.18 | 181.35 | 170.78 | 30.94 | 15.75 | 6.35 | 63.77 | 63.53 | 64.88 |
| | 89 | 165.93 | 154.08 | 143.66 | 24.75 | 12.60 | 3.17 | 63.28 | 63.52 | 64.59 |
| | 79 | 146.63 | 134.69 | 127.53 | 21.66 | 9.45 | 3.17 | 62.03 | 62.73 | 63.56 |
| 60 | 105 | 171.45 | 156.75 | 147.65 | 30.96 | 15.81 | 6.30 | 61.70 | 62.56 | 63.12 |
| | 89 | 144.73 | 132.99 | 123.84 | 24.76 | 12.65 | 3.15 | 61.21 | 61.93 | 62.60 |
| | 79 | 127.79 | 115.94 | 109.91 | 21.67 | 9.48 | 3.15 | 60.83 | 61.30 | 61.65 |
| 50 | 105 | 148.45 | 132.09 | 122.60 | 31.58 | 15.67 | 6.28 | 61.15 | 61.44 | 62.02 |
| | 89 | 124.98 | 111.86 | 102.38 | 25.26 | 12.53 | 3.14 | 60.78 | 61.07 | 61.99 |
| | 79 | 110.22 | 97.18 | 90.84 | 22.10 | 9.40 | 3.14 | 59.71 | 59.87 | 61.22 |
| 40 | 105 | 125.13 | 109.69 | 100.47 | 30.80 | 15.53 | 6.33 | 59.28 | 60.17 | 60.10 |
| | 89 | 105.26 | 92.90 | 83.63 | 24.64 | 12.42 | 3.16 | 58.99 | 59.70 | 59.39 |
| | 79 | 92.96 | 80.60 | 74.43 | 21.56 | 9.32 | 3.16 | 58.34 | 59.13 | 59.10 |
| 30 | 105 | 99.90 | 87.05 | 78.51 | 29.97 | 16.96 | 8.02 | 57.75 | 57.67 | 57.79 |
| | 89 | 83.68 | 73.41 | 64.20 | 23.97 | 13.57 | 4.01 | 57.15 | 57.27 | 57.53 |
| | 79 | 73.80 | 63.11 | 57.25 | 20.97 | 10.17 | 4.01 | 56.15 | 56.69 | 57.52 |
| 20 | 105 | 75.67 | 60.52 | 52.25 | 30.13 | 15.07 | 6.05 | 52.26 | 55.58 | 53.70 |
| | 89 | 62.91 | 50.78 | 42.39 | 24.10 | 12.05 | 3.02 | 49.50 | 55.56 | 52.49 |
| | 79 | 55.33 | 43.21 | 37.76 | 21.09 | 9.04 | 3.02 | 49.55 | 53.98 | 52.85 |
| 10 | 105 | 54.29 | 39.01 | 29.24 | 30.07 | 14.95 | 5.57 | 51.99 | 51.59 | 50.57 |
| | 89 | 44.81 | 32.59 | 23.07 | 24.06 | 11.96 | 2.78 | 50.86 | 51.53 | 50.56 |
| | 79 | 39.50 | 27.30 | 20.82 | 21.05 | 8.97 | 2.78 | 50.48 | 49.25 | 49.81 |
| 1 | 105 | 30.38 | 16.49 | 8.02 | 27.98 | 14.09 | 5.62 | 32.73 | 33.24 | 33.04 |
| | 89 | 24.44 | 13.32 | 4.86 | 22.39 | 11.27 | 2.81 | 32.47 | 33.24 | 33.04 |
| | 79 | 21.42 | 10.28 | 4.64 | 19.59 | 8.45 | 2.81 | 31.85 | 32.72 | 33.04 |

Table A2: Performance scores for ViT-B_16 with Random Initialization on CIFAR10 and its Effective data per epoch (EDPE), method=CRAIG

| EDPE (%) | Epoch | Total Time (Mins) | | | Total Selection Time (Mins) | | | Accuracy | | |
|---|---|---|---|---|---|---|---|---|---|---|
| 100 | 105 | 65.45 | | | - | | | 97.96 | | |
| | 89 | 55.87 | | | - | | | 97.10 | | |
| | 79 | 49.74 | | | - | | | 97.09 | | |
| | | *SSI=10* | *SSI=20* | *SSI=50* | *SSI=10* | *SSI=20* | *SSI=50* | *SSI=10* | *SSI=20* | *SSI=50* |
| 90 | 105 | 152.17 | 144.45 | 134.64 | 16.73 | 11.36 | 5.05 | 97.67 | 97.76 | 97.39 |
| | 89 | 129.03 | 122.73 | 113.18 | 13.39 | 9.09 | 2.52 | 97.67 | 97.70 | 97.33 |
| | 79 | 114.02 | 107.35 | 100.41 | 11.71 | 6.81 | 2.52 | 97.52 | 97.70 | 97.37 |
| 80 | 105 | 130.57 | 118.59 | 120.97 | 15.93 | 7.69 | 3.50 | 97.90 | 97.67 | 97.56 |
| | 89 | 110.58 | 100.79 | 102.00 | 12.74 | 6.15 | 1.75 | 97.81 | 97.67 | 97.30 |
| | 79 | 97.65 | 88.28 | 90.38 | 11.15 | 4.61 | 1.75 | 97.57 | 97.39 | 97.17 |
| 70 | 105 | 118.97 | 113.80 | 107.29 | 15.44 | 7.99 | 3.26 | 97.74 | 97.66 | 97.69 |
| | 89 | 100.80 | 96.79 | 90.51 | 12.35 | 6.39 | 1.63 | 97.58 | 97.60 | 97.58 |
| | 79 | 89.10 | 84.82 | 80.31 | 10.80 | 4.79 | 1.63 | 97.62 | 97.50 | 97.44 |
| 60 | 105 | 102.69 | 96.62 | 91.03 | 15.36 | 7.91 | 2.98 | 97.92 | 97.80 | 97.56 |
| | 89 | 86.85 | 76.66 | 76.65 | 12.29 | 6.33 | 1.49 | 97.92 | 97.80 | 97.54 |
| | 79 | 76.71 | 71.75 | 67.99 | 10.75 | 4.74 | 1.49 | 97.82 | 97.36 | 97.54 |
| 50 | 105 | 84.95 | 77.55 | 75.40 | 13.88 | 6.59 | 2.77 | 97.82 | 97.59 | 97.55 |
| | 89 | 71.74 | 65.81 | 63.35 | 11.11 | 5.27 | 1.38 | 97.73 | 97.54 | 97.46 |
| | 79 | 63.30 | 57.45 | 56.14 | 9.72 | 3.95 | 1.38 | 97.44 | 97.45 | 97.27 |
| 40 | 105 | 72.94 | 64.51 | 61.54 | 13.65 | 6.51 | 2.70 | 97.69 | 97.55 | 97.59 |
| | 89 | 61.60 | 54.78 | 51.64 | 10.92 | 5.21 | 1.35 | 97.61 | 97.55 | 97.59 |
| | 79 | 54.44 | 47.81 | 45.89 | 9.55 | 3.90 | 1.35 | 97.51 | 97.31 | 97.47 |
| 30 | 105 | 55.72 | 49.19 | 46.30 | 13.29 | 6.42 | 2.70 | 97.49 | 97.52 | 97.25 |
| | 89 | 46.86 | 41.66 | 38.58 | 10.63 | 5.13 | 1.35 | 97.46 | 97.52 | 97.16 |
| | 79 | 41.35 | 36.16 | 34.28 | 9.30 | 3.85 | 1.35 | 97.16 | 97.28 | 97.25 |
| 20 | 105 | 42.39 | 34.51 | 31.11 | 13.92 | 6.43 | 2.68 | 97.28 | 97.16 | 97.07 |
| | 89 | 35.39 | 29.07 | 25.56 | 11.13 | 5.14 | 1.34 | 97.28 | 97.02 | 96.98 |
| | 79 | 31.15 | 24.97 | 22.71 | 9.74 | 3.86 | 1.34 | 97.16 | 96.82 | 96.91 |
| 10 | 105 | 27.07 | 20.67 | 17.29 | 12.87 | 6.28 | 2.67 | 96.67 | 96.70 | 96.38 |
| | 89 | 22.47 | 17.36 | 13.87 | 10.30 | 5.03 | 1.33 | 96.56 | 96.70 | 96.36 |
| | 79 | 19.83 | 14.73 | 12.47 | 9.01 | 3.77 | 1.33 | 96.28 | 96.67 | 96.27 |
| 1 | 105 | 14.30 | 7.87 | 4.13 | 12.87 | 6.43 | 2.70 | 92.31 | 93.00 | 93.14 |
| | 89 | 11.52 | 6.38 | 2.58 | 10.30 | 5.15 | 1.35 | 92.31 | 93.00 | 93.12 |
| | 79 | 10.10 | 4.95 | 2.44 | 9.01 | 3.86 | 1.35 | 92.31 | 93.00 | 93.00 |

Table A3: Performance scores for ResNet50V2 with ImageNet21k weights Initialization on CIFAR10 and its Effective data per epoch (EDPE), method=CRAIG

| EDPE (%) | Epoch | Total Time (Mins) | | | Total Selection Time (Mins) | | | Accuracy | | |
|---|---|---|---|---|---|---|---|---|---|---|
| 100 | 105 | 65.45 | | | - | | | 88.90 | | |
| | 89 | 55.87 | | | - | | | 88.64 | | |
| | 79 | 49.74 | | | - | | | 88.10 | | |
| | | *SSI=10* | *SSI=20* | *SSI=50* | *SSI=10* | *SSI=20* | *SSI=50* | *SSI=10* | *SSI=20* | *SSI=50* |
| 90 | 105 | 152.17 | 144.45 | 134.64 | 16.73 | 11.36 | 5.05 | 89.59 | 89.46 | 89.64 |
| | 89 | 129.03 | 122.73 | 113.18 | 13.39 | 9.09 | 2.52 | 89.31 | 89.24 | 89.48 |
| | 79 | 114.02 | 107.35 | 100.41 | 11.71 | 6.81 | 2.52 | 89.03 | 89.10 | 89.03 |
| 80 | 105 | 130.57 | 118.59 | 120.97 | 15.93 | 7.69 | 3.50 | 88.70 | 89.06 | 89.33 |
| | 89 | 110.58 | 100.79 | 102.00 | 12.74 | 6.15 | 1.75 | 88.37 | 88.80 | 89.18 |
| | 79 | 97.65 | 88.28 | 90.38 | 11.15 | 4.61 | 1.75 | 88.05 | 88.34 | 88.47 |
| 70 | 105 | 118.97 | 113.80 | 107.29 | 15.44 | 7.99 | 3.26 | 87.31 | 87.38 | 87.97 |
| | 89 | 100.80 | 96.79 | 90.51 | 12.35 | 6.39 | 1.63 | 86.97 | 86.75 | 87.91 |
| | 79 | 89.10 | 84.82 | 80.31 | 10.80 | 4.79 | 1.63 | 86.51 | 86.54 | 87.04 |
| 60 | 105 | 102.69 | 96.62 | 91.03 | 15.36 | 7.91 | 2.98 | 86.14 | 86.01 | 86.79 |
| | 89 | 86.85 | 76.66 | 76.65 | 12.29 | 6.33 | 1.49 | 85.94 | 85.47 | 86.55 |
| | 79 | 76.71 | 71.75 | 67.99 | 10.75 | 4.74 | 1.49 | 85.24 | 84.98 | 85.80 |
| 50 | 105 | 84.95 | 77.55 | 75.40 | 13.88 | 6.59 | 2.77 | 83.31 | 83.70 | 84.60 |
| | 89 | 71.74 | 65.81 | 63.35 | 11.11 | 5.27 | 1.38 | 82.79 | 83.70 | 84.25 |
| | 79 | 63.30 | 57.45 | 56.14 | 9.72 | 3.95 | 1.38 | 82.14 | 82.56 | 83.81 |
| 40 | 105 | 72.94 | 64.51 | 61.54 | 13.65 | 6.51 | 2.70 | 80.91 | 80.98 | 81.99 |
| | 89 | 61.60 | 54.78 | 51.64 | 10.92 | 5.21 | 1.35 | 80.32 | 80.66 | 81.76 |
| | 79 | 54.44 | 47.81 | 45.89 | 9.55 | 3.90 | 1.35 | 79.52 | 79.75 | 81.21 |
| 30 | 105 | 55.72 | 49.19 | 46.30 | 13.29 | 6.42 | 2.70 | 78.02 | 77.77 | 77.76 |
| | 89 | 46.86 | 41.66 | 38.58 | 10.63 | 5.13 | 1.35 | 77.20 | 77.25 | 77.69 |
| | 79 | 41.35 | 36.16 | 34.28 | 9.30 | 3.85 | 1.35 | 76.16 | 76.46 | 77.06 |
| 20 | 105 | 42.39 | 34.51 | 31.11 | 13.92 | 6.43 | 2.68 | 73.00 | 72.28 | 71.22 |
| | 89 | 35.39 | 29.07 | 25.56 | 11.13 | 5.14 | 1.34 | 72.50 | 71.75 | 71.16 |
| | 79 | 31.15 | 24.97 | 22.71 | 9.74 | 3.86 | 1.34 | 70.47 | 70.32 | 70.23 |
| 10 | 105 | 27.07 | 20.67 | 17.29 | 12.87 | 6.28 | 2.67 | 63.40 | 62.56 | 61.27 |
| | 89 | 22.47 | 17.36 | 13.87 | 10.30 | 5.03 | 1.33 | 62.77 | 62.45 | 61.27 |
| | 79 | 19.83 | 14.73 | 12.47 | 9.01 | 3.77 | 1.33 | 62.61 | 61.60 | 60.73 |
| 1 | 105 | 14.30 | 7.87 | 4.13 | 12.87 | 6.43 | 2.70 | 27.29 | 27.72 | 28.98 |
| | 89 | 11.52 | 6.38 | 2.58 | 10.30 | 5.15 | 1.35 | 27.24 | 27.72 | 28.80 |
| | 79 | 10.10 | 4.95 | 2.44 | 9.01 | 3.86 | 1.35 | 27.24 | 26.84 | 28.41 |

Table A4: Performance scores for ResNet50V2 with Random Initialization on CIFAR10 and its Effective data per epoch (EDPE), method=CRAIG

| EDPE (%) | Epoch | Total Time (Mins) | | | Total Selection Time (Mins) | | | Accuracy | | |
|---|---|---|---|---|---|---|---|---|---|---|
| | 105 | | 147.46 | | | - | | | 99.01 | |
| 100 | 89 | | 125.88 | | | - | | | 98.61 | |
| | 79 | | 112.06 | | | - | | | 98.86 | |
| | | *SSI=10* | *SSI=20* | *SSI=50* | *SSI=10* | *SSI=20* | *SSI=50* | *SSI=10* | *SSI=20* | *SSI=50* |
| | 105 | 209.07 | 193.99 | 183.22 | 30.87 | 15.80 | 5.03 | 99.08 | 99.03 | 99.06 |
| 80 | 89 | 176.77 | 164.72 | 154.59 | 24.70 | 12.64 | 2.51 | 99.08 | 99.03 | 99.06 |
| | 79 | 156.05 | 143.92 | 136.95 | 21.61 | 9.48 | 2.51 | 99.04 | 99.03 | 99.06 |
| | 105 | 138.02 | 123.88 | 116.13 | 26.94 | 12.80 | 5.04 | 99.04 | 98.96 | 98.94 |
| 50 | 89 | 116.32 | 105.01 | 97.29 | 21.55 | 10.24 | 2.52 | 99.03 | 98.96 | 98.94 |
| | 79 | 102.61 | 91.43 | 86.27 | 18.85 | 7.68 | 2.52 | 99.03 | 98.96 | 98.94 |
| | 105 | 48.38 | 35.97 | 28.16 | 25.23 | 12.82 | 5.01 | 98.87 | 98.72 | 98.53 |
| 10 | 89 | 40.02 | 30.09 | 22.34 | 20.19 | 10.26 | 2.50 | 98.87 | 98.72 | 98.53 |
| | 79 | 35.29 | 25.32 | 20.14 | 17.66 | 7.69 | 2.50 | 98.87 | 98.71 | 98.53 |
| | 105 | 27.15 | 14.95 | 7.25 | 24.84 | 12.63 | 4.93 | 97.90 | 97.88 | 97.57 |
| 1 | 89 | 21.85 | 12.09 | 4.45 | 19.87 | 10.10 | 2.46 | 97.89 | 97.88 | 97.57 |
| | 79 | 19.15 | 9.34 | 4.23 | 17.38 | 8.45 | 2.46 | 97.87 | 97.87 | 97.57 |

Table A5: Performance scores for ViT-B_16 with ImageNet 21K weights on CIFAR10 and its Effective data per epoch (EDPE), method=GradMatch

| EDPE (%) | Epoch | Total Time (Mins) | | | Total Selection Time (Mins) | | | Accuracy | | |
|---|---|---|---|---|---|---|---|---|---|---|
| | 105 | | 147.46 | | | - | | | 66.84 | |
| 100 | 89 | | 125.88 | | | - | | | 62.21 | |
| | 79 | | 112.06 | | | - | | | 64.15 | |
| | | *SSI=10* | *SSI=20* | *SSI=50* | *SSI=10* | *SSI=20* | *SSI=50* | *SSI=10* | *SSI=20* | *SSI=50* |
| | 105 | 209.07 | 193.99 | 183.22 | 30.87 | 15.80 | 5.03 | 66.79 | 67.05 | 67.22 |
| 80 | 89 | 176.77 | 164.72 | 154.59 | 24.70 | 12.64 | 2.51 | 66.33 | 66.70 | 66.76 |
| | 79 | 156.05 | 143.92 | 136.95 | 21.61 | 9.48 | 2.51 | 65.92 | 65.45 | 66.08 |
| | 105 | 138.02 | 123.88 | 116.13 | 26.94 | 12.80 | 5.04 | 62.90 | 63.56 | 63.53 |
| 50 | 89 | 116.32 | 105.01 | 97.29 | 21.55 | 10.24 | 2.52 | 62.66 | 63.00 | 63.08 |
| | 79 | 102.61 | 91.43 | 86.27 | 18.85 | 7.68 | 2.52 | 61.57 | 62.74 | 62.49 |
| | 105 | 48.38 | 35.97 | 28.16 | 25.23 | 12.82 | 5.01 | 50.53 | 50.16 | 49.04 |
| 10 | 89 | 40.02 | 30.09 | 22.34 | 20.19 | 10.26 | 2.50 | 50.45 | 50.00 | 48.95 |
| | 79 | 35.29 | 25.32 | 20.14 | 17.66 | 7.69 | 2.50 | 50.06 | 49.44 | 48.45 |
| | 105 | 27.15 | 14.95 | 7.25 | 24.84 | 12.63 | 4.93 | 27.77 | 27.55 | 30.00 |
| 1 | 89 | 21.85 | 12.09 | 4.45 | 19.87 | 10.10 | 2.46 | 27.51 | 27.55 | 30.00 |
| | 79 | 19.15 | 9.34 | 4.23 | 17.38 | 8.45 | 2.46 | 22.27 | 27.55 | 30.00 |

Table A6: Performance scores for ViT-B_16 with random weights weights on CIFAR10 and its Effective data per epoch (EDPE), method=GradMatch

| EDPE (%) | Epoch | Total Time (Mins) | | | Total Selection Time (Mins) | | | Accuracy | | |
|---|---|---|---|---|---|---|---|---|---|---|
| 100 | 105 | 65.45 | | | - | | | 97.96 | | |
| | 89 | 55.87 | | | - | | | 97.10 | | |
| | 79 | 49.74 | | | - | | | 97.09 | | |
| | | SSI=10 | SSI=20 | SSI=50 | SSI=10 | SSI=20 | SSI=50 | SSI=10 | SSI=20 | SSI=50 |
| 80 | 105 | 124.89 | 119.48 | 115.79 | 11.43 | 6.02 | 2.33 | 97.93 | 98.00 | 97.88 |
| | 89 | 105.98 | 101.65 | 97.99 | 9.15 | 4.82 | 1.16 | 97.93 | 97.94 | 97.81 |
| | 79 | 93.61 | 89.21 | 86.77 | 8.00 | 3.61 | 1.16 | 97.90 | 97.93 | 97.80 |
| 50 | 105 | 82.37 | 77.27 | 73.27 | 11.64 | 6.54 | 2.54 | 97.87 | 97.90 | 97.63 |
| | 89 | 69.65 | 65.57 | 61.61 | 9.31 | 5.23 | 1.27 | 97.85 | 97.90 | 97.58 |
| | 79 | 61.47 | 57.25 | 54.59 | 8.15 | 3.92 | 1.27 | 97.80 | 97.72 | 97.58 |
| 10 | 105 | 30.69 | 22.63 | 18.14 | 15.95 | 7.9 | 3.40 | 97.70 | 97.30 | 96.65 |
| | 89 | 25.39 | 18.94 | 14.33 | 12.76 | 6.31 | 1.70 | 97.60 | 97.25 | 96.65 |
| | 79 | 22.39 | 15.96 | 12.92 | 5.15 | 4.73 | 1.70 | 97.48 | 97.07 | 96.56 |
| 1 | 105 | 11.31 | 6.65 | 3.62 | 9.84 | 5.17 | 2.15 | 93.14 | 92.31 | 92.91 |
| | 89 | 9.13 | 5.40 | 2.33 | 7.87 | 4.14 | 1.07 | 92.93 | 92.29 | 92.91 |
| | 79 | 8.01 | 4.23 | 2.19 | 6.89 | 3.10 | 1.07 | 92.93 | 92.29 | 92.91 |

Table A7: Performance scores for ResNet50V2 with ImageNet21k weights Initialization on CIFAR10 and its Effective data per epoch (EDPE), method=GradMatch

| EDPE (%) | Epoch | Total Time (Mins) | | | Total Selection Time (Mins) | | | Accuracy | | |
|---|---|---|---|---|---|---|---|---|---|---|
| 100 | 105 | 65.45 | | | - | | | 88.90 | | |
| | 89 | 55.87 | | | - | | | 88.64 | | |
| | 79 | 49.74 | | | - | | | 88.10 | | |
| | | SSI=10 | SSI=20 | SSI=50 | SSI=10 | SSI=20 | SSI=50 | SSI=10 | SSI=20 | SSI=50 |
| 80 | 105 | 124.89 | 119.48 | 115.79 | 11.43 | 6.02 | 2.33 | 89.55 | 89.78 | 89.55 |
| | 89 | 105.98 | 101.65 | 97.99 | 9.15 | 4.82 | 1.16 | 89.40 | 89.69 | 89.52 |
| | 79 | 93.61 | 89.21 | 86.77 | 8.00 | 3.61 | 1.16 | 89.01 | 89.31 | 89.40 |
| 50 | 105 | 82.37 | 77.27 | 73.27 | 11.64 | 6.54 | 2.54 | 86.81 | 86.26 | 85.09 |
| | 89 | 69.65 | 65.57 | 61.61 | 9.31 | 5.23 | 1.27 | 86.65 | 86.19 | 85.09 |
| | 79 | 61.47 | 57.25 | 54.59 | 8.15 | 3.92 | 1.27 | 86.08 | 85.74 | 84.86 |
| 10 | 105 | 30.69 | 23.33 | 18.57 | 15.95 | 8.59 | 3.83 | 63.91 | 64.35 | 60.61 |
| | 89 | 25.39 | 19.50 | 14.54 | 12.76 | 6.87 | 1.91 | 63.59 | 61.49 | 60.61 |
| | 79 | 22.39 | 16.38 | 13.14 | 5.15 | 3.77 | 1.91 | 61.99 | 61.49 | 60.61 |
| 1 | 105 | 11.31 | 6.65 | 3.62 | 9.84 | 5.17 | 2.15 | 23.74 | 24.92 | 27.15 |
| | 89 | 9.13 | 5.40 | 2.33 | 7.87 | 4.14 | 1.07 | 23.74 | 24.92 | 27.08 |
| | 79 | 8.01 | 4.23 | 2.19 | 6.89 | 3.10 | 1.07 | 22.83 | 24.92 | 26.76 |

Table A8: Performance scores for ResNet50V2 with Random Initialization on CIFAR10 and its Effective data per epoch (EDPE), method=GradMatch

| EDPE (%) | Epoch | Total Time (Mins) | | | Total Selection Time (Mins) | | | Accuracy | | |
|---|---|---|---|---|---|---|---|---|---|---|
| 100 | 105 | | 311.64 | | | - | | | 89.44 | |
| | 89 | | 266.52 | | | - | | | 88.94 | |
| | 79 | | 236.23 | | | - | | | 88.94 | |
| | | SSI=10 | SSI=20 | SSI=50 | SSI=10 | SSI=20 | SSI=50 | SSI=10 | SSI=20 | SSI=50 |
| 80 | 105 | 575.97 | 474.68 | 421.88 | 186.06 | 84.77 | 31.97 | 89.46 | 89.28 | 89.23 |
| | 89 | 480.69 | 399.65 | 347.82 | 148.85 | 67.81 | 15.98 | 89.46 | 89.28 | 89.23 |
| | 79 | 425.21 | 345.83 | 310.95 | 130.24 | 50.86 | 15.98 | 89.41 | 89.28 | 89.12 |
| 50 | 105 | 429.41 | 328.12 | 275.32 | 186.06 | 84.77 | 31.97 | 89.97 | 89.59 | 89.18 |
| | 89 | 356.91 | 275.87 | 224.04 | 148.85 | 67.81 | 15.98 | 89.90 | 89.59 | 89.16 |
| | 79 | 314.60 | 235.21 | 200.34 | 130.24 | 50.86 | 15.98 | 89.90 | 89.42 | 89.04 |
| 30 | 105 | 332.62 | 231.33 | 178.53 | 186.06 | 84.77 | 31.97 | 88.48 | 89.05 | 89.01 |
| | 89 | 272.63 | 191.59 | 139.76 | 148.85 | 67.81 | 15.98 | 88.40 | 88.81 | 88.83 |
| | 79 | 240.85 | 161.47 | 126.60 | 130.24 | 50.86 | 15.98 | 88.33 | 88.77 | 88.70 |
| 10 | 105 | 235.84 | 134.54 | 81.75 | 186.06 | 84.77 | 31.97 | 84.97 | 84.46 | 86.00 |
| | 89 | 190.99 | 109.95 | 58.12 | 148.85 | 67.81 | 15.98 | 84.62 | 84.46 | 86.00 |
| | 79 | 167.11 | 87.73 | 52.85 | 130.24 | 50.86 | 15.98 | 83.99 | 83.95 | 86.00 |
| 1 | 105 | 191.59 | 90.30 | 37.50 | 186.06 | 84.77 | 31.97 | 48.30 | 39.39 | 33.73 |
| | 89 | 153.59 | 72.55 | 20.72 | 148.85 | 67.81 | 15.98 | 48.04 | 38.83 | 33.70 |
| | 79 | 134.45 | 55.07 | 20.20 | 130.24 | 50.86 | 15.98 | 44.97 | 38.00 | 33.70 |

Table A9: Performance scores for ViT-B_16 with ImageNet-21k weights Initialization on Tiny Imagenet and its Effective data per epoch (EDPE), method=CRAIG

| EDPE (%) | Epoch | Total Time (Mins) | | | Total Selection Time (Mins) | | | Accuracy | | |
|---|---|---|---|---|---|---|---|---|---|---|
| 100 | 105 | | 311.64 | | | - | | | 89.44 | |
| | 89 | | 266.52 | | | - | | | 88.94 | |
| | 79 | | 236.23 | | | - | | | 88.94 | |
| | | SSI=10 | SSI=20 | SSI=50 | SSI=10 | SSI=20 | SSI=50 | SSI=10 | SSI=20 | SSI=50 |
| 80 | 105 | 518.21 | 490.10 | 452.71 | 120.78 | 61.71 | 24.32 | 89.50 | 90.19 | 89.96 |
| | 89 | 434.47 | 415.78 | 378.57 | 96.62 | 49.37 | 12.16 | 89.50 | 90.19 | 89.96 |
| | 79 | 384.77 | 361.86 | 336.99 | 84.55 | 37.03 | 12.16 | 89.50 | 90.19 | 89.96 |
| 50 | 105 | 371.64 | 331.07 | 292.52 | 128.95 | 63.67 | 25.12 | 89.82 | 90.02 | 89.65 |
| | 89 | 310.69 | 279.62 | 241.24 | 103.16 | 50.93 | 12.56 | 89.82 | 90.02 | 89.65 |
| | 79 | 274.16 | 240.89 | 215.25 | 90.26 | 38.20 | 12.56 | 89.68 | 90.02 | 89.65 |
| 30 | 105 | 274.86 | 220.17 | 185.96 | 120.29 | 59.18 | 24.97 | 89.89 | 89.72 | 89.22 |
| | 89 | 226.41 | 185.07 | 150.21 | 96.23 | 47.34 | 12.48 | 89.89 | 89.72 | 89.22 |
| | 79 | 200.42 | 157.64 | 134.62 | 84.20 | 35.51 | 12.48 | 89.76 | 89.72 | 89.22 |
| 10 | 105 | 178.07 | 113.64 | 78.59 | 119.51 | 59.07 | 24.02 | 89.07 | 87.98 | 88.06 |
| | 89 | 144.77 | 94.03 | 58.78 | 95.61 | 47.26 | 12.01 | 88.88 | 87.98 | 87.51 |
| | 79 | 126.67 | 77.02 | 53.58 | 83.66 | 35.44 | 12.01 | 88.77 | 87.98 | 87.51 |
| 1 | 105 | 133.82 | 65.22 | 29.64 | 128.29 | 59.76 | 24.19 | 59.51 | 54.25 | 49.46 |
| | 89 | 107.37 | 52.49 | 16.77 | 102.63 | 47.81 | 12.09 | 59.09 | 50.07 | 41.88 |
| | 79 | 94.02 | 40.01 | 16.25 | 89.80 | 35.86 | 12.09 | 58.49 | 43.18 | 38.26 |

Table A10: Performance scores for ViT-B_16 with ImageNet-21k weights on Tiny Imagenet and its Effective data per epoch (EDPE), method=GradMatch

| EDPE (%) | Epoch | Total Time (Mins) | | | Total Selection Time (Mins) | | | Accuracy | | |
|---|---|---|---|---|---|---|---|---|---|---|
| 100 | 105 | | 311.64 | | | - | | | 89.44 | |
| | 89 | | 266.52 | | | - | | | 88.94 | |
| | 79 | | 236.23 | | | - | | | 88.94 | |
| | | *SSI=10* | *SSI=20* | *SSI=50* | *SSI=10* | *SSI=20* | *SSI=50* | *SSI=10* | *SSI=20* | *SSI=50* |
| 80 | 105 | 472.06 | 490.10 | 452.71 | 82.14 | 39.50 | 15.48 | 89.60 | 90.19 | 89.96 |
| | 89 | 397.56 | 415.78 | 378.57 | 65.71 | 31.60 | 7.74 | 89.56 | 90.19 | 89.96 |
| | 79 | 352.47 | 361.86 | 336.99 | 57.50 | 23.70 | 7.74 | 89.56 | 90.19 | 89.96 |
| 50 | 105 | 325.49 | 331.07 | 292.52 | 82.14 | 39.50 | 15.48 | 89.69 | 90.02 | 89.65 |
| | 89 | 273.77 | 279.62 | 241.24 | 65.71 | 31.60 | 7.74 | 89.56 | 90.02 | 89.65 |
| | 79 | 241.86 | 240.89 | 215.25 | 57.50 | 23.70 | 7.74 | 89.49 | 90.02 | 89.65 |
| 30 | 105 | 228.71 | 220.17 | 185.96 | 82.14 | 39.50 | 15.48 | 89.82 | 89.72 | 89.22 |
| | 89 | 189.50 | 185.07 | 150.21 | 65.71 | 31.60 | 7.74 | 89.77 | 89.72 | 89.22 |
| | 79 | 168.11 | 157.64 | 134.62 | 57.50 | 23.70 | 7.74 | 89.73 | 89.72 | 89.22 |
| 10 | 105 | 131.92 | 95.83 | 71.01 | 82.14 | 39.50 | 15.48 | 88.89 | 88.20 | 87.75 |
| | 89 | 107.85 | 79.78 | 54.99 | 65.71 | 31.60 | 7.74 | 88.81 | 88.20 | 87.38 |
| | 79 | 94.37 | 66.33 | 49.79 | 57.50 | 23.70 | 7.74 | 88.63 | 88.20 | 87.38 |
| 1 | 105 | 87.67 | 45.03 | 21.01 | 82.14 | 39.50 | 15.48 | 45.59 | - | 39.03 |
| | 89 | 70.45 | 36.34 | 12.48 | 65.71 | 31.60 | 7.74 | 45.43 | - | 39.03 |
| | 79 | 61.71 | 27.91 | 11.95 | 57.50 | 23.70 | 7.74 | 44.75 | - | 39.03 |

Table A11: Performance scores for ViT-B_16 with ImageNet-21k weights on Tiny Imagenet and its Effective data per epoch (EDPE), method=GLISTER

| EDPE (%) | Epoch | Total Time (Mins) | | | Total Selection Time (Mins) | | | Accuracy | | |
|---|---|---|---|---|---|---|---|---|---|---|
| 100 | 105 | | 311.64 | | | - | | | 89.44 | |
| | 89 | | 266.52 | | | - | | | 88.94 | |
| | 79 | | 236.23 | | | - | | | 88.94 | |
| | | *SSI=10* | *SSI=20* | *SSI=50* | *SSI=10* | *SSI=20* | *SSI=50* | *SSI=10* | *SSI=20* | *SSI=50* |
| 80 | 105 | 389.91 | 389.91 | 389.91 | 0.00 | 0.00 | 0.00 | 89.33 | 90.04 | 89.93 |
| | 89 | 331.84 | 331.84 | 331.84 | 0.00 | 0.00 | 0.00 | 89.33 | 90.04 | 89.93 |
| | 79 | 294.97 | 294.97 | 294.97 | 0.00 | 0.00 | 0.00 | 89.33 | 90.04 | 89.93 |
| 50 | 105 | 243.35 | 243.35 | 243.35 | 0.00 | 0.00 | 0.00 | 88.67 | 89.81 | 89.60 |
| | 89 | 208.05 | 208.05 | 208.05 | 0.00 | 0.00 | 0.00 | 88.67 | 89.81 | 89.60 |
| | 79 | 184.35 | 184.35 | 184.35 | 0.00 | 0.00 | 0.00 | 88.67 | 89.81 | 89.60 |
| 30 | 105 | 146.56 | 146.56 | 146.56 | 0.00 | 0.00 | 0.00 | 88.44 | 89.24 | 89.39 |
| | 89 | 123.78 | 123.78 | 123.78 | 0.00 | 0.00 | 0.00 | 88.44 | 89.24 | 89.39 |
| | 79 | 110.61 | 110.61 | 110.61 | 0.00 | 0.00 | 0.00 | 88.44 | 89.24 | 89.39 |
| 10 | 105 | 49.77 | 49.77 | 49.77 | 0.00 | 0.00 | 0.00 | 86.69 | 87.62 | 87.69 |
| | 89 | 42.13 | 42.13 | 42.13 | 0.00 | 0.00 | 0.00 | 86.69 | 87.62 | 87.69 |
| | 79 | 36.87 | 36.87 | 36.87 | 0.00 | 0.00 | 0.00 | 86.69 | 87.62 | 87.69 |
| 1 | 105 | 5.53 | 5.53 | 5.53 | 0.00 | 0.00 | 0.00 | 40.24 | 41.72 | 41.87 |
| | 89 | 4.74 | 4.74 | 4.74 | 0.00 | 0.00 | 0.00 | 39.96 | 40.57 | 39.95 |
| | 79 | 4.21 | 4.21 | 4.21 | 0.00 | 0.00 | 0.00 | 39.41 | 40.21 | 39.84 |

Table A12: Performance scores for ViT-B_16 with ImageNet-21k weights on Tiny Imagenet and its Effective data per epoch (EDPE), method=Random

| EDPE (%) | Epoch | Total Time (Mins) | | | Total Selection Time (Mins) | | | Accuracy | | |
|---|---|---|---|---|---|---|---|---|---|---|
| | 105 | 111.61 | | | - | | | 79.02 | | |
| 100 | 89 | 95.45 | | | - | | | 77.94 | | |
| | 79 | 84.60 | | | - | | | 77.54 | | |
| | | *SSI=10* | *SSI=20* | *SSI=50* | *SSI=10* | *SSI=20* | *SSI=50* | *SSI=10* | *SSI=20* | *SSI=50* |
| | 105 | 389.09 | 326.49 | 279.40 | 136.72 | 74.11 | 27.02 | 79.76 | 79.70 | 79.16 |
| 80 | 89 | 324.16 | 274.08 | 228.30 | 109.37 | 59.29 | 13.51 | 79.52 | 79.70 | 79.16 |
| | 79 | 286.62 | 235.39 | 204.43 | 95.70 | 44.47 | 13.51 | 79.24 | 79.29 | 79.16 |
| | 105 | 294.23 | 231.62 | 184.53 | 136.72 | 74.11 | 27.02 | 80.87 | 80.23 | 78.89 |
| 50 | 89 | 244.04 | 193.96 | 148.18 | 109.37 | 59.29 | 13.51 | 80.71 | 79.94 | 78.77 |
| | 79 | 215.03 | 163.79 | 132.84 | 95.70 | 44.47 | 13.51 | 80.41 | 79.66 | 78.64 |
| | 105 | 231.58 | 168.98 | 121.89 | 136.72 | 74.11 | 27.02 | 78.94 | 79.19 | 77.71 |
| 30 | 89 | 189.49 | 139.41 | 93.63 | 109.37 | 59.29 | 13.51 | 78.89 | 78.81 | 77.68 |
| | 79 | 167.30 | 116.06 | 85.11 | 95.70 | 44.47 | 13.51 | 78.55 | 78.18 | 77.28 |
| | 105 | 168.94 | 106.33 | 59.24 | 136.72 | 74.11 | 27.02 | 72.21 | 73.59 | 73.18 |
| 10 | 89 | 136.65 | 86.56 | 40.78 | 109.37 | 59.29 | 13.51 | 71.71 | 73.59 | 73.12 |
| | 79 | 119.57 | 68.33 | 37.37 | 95.70 | 44.47 | 13.51 | 71.71 | 73.59 | 72.89 |
| | 105 | 140.30 | 77.69 | 30.60 | 136.72 | 74.11 | 27.02 | 60.83 | 55.63 | 49.00 |
| 1 | 89 | 112.44 | 62.36 | 16.58 | 109.37 | 59.29 | 13.51 | 59.86 | 54.56 | 49.00 |
| | 79 | 98.43 | 47.19 | 16.24 | 95.70 | 44.47 | 13.51 | 58.10 | 50.75 | 49.00 |

Table A13: Performance scores for ResNet50V2 with ImageNet-21k Initialization on Tiny-ImageNet and its Effective data per epoch (EDPE), method=CRAIG

| EDPE (%) | Epoch | Total Time (Mins) | | | Total Selection Time (Mins) | | | Accuracy | | |
|---|---|---|---|---|---|---|---|---|---|---|
| | 105 | 111.61 | | | - | | | 79.02 | | |
| 100 | 89 | 95.45 | | | - | | | 77.94 | | |
| | 79 | 84.60 | | | - | | | 77.54 | | |
| | | *SSI=10* | *SSI=20* | *SSI=50* | *SSI=10* | *SSI=20* | *SSI=50* | *SSI=10* | *SSI=20* | *SSI=50* |
| | 105 | 321.22 | 283.43 | 258.94 | 77.97 | 40.18 | 15.70 | 79.40 | 79.42 | 79.16 |
| 80 | 89 | 269.39 | 239.16 | 214.86 | 62.38 | 32.15 | 7.85 | 79.37 | 79.38 | 79.16 |
| | 79 | 238.59 | 208.12 | 191.86 | 0.00 | 0.00 | 0.00 | 79.37 | 79.16 | 79.16 |
| | 105 | 229.79 | 192.00 | 167.51 | 77.97 | 40.18 | 15.70 | 80.41 | 80.04 | 78.61 |
| 50 | 89 | 192.17 | 161.94 | 137.64 | 62.38 | 32.15 | 7.85 | 80.28 | 79.84 | 78.61 |
| | 79 | 169.59 | 139.12 | 122.86 | 54.58 | 24.11 | 7.85 | 79.64 | 79.01 | 78.61 |
| | 105 | 169.40 | 131.62 | 107.13 | 77.97 | 40.18 | 15.70 | 81.09 | 80.29 | 77.74 |
| 30 | 89 | 139.60 | 109.37 | 85.07 | 62.38 | 32.15 | 7.85 | 80.96 | 80.12 | 77.65 |
| | 79 | 123.58 | 93.11 | 76.85 | 54.58 | 24.11 | 7.85 | 80.46 | 79.05 | 77.54 |
| | 105 | 109.02 | 71.24 | 46.75 | 77.97 | 40.18 | 15.70 | 80.24 | 78.43 | 75.36 |
| 10 | 89 | 88.66 | 58.43 | 34.13 | 62.38 | 32.15 | 7.85 | 79.98 | 78.20 | 75.36 |
| | 79 | 77.58 | 47.11 | 30.85 | 54.58 | 24.11 | 7.85 | 79.11 | 76.95 | 74.98 |
| | 105 | 81.42 | 43.64 | 19.15 | 77.97 | 40.18 | 15.70 | 72.49 | 68.88 | 60.88 |
| 1 | 89 | 65.33 | 35.10 | 10.80 | 62.38 | 32.15 | 7.85 | 71.71 | 68.61 | 60.80 |
| | 79 | 57.21 | 26.74 | 10.47 | 54.58 | 24.11 | 7.85 | 70.50 | 66.61 | 60.06 |

Table A14: Performance scores for ResNet50V2 with ImageNet-21k weights on Tiny Imagenet and its Effective data per epoch (EDPE), method=GradMatch

| EDPE (%) | Epoch | Total Time (Mins) | | | Total Selection Time (Mins) | | | Accuracy | | |
|---|---|---|---|---|---|---|---|---|---|---|
| 100 | 105 | | 111.61 | | | - | | | 79.02 | |
| | 89 | | 95.45 | | | - | | | 77.94 | |
| | 79 | | 84.60 | | | - | | | 77.54 | |
| | | *SSI=10* | *SSI=20* | *SSI=50* | *SSI=10* | *SSI=20* | *SSI=50* | *SSI=10* | *SSI=20* | *SSI=50* |
| 80 | 105 | 309.69 | 278.64 | 259.50 | 62.48 | 31.44 | 12.29 | 79.09 | 79.09 | 79.09 |
| | 89 | 261.88 | 237.04 | 218.03 | 49.99 | 25.15 | 6.14 | 79.09 | 79.09 | 79.09 |
| | 79 | 232.08 | 207.21 | 194.49 | 43.74 | 18.86 | 6.14 | 79.09 | 79.09 | 79.09 |
| 50 | 105 | 217.87 | 186.82 | 167.68 | 62.48 | 31.44 | 12.29 | 78.11 | 78.11 | 78.11 |
| | 89 | 181.16 | 156.32 | 137.31 | 49.99 | 25.15 | 6.14 | 78.11 | 78.11 | 78.11 |
| | 79 | 161.45 | 136.58 | 123.86 | 43.74 | 18.86 | 6.14 | 78.11 | 78.11 | 78.11 |
| 30 | 105 | 154.30 | 123.26 | 104.11 | 62.48 | 31.44 | 12.29 | 76.73 | 76.11 | 75.55 |
| | 89 | 127.34 | 102.51 | 83.50 | 49.99 | 25.15 | 6.14 | 76.42 | 75.81 | 75.55 |
| | 79 | 111.00 | 86.13 | 73.41 | 43.74 | 18.86 | 6.14 | 76.42 | 75.74 | 75.55 |
| 10 | 105 | 94.27 | 63.225 | 44.07 | 62.48 | 31.44 | 12.29 | 75.08 | 72.32 | 71.22 |
| | 89 | 76.89 | 52.06 | 33.05 | 49.99 | 25.15 | 6.14 | 74.33 | 72.32 | 71.22 |
| | 79 | 67.28 | 42.40 | 29.69 | 43.74 | 18.86 | 6.14 | 74.33 | 72.32 | 71.22 |
| 1 | 105 | 66.01 | 34.97 | 15.82 | 62.48 | 31.44 | 12.29 | 63.99 | 64.10 | 57.57 |
| | 89 | 53.01 | 28.18 | 9.17 | 49.99 | 25.15 | 6.14 | 63.71 | 62.11 | 57.57 |
| | 79 | 46.43 | 21.55 | 8.83 | 43.74 | 18.86 | 6.14 | 63.71 | 61.24 | 57.57 |

Table A15: Performance scores for ResNet50V2 with ImageNet-21k weights on Tiny Imagenet and its Effective data per epoch (EDPE), method=GLISTER

| EDPE (%) | Epoch | Total Time (Mins) | | | Total Selection Time (Mins) | | | Accuracy | | |
|---|---|---|---|---|---|---|---|---|---|---|
| 100 | 105 | | 111.61 | | | - | | | 79.02 | |
| | 89 | | 95.45 | | | - | | | 77.94 | |
| | 79 | | 84.60 | | | - | | | 77.54 | |
| | | *SSI=10* | *SSI=20* | *SSI=50* | *SSI=10* | *SSI=20* | *SSI=50* | *SSI=10* | *SSI=20* | *SSI=50* |
| 80 | 105 | 250.73 | 250.73 | 250.73 | 0.00 | 0.00 | 0.00 | 78.81 | 78.81 | 78.81 |
| | 89 | 211.88 | 211.88 | 211.88 | 0.00 | 0.00 | 0.00 | 78.81 | 78.81 | 78.81 |
| | 79 | 188.34 | 188.34 | 188.34 | 0.00 | 0.00 | 0.00 | 78.81 | 78.81 | 78.81 |
| 50 | 105 | 155.38 | 155.38 | 155.38 | 0.00 | 0.00 | 0.00 | 78.17 | 78.17 | 78.17 |
| | 89 | 131.17 | 131.17 | 131.17 | 0.00 | 0.00 | 0.00 | 78.17 | 78.17 | 78.17 |
| | 79 | 117.71 | 117.71 | 117.71 | 0.00 | 0.00 | 0.00 | 78.17 | 78.17 | 78.17 |
| 30 | 105 | 95.35 | 95.35 | 95.35 | 0.00 | 0.00 | 0.00 | 75.59 | 75.59 | 75.59 |
| | 89 | 80.72 | 80.72 | 80.72 | 0.00 | 0.00 | 0.00 | 75.59 | 75.59 | 75.59 |
| | 79 | 70.62 | 70.62 | 70.62 | 0.00 | 0.00 | 0.00 | 75.59 | 75.59 | 75.59 |
| 10 | 105 | 31.78 | 31.78 | 31.78 | 0.00 | 0.00 | 0.00 | 72.07 | 71.52 | 71.51 |
| | 89 | 26.90 | 26.90 | 26.90 | 0.00 | 0.00 | 0.00 | 72.07 | 71.52 | 71.51 |
| | 79 | 23.54 | 23.54 | 23.54 | 0.00 | 0.00 | 0.00 | 72.07 | 71.52 | 71.51 |
| 1 | 105 | 3.53 | 3.53 | 3.53 | 0.00 | 0.00 | 0.00 | 52.21 | 57.02 | 56.76 |
| | 89 | 3.02 | 3.02 | 3.02 | 0.00 | 0.00 | 0.00 | 52.21 | 57.02 | 56.76 |
| | 79 | 2.69 | 2.69 | 2.69 | 0.00 | 0.00 | 0.00 | 52.21 | 57.02 | 56.76 |

Table A16: Performance scores for ResNet50V2 with ImageNet-21k weights on Tiny Imagenet and its Effective data per epoch (EDPE), method=Random

| EDPE (%) | Epoch | Total Time (Mins) | | | Total Selection Time (Mins) | | | Accuracy | | |
|---|---|---|---|---|---|---|---|---|---|---|
| | 105 | | 113.96 | | | - | | | 74.83 | |
| 100 | 89 | | 97.68 | | | - | | | 74.79 | |
| | 79 | | 86.82 | | | - | | | 74.56 | |
| | | *SSI=10* | *SSI=20* | *SSI=50* | *SSI=10* | *SSI=20* | *SSI=50* | *SSI=10* | *SSI=20* | *SSI=50* |
| | 105 | 398.84 | 307.91 | 265.86 | 161.91 | 70.98 | 28.93 | 75.76 | 75.24 | 74.88 |
| 80 | 89 | 331.17 | 258.42 | 216.10 | 129.53 | 56.78 | 14.46 | 75.51 | 75.17 | 74.88 |
| | 79 | 292.57 | 221.82 | 193.70 | 113.33 | 42.59 | 14.46 | 75.24 | 75.05 | 74.47 |
| | 105 | 309.78 | 218.85 | 176.80 | 161.91 | 70.98 | 28.93 | 75.35 | 74.73 | 73.99 |
| 50 | 89 | 255.95 | 183.21 | 140.89 | 129.53 | 56.78 | 14.46 | 75.11 | 74.36 | 73.93 |
| | 79 | 225.36 | 154.61 | 126.49 | 113.33 | 42.59 | 14.46 | 75.01 | 74.16 | 73.75 |
| | 105 | 250.97 | 160.04 | 117.99 | 161.91 | 70.98 | 28.93 | 72.10 | 72.71 | 71.72 |
| 30 | 89 | 204.74 | 132.00 | 89.68 | 129.53 | 56.78 | 14.46 | 72.04 | 72.29 | 71.55 |
| | 79 | 180.55 | 109.80 | 81.68 | 113.33 | 42.59 | 14.46 | 71.46 | 71.90 | 71.02 |
| | 105 | 192.15 | 101.23 | 59.18 | 161.91 | 70.98 | 28.93 | 59.15 | 60.22 | 62.17 |
| 10 | 89 | 155.13 | 82.39 | 40.07 | 129.53 | 56.78 | 14.46 | 57.16 | 60.22 | 62.03 |
| | 79 | 135.74 | 64.99 | 36.87 | 113.33 | 42.59 | 14.46 | 57.16 | 60.22 | 60.43 |
| | 105 | 165.27 | 74.34 | 32.29 | 161.91 | 70.98 | 28.93 | 17.15 | 18.36 | 19.08 |
| 1 | 89 | 132.41 | 59.66 | 17.34 | 129.53 | 56.78 | 14.46 | 16.83 | 17.88 | 18.98 |
| | 79 | 115.89 | 45.15 | 17.02 | 113.33 | 42.59 | 14.46 | 14.95 | 17.38 | 18.98 |

Table A17: Performance scores for MobileNetV3 with ImageNet-1k weights Initialization on Tiny Imagenet and its Effective data per epoch (EDPE), method=CRAIG

| EDPE (%) | Epoch | Total Time (Mins) | | | Total Selection Time (Mins) | | | Accuracy | | |
|---|---|---|---|---|---|---|---|---|---|---|
| | 105 | | 113.96 | | | - | | | 74.83 | |
| 100 | 89 | | 97.68 | | | - | | | 74.79 | |
| | 79 | | 86.82 | | | - | | | 74.56 | |
| | | *SSI=10* | *SSI=20* | *SSI=50* | *SSI=10* | *SSI=20* | *SSI=50* | *SSI=10* | *SSI=20* | *SSI=50* |
| | 105 | 314.47 | 275.92 | 252.20 | 77.55 | 38.99 | 15.27 | 75.39 | 75.58 | 74.92 |
| 80 | 89 | 263.68 | 232.83 | 209.27 | 62.04 | 31.19 | 7.63 | 75.31 | 75.29 | 74.91 |
| | 79 | 233.52 | 202.63 | 186.87 | 54.28 | 23.39 | 7.63 | 75.25 | 74.87 | 74.82 |
| | 105 | 225.42 | 186.86 | 163.14 | 77.55 | 38.99 | 15.27 | 75.75 | 75.23 | 73.66 |
| 50 | 89 | 188.46 | 157.62 | 134.06 | 62.04 | 31.19 | 7.63 | 75.64 | 75.15 | 73.66 |
| | 79 | 166.30 | 135.41 | 119.66 | 54.28 | 23.39 | 7.63 | 75.39 | 74.87 | 73.66 |
| | 105 | 166.60 | 128.05 | 104.33 | 77.55 | 38.99 | 15.27 | 75.47 | 74.60 | 71.82 |
| 30 | 89 | 137.25 | 106.40 | 82.85 | 62.04 | 31.19 | 7.63 | 75.40 | 74.44 | 71.71 |
| | 79 | 121.49 | 90.60 | 74.85 | 54.28 | 23.39 | 7.63 | 74.98 | 73.63 | 71.52 |
| | 105 | 107.79 | 69.23 | 45.52 | 77.55 | 38.99 | 15.27 | 72.42 | 70.02 | 66.05 |
| 10 | 89 | 87.64 | 56.79 | 33.24 | 62.04 | 31.19 | 7.63 | 71.61 | 69.70 | 65.85 |
| | 79 | 76.68 | 45.79 | 30.04 | 54.28 | 23.39 | 7.63 | 70.60 | 68.96 | 65.54 |
| | 105 | 80.91 | 42.35 | 18.63 | 77.55 | 38.99 | 15.27 | 29.35 | 29.78 | 27.18 |
| 1 | 89 | 64.92 | 34.07 | 10.51 | 62.04 | 31.19 | 7.63 | 27.61 | 29.47 | 26.21 |
| | 79 | 56.84 | 25.95 | 10.19 | 54.28 | 23.39 | 7.63 | 25.58 | 27.96 | 25.01 |

Table A18: Performance scores for MobileNetV3 with ImageNet-1k weights Initialization on Tiny Imagenet and its Effective data per epoch (EDPE), method=GradMatch

| EDPE (%) | Epoch | Total Time (Mins) | | | Total Selection Time (Mins) | | | Accuracy | | |
|---|---|---|---|---|---|---|---|---|---|---|
| | 105 | | 113.96 | | | - | | | 74.83 | |
| 100 | 89 | | 97.68 | | | - | | | 74.79 | |
| | 79 | | 86.82 | | | - | | | 74.56 | |
| | | *SSI=10* | *SSI=20* | *SSI=50* | *SSI=10* | *SSI=20* | *SSI=50* | *SSI=10* | *SSI=20* | *SSI=50* |
| | 105 | 275.87 | 256.01 | 244.63 | 38.94 | 19.08 | 7.70 | 75.87 | 75.63 | 75.22 |
| 80 | 89 | 232.79 | 216.90 | 205.49 | 31.15 | 15.26 | 3.85 | 75.71 | 75.48 | 75.04 |
| | 79 | 206.49 | 190.68 | 183.08 | 27.25 | 11.44 | 3.85 | 75.45 | 75.24 | 74.76 |
| | 105 | 186.81 | 166.95 | 155.57 | 38.94 | 19.08 | 7.70 | 75.45 | 75.36 | 73.88 |
| 50 | 89 | 157.57 | 130.27 | 130.27 | 31.15 | 15.26 | 3.85 | 74.95 | 74.62 | 73.31 |
| | 79 | 139.28 | 115.87 | 126.49 | 27.25 | 11.44 | 3.85 | 74.60 | 74.02 | 73.31 |
| | 105 | 127.99 | 108.14 | 96.76 | 38.94 | 19.08 | 7.70 | 75.15 | 73.94 | 72.12 |
| 30 | 89 | 106.36 | 90.48 | 79.06 | 31.15 | 15.26 | 3.85 | 73.49 | 72.62 | 71.42 |
| | 79 | 94.47 | 78.66 | 71.06 | 27.25 | 11.44 | 3.85 | 72.91 | 71.40 | 71.42 |
| | 105 | 69.18 | 49.32 | 37.94 | 38.94 | 19.08 | 7.70 | 68.37 | 65.32 | 63.78 |
| 10 | 89 | 56.75 | 40.87 | 29.45 | 31.15 | 15.26 | 3.85 | 67.78 | 64.93 | 63.68 |
| | 79 | 49.66 | 33.85 | 26.25 | 27.25 | 11.44 | 3.85 | 66.93 | 64.93 | 63.19 |
| | 105 | 42.30 | 22.44 | 11.06 | 38.94 | 19.08 | 7.70 | 20.68 | 26.97 | 27.05 |
| 1 | 89 | 34.03 | 18.14 | 6.73 | 31.15 | 15.26 | 3.85 | 17.57 | 25.78 | 26.42 |
| | 79 | 29.81 | 14.00 | 6.41 | 27.25 | 11.44 | 3.85 | 14.95 | 23.64 | 25.71 |

Table A19: Performance scores for MobileNetV3 with ImageNet-1k weights Initialization on Tiny Imagenet and its Effective data per epoch (EDPE), method=GLISTER

| EDPE (%) | Epoch | Total Time (Mins) | | | Total Selection Time (Mins) | | | Accuracy | | |
|---|---|---|---|---|---|---|---|---|---|---|
| | 105 | | 113.96 | | | - | | | 74.83 | |
| 100 | 89 | | 97.68 | | | - | | | 74.79 | |
| | 79 | | 86.82 | | | - | | | 74.56 | |
| | | *SSI=10* | *SSI=20* | *SSI=50* | *SSI=10* | *SSI=20* | *SSI=50* | *SSI=10* | *SSI=20* | *SSI=50* |
| | 105 | 236.92 | 236.92 | 236.92 | 0.0 | 0.0 | 0.0 | 74.44 | 74.12 | 74.92 |
| 80 | 89 | 201.64 | 201.64 | 201.64 | 0.0 | 0.0 | 0.0 | 74.40 | 74.10 | 74.91 |
| | 79 | 179.23 | 179.23 | 179.23 | 0.0 | 0.0 | 0.0 | 74.12 | 74.00 | 74.82 |
| | 105 | 147.87 | 147.87 | 147.87 | 0.0 | 0.0 | 0.0 | 72.34 | 72.39 | 73.85 |
| 50 | 89 | 126.42 | 126.42 | 126.42 | 0.0 | 0.0 | 0.0 | 72.28 | 72.39 | 73.72 |
| | 79 | 112.02 | 112.02 | 112.02 | 0.0 | 0.0 | 0.0 | 72.05 | 72.39 | 73.69 |
| | 105 | 89.05 | 89.05 | 89.05 | 0.0 | 0.0 | 0.0 | 70.60 | 70.68 | 72.35 |
| 30 | 89 | 75.21 | 75.21 | 75.21 | 0.0 | 0.0 | 0.0 | 70.52 | 70.60 | 72.26 |
| | 79 | 67.21 | 67.21 | 67.21 | 0.0 | 0.0 | 0.0 | 69.91 | 70.33 | 72.22 |
| | 105 | 30.24 | 30.24 | 30.24 | 0.0 | 0.0 | 0.0 | 62.58 | 64.29 | 65.77 |
| 10 | 89 | 25.60 | 25.60 | 25.60 | 0.0 | 0.0 | 0.0 | 62.15 | 64.29 | 65.77 |
| | 79 | 22.40 | 22.40 | 22.40 | 0.0 | 0.0 | 0.0 | 61.90 | 63.87 | 65.77 |
| | 105 | 3.36 | 3.36 | 3.36 | 0.0 | 0.0 | 0.0 | 22.72 | 25.81 | 29.05 |
| 1 | 89 | 2.88 | 2.88 | 2.88 | 0.0 | 0.0 | 0.0 | 22.19 | 25.35 | 28.30 |
| | 79 | 2.56 | 2.56 | 2.56 | 0.0 | 0.0 | 0.0 | 21.40 | 24.47 | 27.17 |

Table A20: Performance scores for MobileNetV3 with ImageNet-1k weights Initialization on Tiny Imagenet and its Effective data per epoch (EDPE), method=Random

| EDPE (%) | Epoch | Total Time (Mins) | | | Total Selection Time (Mins) | | | Accuracy | | |
|---|---|---|---|---|---|---|---|---|---|---|
| 100 | 105 | | 311.64 | | | - | | | 89.44 | |
| | 89 | | 266.52 | | | - | | | 88.94 | |
| | 79 | | 236.23 | | | - | | | 88.94 | |
| | | SSI=10 | SSI=20 | SSI=50 | SSI=10 | SSI=20 | SSI=50 | SSI=10 | SSI=20 | SSI=50 |
| 80 | 105 | 1150.82 | 307.91 | 265.86 | 226.14 | 103.75 | 40.27 | 90.89 | - | - |
| | 89 | 967.87 | 258.42 | 216.10 | 180.91 | 83.00 | 20.13 | 90.78 | - | - |
| | 79 | 857.81 | 221.82 | 193.70 | 158.30 | 62.25 | 20.13 | 90.60 | - | - |
| 50 | 105 | 803.24 | 680.85 | 617.37 | 226.14 | 103.75 | 40.27 | 91.51 | 91.49 | 90.90 |
| | 89 | 674.32 | 576.41 | 513.54 | 180.91 | 83.00 | 20.13 | 91.51 | 91.41 | 90.78 |
| | 79 | 595.50 | 499.44 | 457.33 | 158.30 | 62.25 | 20.13 | 91.28 | 91.14 | 90.62 |
| 30 | 105 | 573.72 | 451.32 | 387.84 | 226.14 | 103.75 | 40.27 | 91.28 | 91.05 | 90.42 |
| | 89 | 474.46 | 376.55 | 313.68 | 180.91 | 83.00 | 20.13 | 91.15 | 90.70 | 90.33 |
| | 79 | 420.62 | 324.57 | 282.45 | 158.30 | 62.25 | 20.13 | 90.65 | 90.56 | 90.24 |
| 10 | 105 | 343.56 | 221.79 | 155.19 | 226.14 | 103.75 | 40.27 | 86.60 | 86.83 | 88.26 |
| | 89 | 280.85 | 182.93 | 113.82 | 180.91 | 83.00 | 20.13 | 86.40 | 86.83 | 87.60 |
| | 79 | 245.74 | 149.69 | 107.57 | 158.30 | 62.25 | 20.13 | 85.43 | 86.83 | 87.46 |
| 1 | 105 | 239.26 | 116.87 | 53.38 | 226.14 | 103.75 | 40.27 | 76.14 | 72.53 | 66.49 |
| | 89 | 192.16 | 94.24 | 31.37 | 180.91 | 83.00 | 20.13 | 75.39 | 72.00 | 66.15 |
| | 79 | 168.29 | 72.24 | 30.12 | 158.30 | 62.25 | 20.13 | 72.97 | 69.72 | 65.89 |

Table A21: Performance scores for Swin TransformerV2 with ImageNet-21K and ImageNet1k fine-tuned weights Initialization on Tiny Imagenet and its Effective data per epoch (EDPE), method=CRAIG

| EDPE (%) | Epoch | Total Time (Mins) | | | Total Selection Time (Mins) | | | Accuracy | | |
|---|---|---|---|---|---|---|---|---|---|---|
| 100 | 105 | | 311.64 | | | - | | | 89.44 | |
| | 89 | | 266.52 | | | - | | | 88.94 | |
| | 79 | | 236.23 | | | - | | | 88.94 | |
| | | SSI=10 | SSI=20 | SSI=50 | SSI=10 | SSI=20 | SSI=50 | SSI=10 | SSI=20 | SSI=50 |
| 80 | 105 | 398.84 | 307.91 | 265.86 | 125.24 | 62.45 | 25.20 | - | - | - |
| | 89 | 331.17 | 258.42 | 216.10 | 100.19 | 49.96 | 12.60 | - | - | - |
| | 79 | 292.57 | 221.82 | 193.70 | 87.67 | 37.47 | 20.13 | - | - | - |
| 50 | 105 | 702.34 | 639.55 | 602.30 | 125.24 | 62.45 | 25.20 | 91.44 | 91.43 | 90.76 |
| | 89 | 593.60 | 543.37 | 506.01 | 100.19 | 49.96 | 12.60 | 91.34 | 91.34 | 90.69 |
| | 79 | 524.87 | 474.66 | 449.80 | 87.67 | 37.47 | 20.13 | 91.24 | 91.21 | 90.69 |
| 30 | 105 | 472.82 | 410.02 | 372.78 | 125.24 | 62.45 | 25.20 | 91.90 | 91.48 | 90.66 |
| | 89 | 393.74 | 343.50 | 306.15 | 100.19 | 49.96 | 12.60 | 91.71 | 91.43 | 90.58 |
| | 79 | 349.99 | 299.79 | 274.92 | 87.67 | 37.47 | 20.13 | 91.46 | 91.21 | 90.56 |
| 10 | 105 | 243.29 | 180.49 | 143.25 | 125.24 | 62.45 | 25.20 | 91.48 | 90.93 | 89.56 |
| | 89 | 200.13 | 149.89 | 112.53 | 100.19 | 49.96 | 12.60 | 91.31 | 90.66 | 89.56 |
| | 79 | 175.11 | 124.91 | 100.04 | 87.67 | 37.47 | 20.13 | 91.17 | 90.62 | 89.51 |
| 1 | 105 | 138.36 | 75.56 | 38.32 | 125.24 | 62.45 | 25.20 | 85.37 | 83.72 | 77.42 |
| | 89 | 111.44 | 61.20 | 23.84 | 100.19 | 49.96 | 12.60 | 84.96 | 83.59 | 77.42 |
| | 79 | 97.66 | 47.46 | 22.59 | 87.67 | 37.47 | 20.13 | 84.00 | 81.87 | 76.37 |

Table A22: Performance scores for Swin TransformerV2 with ImageNet-21K and ImageNet1k fine-tuned weights Initialization on Tiny Imagenet and its Effective data per epoch (EDPE), method=GradMatch

| EDPE (%) | Epoch | Total Time (Mins) | | | Total Selection Time (Mins) | | | Accuracy | | |
|---|---|---|---|---|---|---|---|---|---|---|
| 100 | 105 | | 311.64 | | | - | | | 89.44 | |
| | 89 | | 266.52 | | | - | | | 88.94 | |
| | 79 | | 236.23 | | | - | | | 88.94 | |
| | | *SSI=10* | *SSI=20* | *SSI=50* | *SSI=10* | *SSI=20* | *SSI=50* | *SSI=10* | *SSI=20* | *SSI=50* |
| 80 | 105 | 1024.94 | 307.91 | 265.86 | 226.14 | 103.75 | 40.27 | 90.97 | - | - |
| | 89 | 867.17 | 258.42 | 216.10 | 180.91 | 83.00 | 20.13 | 90.95 | - | - |
| | 79 | 769.70 | 221.82 | 193.70 | 158.30 | 62.25 | 20.13 | 90.95 | - | - |
| 50 | 105 | 677.37 | 680.85 | 617.37 | 226.14 | 103.75 | 40.27 | 91.47 | 91.49 | 90.90 |
| | 89 | 573.62 | 576.41 | 513.54 | 180.91 | 83.00 | 20.13 | 91.47 | 91.41 | 90.78 |
| | 79 | 507.39 | 499.44 | 457.33 | 158.30 | 62.25 | 20.13 | 91.27 | 91.14 | 90.62 |
| 30 | 105 | 447.84 | 451.32 | 387.84 | 226.14 | 103.75 | 40.27 | 91.87 | 91.05 | 90.42 |
| | 89 | 373.76 | 376.55 | 313.68 | 180.91 | 83.00 | 20.13 | 91.69 | 90.70 | 90.33 |
| | 79 | 332.51 | 324.57 | 282.45 | 158.30 | 62.25 | 20.13 | 91.69 | 90.56 | 90.24 |
| 10 | 105 | 218.32 | 220.54 | 155.19 | 226.14 | 103.75 | 40.27 | 91.63 | 86.83 | 88.26 |
| | 89 | 180.15 | 182.93 | 113.82 | 180.91 | 83.00 | 20.13 | 91.05 | 86.83 | 87.60 |
| | 79 | 157.63 | 149.69 | 107.57 | 158.30 | 62.25 | 20.13 | 90.81 | 86.83 | 87.46 |
| 1 | 105 | 113.39 | 116.87 | 53.38 | 100.27 | 103.75 | 40.27 | 86.41 | 72.53 | 66.49 |
| | 89 | 91.46 | 94.24 | 31.37 | 80.22 | 83.00 | 20.13 | 85.82 | 72.00 | 66.15 |
| | 79 | 80.18 | 72.24 | 30.12 | 70.19 | 62.25 | 20.13 | 84.47 | 69.72 | 65.89 |

Table A23: Performance scores for Swin TransformerV2 with ImageNet-21K and ImageNet1k fine-tuned weights Initialization on Tiny Imagenet and its Effective data per epoch (EDPE), method=GLISTER

| EDPE (%) | Epoch | Total Time (Mins) | | | Total Selection Time (Mins) | | | Accuracy | | |
|---|---|---|---|---|---|---|---|---|---|---|
| 100 | 105 | | 311.64 | | | - | | | 89.44 | |
| | 89 | | 266.52 | | | - | | | 88.94 | |
| | 79 | | 236.23 | | | - | | | 88.94 | |
| | | *SSI=10* | *SSI=20* | *SSI=50* | *SSI=10* | *SSI=20* | *SSI=50* | *SSI=10* | *SSI=20* | *SSI=50* |
| 80 | 105 | 924.67 | 924.67 | 924.67 | 0.0 | 0.0 | 0.0 | - | - | - |
| | 89 | 786.95 | 786.95 | 786.95 | 0.0 | 0.0 | 0.0 | - | - | - |
| | 79 | 699.51 | 699.51 | 699.51 | 0.0 | 0.0 | 0.0 | - | - | - |
| 50 | 105 | 577.09 | 577.09 | 577.09 | 0.0 | 0.0 | 0.0 | 90.00 | 91.43 | 90.76 |
| | 89 | 493.40 | 493.40 | 493.40 | 0.0 | 0.0 | 0.0 | 90.00 | 91.34 | 90.69 |
| | 79 | 437.19 | 437.19 | 437.19 | 0.0 | 0.0 | 0.0 | 90.00 | 91.21 | 90.69 |
| 30 | 105 | 347.57 | 347.57 | 347.57 | 0.0 | 0.0 | 0.0 | 90.02 | 91.48 | 90.66 |
| | 89 | 293.54 | 343.50 | 306.15 | 0.0 | 0.0 | 0.0 | 90.02 | 91.43 | 90.58 |
| | 79 | 262.31 | 299.79 | 274.92 | 0.0 | 0.0 | 0.0 | 90.02 | 91.21 | 90.56 |
| 10 | 105 | 118.04 | 118.04 | 118.04 | 0.0 | 0.0 | 0.0 | 87.77 | 90.93 | 89.56 |
| | 89 | 99.93 | 149.89 | 112.53 | 0.0 | 0.0 | 0.0 | 87.77 | 90.66 | 89.56 |
| | 79 | 87.43 | 124.91 | 100.04 | 0.0 | 0.0 | 0.0 | 87.77 | 90.62 | 89.51 |
| 1 | 105 | 13.11 | 13.11 | 13.11 | 0.0 | 0.0 | 0.0 | 64.97 | 83.72 | 77.42 |
| | 89 | 11.24 | 11.24 | 11.24 | 0.0 | 0.0 | 0.0 | 64.91 | 83.59 | 77.42 |
| | 79 | 9.99 | 9.99 | 9.99 | 0.0 | 0.0 | 0.0 | 64.52 | 81.87 | 76.37 |

Table A24: Performance scores for Swin TransformerV2 with ImageNet-21K and ImageNet1k fine-tuned weights Initialization on Tiny Imagenet and its Effective data per epoch (EDPE), method=Random

are conducted five times and results reported in the table are average with their standard deviation. Tables [A25, A26] shows comparative study of these models, and we used Fig. 6 to show that when transformers are initialized with ImageNet21k they outperform CNNs across all the EDPE values and coreset methods. When the models are randomly initialized, it can be seen that transformers is suffering and CNNs are outperforming them.

### A.4.1  CRAIG, GradMatch, GLISTER and Random Results on UltraMNIST

To examine how well transformers can learn a robust representation in a dataset, we used UltraMNIST to compare ViT_B16 and ResNet50. Tables [A27, A28, A29, A30, A31, A32, A33, A34] delivers a comparative study for these models, and we used Fig. 8 to exhibit that CNN outperforms across all coreset sizes and that due to nonavailability of similar pretraining on such non-natural image dataset, transformers tend to suffer in learning desired semantic context as mentioned in the section 'Learning semantic coherency in non-natural images' in the main paper. For UltraMNIST, GradMatch seems to perform best when compared to other methods, and random selection performs worst. These comments are uniform with varying SSI values, as shown in Fig. 9 in the main document.

### A.4.2  CRAIG, GradMatch, GLISTER and Random Results on Medical Dataset

We further expanded our experiments vision to a medical dataset, to study the effect of pretraining when using ViT_B16 and ResNet50 with different coreset sizes. From the tables [A35, A36, A37, A38, A39, A40, A41, A42] and the generated Fig. 7 from these tables assisted us to show that without pretraining CNNs outperform on Transformers for different coreset values using random selection. It was also intriguing to see that random selection is better or almost at par with GradMatch, which is itself superior to other coreset methods.

| Model | Baseline | EDPE | Total Training Time | Accuracy (Pretrain) | Accuracy (Random) |
|---|---|---|---|---|---|
| ResNet50 | Full data | 100 | 5.74 | 91.04 ± 0.29 | 21.39 ± 0.98 |
| MobileNetV3 | Full data | 100 | 4.68 | 86.51 ± 0.31 | 19.94 ± 1.25 |
| **Model** | **Coreset Method** | **EDPE** | **Total Training Time** | **Accuracy (Pretrain)** | **Accuracy (Random)** |
| ResNet50 | CRAIG | 1 | 4.64 | 44.68 ± 4.17 | 3.40 ± 0.17 |
| ResNet50 | CRAIG | 10 | 4.68 | 81.76 ± 0.94 | 3.05 ± 0.45 |
| ResNet50 | CRAIG | 30 | 6.56 | 86.74 ± 0.55 | 5.08 ± 0.33 |
| ResNet50 | CRAIG | 50 | 9.45 | 88.96 ± 0.33 | 12.04 ± 1.02 |
| ResNet50 | CRAIG | 80 | 12.65 | 89.45 ± 0.64 | 19.88 ± 1.04 |
| ResNet50 | GradMatch | 1 | 2.37 | 44.68 ± 4.17 | 3.24 ± 0.24 |
| ResNet50 | GradMatch | 10 | 2.40 | 84.95 ± 1.34 | 2.96 ± 0.46 |
| ResNet50 | GradMatch | 30 | 4.28 | 87.92 ± 0.58 | 5.05 ± 0.64 |
| ResNet50 | GradMatch | 50 | 7.17 | 89.32 ± 0.27 | 11.91 ± 0.93 |
| ResNet50 | GradMatch | 80 | 10.38 | 89.76 ± 0.28 | 19.77 ± 0.79 |
| ResNet50 | GLISTER | 1 | 1.15 | 46.87 ± 2.77 | 3.57 ± 0.31 |
| ResNet50 | GLISTER | 10 | 1.18 | 84.68 ± 0.34 | 2.81 ± 0.09 |
| ResNet50 | GLISTER | 30 | 3.06 | 88.48 ± 0.48 | 5.15 ± 0.50 |
| ResNet50 | GLISTER | 50 | 5.95 | 89.28 ± 0.50 | 13.08 ± 1.21 |
| ResNet50 | GLISTER | 80 | 9.16 | 89.68 ± 0.42 | 20.16 ± 0.98 |
| ResNet50 | Random | 1 | 0.01 | 44.68 ± 4.17 | 3.16 ± 0.13 |
| ResNet50 | Random | 10 | 0.04 | 83.83 ± 1.79 | 3.03 ± 0.22 |
| ResNet50 | Random | 30 | 1.92 | 87.78 ± 0.33 | 5.55 ± 0.36 |
| ResNet50 | Random | 50 | 4.81 | 89.48 ± 0.49 | 12.72 ± 1.21 |
| ResNet50 | Random | 80 | 8.02 | 89.66 ± 0.19 | 20.19 ± 0.81 |
| MobileNetV3 | CRAIG | 1 | 3.59 | 28.53 ± 2.34 | 2.72 ± 0.00 |
| MobileNetV3 | CRAIG | 10 | 3.62 | 73.06 ± 1.71 | 2.72 ± 0.00 |
| MobileNetV3 | CRAIG | 30 | 5.44 | 79.41 ± 0.80 | 2.72 ± 0.00 |
| MobileNetV3 | CRAIG | 50 | 9.27 | 84.13 ± 0.71 | 2.72 ± 0.00 |
| MobileNetV3 | CRAIG | 80 | 13.06 | 85.82 ± 0.35 | 8.78 ± 0.90 |
| MobileNetV3 | GradMatch | 1 | 1.32 | 27.42 ± 0.93 | 2.72 ± 0.00 |
| MobileNetV3 | GradMatch | 10 | 1.34 | 76.94 ± 1.45 | 2.72 ± 0.00 |
| MobileNetV3 | GradMatch | 30 | 3.17 | 81.96 ± 0.48 | 2.71 ± 0.01 |
| MobileNetV3 | GradMatch | 50 | 6.99 | 84.76 ± 0.56 | 2.72 ± 0.00 |
| MobileNetV3 | GradMatch | 80 | 10.78 | 86.22 ± 0.33 | 8.04 ± 1.53 |
| MobileNetV3 | GLISTER | 1 | 4.27 | 30.85 ± 0.89 | 2.72 ± 0.00 |
| MobileNetV3 | GLISTER | 10 | 4.29 | 74.53 ± 2.56 | 2.70 ± 0.03 |
| MobileNetV3 | GLISTER | 30 | 6.12 | 81.23 ± 0.94 | 2.72 ± 0.00 |
| MobileNetV3 | GLISTER | 50 | 9.95 | 84.67 ± 0.51 | 2.74 ± 0.05 |
| MobileNetV3 | GLISTER | 80 | 13.74 | 85.96 ± 0.47 | 6.79 ± 1.18 |
| MobileNetV3 | Random | 1 | 0.01 | 27.82 ± 1.65 | 2.72 ± 0.00 |
| MobileNetV3 | Random | 10 | 0.03 | 76.92 ± 1.83 | 2.72 ± 0.00 |
| MobileNetV3 | Random | 30 | 1.86 | 81.89 ± 0.41 | 2.71 ± 0.01 |
| MobileNetV3 | Random | 50 | 5.68 | 84.57 ± 0.63 | 2.86 ± 0.26 |
| MobileNetV3 | Random | 80 | 9.47 | 86.06 ± 0.18 | 7.41 ± 1.36 |

Table A25: Performance scores for ResNet50 and MobileNetV3 with ImageNet-21k weights and Random initialized on Oxford-IIIT Pet.

| Model | Baseline | EDPE | Total Training Time | Accuracy (Pretrain) | Accuracy (Random) |
|-------|----------|------|---------------------|---------------------|-------------------|
| ViT_B16 | Full data | 100 | 13.14 | 93.44 ± 0.11 | 15.65 ± 0.36 |
| Swin | Full data | 100 | 20.85 | 95.55 ± 0.28 | 9.95 ± 0.56 |
| Model | Coreset Method | EDPE | Total Training Time | Accuracy (Pretrain) | Accuracy (Random) |
| ViT_B16 | CRAIG | 1 | 5.42 | 50.15 ± 6.12 | 4.36 ± 0.36 |
| ViT_B16 | CRAIG | 10 | 5.48 | 83.36 ± 1.71 | 5.24 ± 0.43 |
| ViT_B16 | CRAIG | 30 | 8.40 | 91.57 ± 0.11 | 8.40 ± 0.61 |
| ViT_B16 | CRAIG | 50 | 12.23 | 92.82 ± 0.20 | 11.86 ± 0.29 |
| ViT_B16 | CRAIG | 80 | 16.79 | 93.15 ± 0.25 | 14.64 ± 0.43 |
| ViT_B16 | GradMatch | 1 | 3.49 | 48.91 ± 6.23 | 4.31 ± 0.32 |
| ViT_B16 | GradMatch | 10 | 3.55 | 88.44 ± 1.41 | 5.93 ± 0.97 |
| ViT_B16 | GradMatch | 30 | 6.47 | 92.17 ± 0.38 | 8.14 ± 0.35 |
| ViT_B16 | GradMatch | 50 | 10.30 | 92.84 ± 0.37 | 12.64 ± 0.72 |
| ViT_B16 | GradMatch | 80 | 14.86 | 93.24 ± 0.09 | 15.18 ± 0.20 |
| ViT_B16 | GLISTER | 1 | 1.82 | 50.33 ± 5.20 | 4.46 ± 0.42 |
| ViT_B16 | GLISTER | 10 | 1.88 | 86.88 ± 1.40 | 6.31 ± 0.28 |
| ViT_B16 | GLISTER | 30 | 4.80 | 92.16 ± 0.15 | 8.73 ± 0.24 |
| ViT_B16 | GLISTER | 50 | 8.63 | 92.81 ± 0.23 | 12.79 ± 0.61 |
| ViT_B16 | GLISTER | 80 | 13.19 | 93.19 ± 0.09 | 15.17 ± 0.52 |
| ViT_B16 | Random | 1 | 0.02 | 49.55 ± 5.10 | 4.44 ± 0.50 |
| ViT_B16 | Random | 10 | 0.07 | 86.84 ± 1.43 | 6.19 ± 0.35 |
| ViT_B16 | Random | 30 | 2.99 | 91.95 ± 0.30 | 8.13 ± 0.28 |
| ViT_B16 | Random | 50 | 6.83 | 92.79 ± 0.28 | 12.26 ± 0.33 |
| ViT_B16 | Random | 80 | 11.38 | 93.05 ± 0.35 | 14.83 ± 0.15 |
| Swin | CRAIG | 1 | 8.87 | 55.15 ± 3.99 | 3.93 ± 0.38 |
| Swin | CRAIG | 10 | 9.12 | 91.23 ± 0.64 | 3.92 ± 0.43 |
| Swin | CRAIG | 30 | 15.01 | 94.16 ± 0.37 | 5.06 ± 0.20 |
| Swin | CRAIG | 50 | 21.30 | 94.61 ± 0.34 | 6.81 ± 0.74 |
| Swin | CRAIG | 80 | 44.01 | 94.97 ± 0.47 | 8.99 ± 0.60 |
| Swin | GradMatch | 1 | 5.81 | 51.93 ± 6.42 | 3.83 ± 0.45 |
| Swin | GradMatch | 10 | 6.06 | 92.54 ± 0.51 | 3.78 ± 0.49 |
| Swin | GradMatch | 30 | 11.95 | 94.69 ± 0.24 | 4.77 ± 0.38 |
| Swin | GradMatch | 50 | 18.23 | 94.91 ± 0.18 | 6.35 ± 0.47 |
| Swin | GradMatch | 80 | 40.95 | 95.34 ± 0.19 | 8.84 ± 1.00 |
| Swin | GLISTER | 1 | 3.08 | 54.85 ± 7.01 | 3.94 ± 0.46 |
| Swin | GLISTER | 10 | 3.33 | 92.90 ± 0.77 | 3.71 ± 0.42 |
| Swin | GLISTER | 30 | 9.22 | 94.39 ± 0.43 | 5.19 ± 0.15 |
| Swin | GLISTER | 50 | 15.50 | 94.83 ± 0.26 | 6.68 ± 0.79 |
| Swin | GLISTER | 80 | 38.22 | 95.34 ± 0.20 | 9.03 ± 1.20 |
| Swin | Random | 1 | 0.07 | 53.24 ± 7.38 | 3.95 ± 0.44 |
| Swin | Random | 10 | 0.32 | 92.47 ± 0.22 | 4.05 ± 0.46 |
| Swin | Random | 30 | 6.21 | 94.50 ± 0.36 | 5.10 ± 0.59 |
| Swin | Random | 50 | 12.50 | 94.68 ± 0.25 | 6.72 ± 0.64 |
| Swin | Random | 80 | 35.21 | 95.28 ± 0.28 | 8.93 ± 1.39 |

Table A26: Performance scores for ViT-B_16 and Swin with ImageNet-21k weights and Random initialized on Oxford-IIIT Pet.

| EDPE (%) | Epoch | Total Time (Mins) | | | Total Selection Time (Mins) | | | Accuracy | | |
|---|---|---|---|---|---|---|---|---|---|---|
| 100 | 105 | | 239.53 | | | - | | | 40.35 | |
| | 89 | | 205.31 | | | - | | | 40.35 | |
| | 79 | | 181.98 | | | - | | | 40.35 | |
| | | SSI=10 | SSI=20 | SSI=50 | SSI=10 | SSI=20 | SSI=50 | SSI=10 | SSI=20 | SSI=50 |
| 50 | 105 | 282.06 | 200.85 | 153.73 | 162.29 | 81.09 | 33.97 | 24.53 | 22.96 | 19.35 |
| | 89 | 232.38 | 167.42 | 119.53 | 129.83 | 64.87 | 16.98 | 24.53 | 22.96 | 19.35 |
| | 79 | 204.23 | 139.28 | 107.61 | 113.60 | 48.65 | 16.98 | 24.10 | 22.25 | 19.35 |
| 30 | 105 | 233.06 | 151.86 | 104.74 | 162.29 | 81.09 | 33.97 | 16.60 | 15.21 | 13.32 |
| | 89 | 190.08 | 125.12 | 77.230 | 129.83 | 64.87 | 16.98 | 16.28 | 15.14 | 13.32 |
| | 79 | 167.32 | 102.36 | 70.69 | 113.60 | 48.65 | 16.98 | 15.96 | 14.60 | 13.32 |
| 10 | 105 | 184.07 | 102.86 | 55.74 | 162.29 | 81.09 | 33.97 | 9.71 | 8.75 | 7.39 |
| | 89 | 148.50 | 83.54 | 35.64 | 129.83 | 64.87 | 16.98 | 9.53 | 8.14 | 7.39 |
| | 79 | 130.19 | 65.24 | 33.57 | 113.60 | 48.65 | 16.98 | 8.75 | 7.82 | 7.39 |
| 1 | 105 | 173.18 | 91.98 | 44.85 | 162.29 | 81.09 | 33.97 | 4.64 | 4.42 | 4.53 |
| | 89 | 139.16 | 74.20 | 26.31 | 129.83 | 64.87 | 16.98 | 4.64 | 4.42 | 4.53 |
| | 79 | 121.69 | 56.74 | 25.07 | 113.60 | 48.65 | 16.98 | 4.64 | 4.42 | 4.53 |

Table A27: Performance scores for ViT-B_16 with ImageNet-21k weights on Ultramnist and its Effective data per epoch (EDPE), method=CRAIG

| EDPE (%) | Epoch | Total Time (Mins) | | | Total Selection Time (Mins) | | | Accuracy | | |
|---|---|---|---|---|---|---|---|---|---|---|
| 100 | 105 | | 239.53 | | | - | | | 40.35 | |
| | 89 | | 205.31 | | | - | | | 40.35 | |
| | 79 | | 181.98 | | | - | | | 40.35 | |
| | | SSI=10 | SSI=20 | SSI=50 | SSI=10 | SSI=20 | SSI=50 | SSI=10 | SSI=20 | SSI=50 |
| 80 | 105 | 276.06 | 233.25 | 207.70 | 85.52 | 42.71 | 17.17 | 36.75 | 33.21 | 32.75 |
| | 89 | 231.73 | 197.49 | 171.90 | 68.42 | 34.17 | 8.58 | 36.57 | 32.71 | 32.75 |
| | 79 | 205.04 | 170.80 | 153.75 | 59.87 | 25.63 | 8.58 | 36.10 | 32.60 | 32.75 |
| 50 | 105 | 205.29 | 162.48 | 136.93 | 85.52 | 42.71 | 17.17 | 25.71 | 22.82 | 19.46 |
| | 89 | 169.52 | 135.27 | 109.68 | 68.42 | 34.17 | 8.58 | 24.96 | 22.57 | 19.28 |
| | 79 | 150.60 | 116.36 | 99.31 | 59.87 | 25.63 | 8.58 | 24.96 | 21.96 | 19.10 |
| 30 | 105 | 156.29 | 113.49 | 87.94 | 85.52 | 42.71 | 17.17 | 18.82 | 14.78 | 13.28 |
| | 89 | 128.04 | 93.79 | 68.20 | 68.42 | 34.17 | 8.58 | 18.21 | 14.78 | 13.28 |
| | 79 | 111.71 | 77.47 | 60.43 | 59.87 | 25.63 | 8.58 | 18.21 | 14.46 | 13.07 |
| 10 | 105 | 107.30 | 64.49 | 38.94 | 85.52 | 42.71 | 17.17 | 9.25 | 7.53 | 7.42 |
| | 89 | 86.56 | 52.32 | 26.73 | 68.42 | 34.17 | 8.58 | 9.25 | 7.53 | 6.75 |
| | 79 | 75.42 | 41.18 | 24.13 | 59.87 | 25.63 | 8.58 | 7.42 | 7.53 | 6.75 |
| 1 | 105 | 96.41 | 53.60 | 28.05 | 85.52 | 42.71 | 17.17 | 3.96 | 4.14 | 4.39 |
| | 89 | 77.75 | 43.50 | 17.91 | 68.42 | 34.17 | 8.58 | 3.96 | 4.14 | 4.32 |
| | 79 | 67.95 | 33.71 | 16.67 | 59.87 | 25.63 | 8.58 | 3.96 | 4.14 | 4.28 |

Table A28: Performance scores for ViT-B_16 with ImageNet-21k weights on Ultramnist and its Effective data per epoch (EDPE), method=GradMatch

| EDPE (%) | Epoch | Total Time (Mins) | | | Total Selection Time (Mins) | | | Accuracy | | |
|---|---|---|---|---|---|---|---|---|---|---|
| 100 | 105 | | 239.53 | | | - | | | 40.35 | |
| | 89 | | 205.31 | | | - | | | 40.35 | |
| | 79 | | 181.98 | | | - | | | 40.35 | |
| | | *SSI=10* | *SSI=20* | *SSI=50* | *SSI=10* | *SSI=20* | *SSI=50* | *SSI=10* | *SSI=20* | *SSI=50* |
| 80 | 105 | 337.64 | 268.18 | 220.34 | 147.10 | 77.64 | 29.80 | 37.35 | 35.42 | 32.78 |
| | 89 | 280.06 | 224.50 | 177.28 | 117.68 | 62.11 | 14.90 | 37.35 | 35.28 | 32.78 |
| | 79 | 246.69 | 190.30 | 158.62 | 102.97 | 46.58 | 14.90 | 37.35 | 35.28 | 32.78 |
| 50 | 105 | 266.86 | 197.41 | 149.57 | 147.10 | 77.64 | 29.80 | 24.35 | 24.28 | 20.14 |
| | 89 | 220.23 | 164.67 | 117.45 | 117.68 | 62.11 | 14.90 | 24.35 | 24.28 | 19.82 |
| | 79 | 193.60 | 137.21 | 105.53 | 102.97 | 46.58 | 14.90 | 24.35 | 24.00 | 19.82 |
| 30 | 105 | 217.87 | 148.41 | 100.57 | 147.10 | 77.64 | 29.80 | 19.07 | 15.64 | 13.53 |
| | 89 | 177.92 | 122.36 | 75.14 | 117.68 | 62.11 | 14.90 | 17.89 | 15.64 | 13.53 |
| | 79 | 156.68 | 100.30 | 68.61 | 102.97 | 46.58 | 14.90 | 17.39 | 14.57 | 13.53 |
| 10 | 105 | 168.87 | 99.42 | 51.58 | 147.10 | 77.64 | 29.80 | 5.00 | 7.46 | 6.92 |
| | 89 | 136.34 | 80.78 | 33.56 | 117.68 | 62.11 | 14.90 | 5.00 | 7.46 | 6.92 |
| | 79 | 119.56 | 63.17 | 31.49 | 102.97 | 46.58 | 14.90 | 5.00 | 7.46 | 6.75 |
| 1 | 105 | 157.99 | 88.53 | 40.69 | 147.10 | 77.64 | 29.80 | 4.14 | 3.96 | 4.21 |
| | 89 | 127.01 | 71.45 | 24.23 | 117.68 | 62.11 | 14.90 | 4.14 | 3.96 | 4.21 |
| | 79 | 111.06 | 54.67 | 22.99 | 102.97 | 46.58 | 14.90 | 4.14 | 3.96 | 4.21 |

Table A29: Performance scores for ViT-B_16 with ImageNet-21k weights on Ultramnist and its Effective data per epoch (EDPE), method=GLISTER

| EDPE (%) | Epoch | Total Time (Mins) | | | Total Selection Time (Mins) | | | Accuracy | | |
|---|---|---|---|---|---|---|---|---|---|---|
| 100 | 105 | | 239.53 | | | - | | | 40.35 | |
| | 89 | | 205.31 | | | - | | | 40.35 | |
| | 79 | | 181.98 | | | - | | | 40.35 | |
| | | *SSI=10* | *SSI=20* | *SSI=50* | *SSI=10* | *SSI=20* | *SSI=50* | *SSI=10* | *SSI=20* | *SSI=50* |
| 80 | 105 | 190.53 | 190.53 | 190.53 | 0.00 | 0.00 | 0.00 | - | 33.89 | - |
| | 89 | 162.38 | 162.38 | 162.38 | 0.00 | 0.00 | 0.00 | - | 33.89 | - |
| | 79 | 143.71 | 143.71 | 143.71 | 0.00 | 0.00 | 0.00 | - | 33.89 | - |
| 50 | 105 | 119.76 | 119.76 | 119.76 | 0.00 | 0.00 | 0.00 | 18.39 | 18.92 | 19.28 |
| | 89 | 102.55 | 102.55 | 102.55 | 0.00 | 0.00 | 0.00 | 18.39 | 18.92 | 19.28 |
| | 79 | 90.62 | 90.62 | 90.62 | 0.00 | 0.00 | 0.00 | 18.39 | 18.92 | 19.25 |
| 30 | 105 | 70.77 | 70.77 | 70.77 | 0.00 | 0.00 | 0.00 | 12.17 | 13.03 | 12.75 |
| | 89 | 60.24 | 60.24 | 60.24 | 0.00 | 0.00 | 0.00 | 12.17 | 13.03 | 12.75 |
| | 79 | 53.71 | 53.71 | 53.71 | 0.00 | 0.00 | 0.00 | 12.17 | 13.03 | 12.75 |
| 10 | 105 | 21.77 | 21.77 | 21.77 | 0.00 | 0.00 | 0.00 | 7.03 | 7.50 | 7.75 |
| | 89 | 18.66 | 18.66 | 18.66 | 0.00 | 0.00 | 0.00 | 7.03 | 7.50 | 7.75 |
| | 79 | 16.59 | 16.59 | 16.59 | 0.00 | 0.00 | 0.00 | 7.03 | 7.50 | 7.75 |
| 1 | 105 | 10.88 | 10.88 | 10.88 | 0.00 | 0.00 | 0.00 | 4.00 | 3.96 | 4.21 |
| | 89 | 9.33 | 9.33 | 9.33 | 0.00 | 0.00 | 0.00 | 4.00 | 3.96 | 4.21 |
| | 79 | 8.08 | 8.08 | 8.08 | 0.00 | 0.00 | 0.00 | 4.00 | 3.96 | 4.21 |

Table A30: Performance scores for ViT-B_16 with ImageNet-21k weights on UltraMNIST and its Effective data per epoch (EDPE), method=Random

| EDPE (%) | Epoch | Total Time (Mins) | | | Total Selection Time (Mins) | | | Accuracy | | |
|---|---|---|---|---|---|---|---|---|---|---|
| | 105 | | 114.99 | | | - | | | 61.28 | |
| 100 | 89 | | 98.56 | | | - | | | 61.28 | |
| | 79 | | 87.36 | | | - | | | 61.21 | |
| | | *SSI=10* | *SSI=20* | *SSI=50* | *SSI=10* | *SSI=20* | *SSI=50* | *SSI=10* | *SSI=20* | *SSI=50* |
| | 105 | 138.70 | 115.05 | 100.78 | 47.23 | 23.58 | 9.31 | 54.28 | 51.78 | 48.21 |
| 80 | 89 | 115.74 | 96.82 | 82.61 | 37.78 | 18.86 | 4.65 | 53.60 | 51.75 | 47.39 |
| | 79 | 102.05 | 83.14 | 73.65 | 33.06 | 14.15 | 4.65 | 52.07 | 50.85 | 46.92 |
| | 105 | 104.72 | 81.07 | 66.80 | 47.23 | 23.58 | 9.31 | 40.28 | 34.25 | 28.82 |
| 50 | 89 | 87.01 | 68.09 | 53.88 | 37.78 | 18.86 | 4.65 | 40.28 | 34.25 | 28.67 |
| | 79 | 76.56 | 57.65 | 48.16 | 33.06 | 14.15 | 4.65 | 39.89 | 32.71 | 28.32 |
| | 105 | 81.20 | 57.55 | 43.28 | 47.23 | 23.58 | 9.31 | 24.57 | 21.39 | 17.39 |
| 30 | 89 | 66.70 | 47.78 | 33.57 | 37.78 | 18.86 | 4.65 | 24.46 | 21.39 | 17.28 |
| | 79 | 58.84 | 39.93 | 30.44 | 33.06 | 14.15 | 4.65 | 24.46 | 20.67 | 17.28 |
| | 105 | 57.68 | 34.03 | 19.76 | 47.23 | 23.58 | 9.31 | 15.78 | 13.00 | 11.46 |
| 10 | 89 | 46.74 | 27.82 | 13.61 | 37.78 | 18.86 | 4.65 | 15.78 | 13.00 | 11.46 |
| | 79 | 41.02 | 22.11 | 12.62 | 33.06 | 14.15 | 4.65 | 15.78 | 13.00 | 11.46 |
| | 105 | 52.45 | 28.81 | 14.54 | 47.23 | 23.58 | 9.31 | 6.92 | 5.50 | 5.85 |
| 1 | 89 | 42.26 | 23.34 | 9.13 | 37.78 | 18.86 | 4.65 | 6.78 | 5.50 | 5.60 |
| | 79 | 36.94 | 18.03 | 8.53 | 33.06 | 14.15 | 4.65 | 6.25 | 5.46 | 5.39 |

Table A31: Performance scores for ResNet50 with ImageNet-21k weights on UltraMNIST and its Effective data per epoch (EDPE), method=CRAIG

| EDPE (%) | Epoch | Total Time (Mins) | | | Total Selection Time (Mins) | | | Accuracy | | |
|---|---|---|---|---|---|---|---|---|---|---|
| | 105 | | 114.99 | | | - | | | 61.28 | |
| 100 | 89 | | 98.56 | | | - | | | 61.28 | |
| | 79 | | 87.36 | | | - | | | 61.21 | |
| | | *SSI=10* | *SSI=20* | *SSI=50* | *SSI=10* | *SSI=20* | *SSI=50* | *SSI=10* | *SSI=20* | *SSI=50* |
| | 105 | 126.53 | 109.21 | 98.26 | 35.06 | 17.74 | 6.79 | 52.71 | 51.46 | 49.03 |
| 80 | 89 | 105.20 | 91.35 | 80.55 | 28.05 | 14.19 | 3.39 | 52.71 | 51.39 | 48.96 |
| | 79 | 94.23 | 80.33 | 73.08 | 24.54 | 10.64 | 3.39 | 52.57 | 50.82 | 48.82 |
| | 105 | 92.55 | 75.23 | 64.29 | 35.06 | 17.74 | 6.79 | 42.17 | 38.07 | 32.32 |
| 50 | 89 | 75.34 | 61.48 | 50.68 | 28.05 | 14.19 | 3.39 | 41.64 | 37.64 | 31.28 |
| | 79 | 66.85 | 52.95 | 45.71 | 24.54 | 10.64 | 3.39 | 41.46 | 36.67 | 31.28 |
| | 105 | 69.03 | 51.71 | 40.76 | 35.06 | 17.74 | 6.79 | 28.92 | 22.46 | 18.89 |
| 30 | 89 | 55.42 | 41.57 | 30.77 | 28.05 | 14.19 | 3.39 | 28.10 | 21.60 | 18.89 |
| | 79 | 49.43 | 35.53 | 28.28 | 24.54 | 10.64 | 3.39 | 26.17 | 21.10 | 18.89 |
| | 105 | 45.51 | 28.19 | 17.24 | 35.06 | 17.74 | 6.79 | 15.10 | 13.28 | 10.10 |
| 10 | 89 | 38.00 | 24.14 | 13.35 | 28.05 | 14.19 | 3.39 | 15.10 | 13.28 | 10.10 |
| | 79 | 32.01 | 18.11 | 10.86 | 24.54 | 10.64 | 3.39 | 14.82 | 13.03 | 10.10 |
| | 105 | 40.29 | 22.96 | 12.02 | 35.06 | 17.74 | 6.79 | 6.53 | 6.67 | 5.10 |
| 1 | 89 | 32.53 | 18.67 | 7.87 | 28.05 | 14.19 | 3.39 | 6.53 | 6.67 | 5.10 |
| | 79 | 28.42 | 14.52 | 7.28 | 24.54 | 10.64 | 3.39 | 6.53 | 5.96 | 5.00 |

Table A32: Performance scores for ResNet50 with ImageNet-21k weights on UltraMNIST and its Effective data per epoch (EDPE), method=GradMatch

| EDPE (%) | Epoch | Total Time (Mins) | | | Total Selection Time (Mins) | | | Accuracy | | |
|---|---|---|---|---|---|---|---|---|---|---|
| | 105 | | 114.99 | | | - | | | 61.28 | |
| 100 | 89 | | 98.56 | | | - | | | 61.28 | |
| | 79 | | 87.36 | | | - | | | 61.21 | |
| | | *SSI=10* | *SSI=20* | *SSI=50* | *SSI=10* | *SSI=20* | *SSI=50* | *SSI=10* | *SSI=20* | *SSI=50* |
| | 105 | 142.94 | 111.20 | 100.78 | 51.47 | 19.73 | 6.78 | 53.92 | 53.00 | - |
| 80 | 89 | 119.13 | 93.74 | 82.61 | 41.18 | 15.78 | 3.39 | 53.92 | 52.67 | - |
| | 79 | 105.02 | 80.83 | 73.65 | 36.03 | 11.84 | 3.39 | 53.92 | 50.82 | - |
| | 105 | 108.97 | 77.23 | 64.28 | 51.47 | 19.73 | 6.78 | 40.53 | 34.85 | 31.00 |
| 50 | 89 | 90.41 | 65.02 | 52.62 | 41.18 | 15.78 | 3.39 | 40.53 | 34.82 | 31.00 |
| | 79 | 79.54 | 55.35 | 46.90 | 36.03 | 11.84 | 3.39 | 39.53 | 34.35 | 31.00 |
| | 105 | 85.45 | 53.71 | 40.76 | 51.47 | 19.73 | 6.78 | 28.25 | 22.78 | 20.71 |
| 30 | 89 | 70.10 | 44.71 | 32.31 | 41.18 | 15.78 | 3.39 | 28.25 | 22.78 | 20.42 |
| | 79 | 61.81 | 37.62 | 29.18 | 36.03 | 11.84 | 3.39 | 27.25 | 11.84 | 20.42 |
| | 105 | 61.92 | 30.19 | 17.24 | 51.47 | 19.73 | 6.78 | 17.50 | 13.64 | 11.46 |
| 10 | 89 | 50.14 | 24.75 | 12.35 | 41.18 | 15.78 | 3.39 | 16.85 | 13.64 | 11.46 |
| | 79 | 43.99 | 19.80 | 11.35 | 36.03 | 11.84 | 3.39 | 16.85 | 13.64 | 11.46 |
| | 105 | 56.70 | 24.96 | 12.01 | 51.47 | 19.73 | 6.78 | 9.50 | 7.82 | 6.03 |
| 1 | 89 | 45.66 | 20.27 | 7.87 | 41.18 | 15.78 | 3.39 | 9.50 | 7.67 | 6.03 |
| | 79 | 39.91 | 15.72 | 7.27 | 36.03 | 11.84 | 3.39 | 8.42 | 7.25 | 6.03 |

Table A33: Performance scores for ResNet50 with ImageNet-21k weights on UltraMNIST and its Effective data per epoch (EDPE), method=GLISTER

| EDPE (%) | Epoch | Total Time (Mins) | | | Total Selection Time (Mins) | | | Accuracy | | |
|---|---|---|---|---|---|---|---|---|---|---|
| | 105 | | 239.53 | | | - | | | 40.35 | |
| 100 | 89 | | 205.31 | | | - | | | 40.35 | |
| | 79 | | 181.98 | | | - | | | 40.35 | |
| | | *SSI=10* | *SSI=20* | *SSI=50* | *SSI=10* | *SSI=20* | *SSI=50* | *SSI=10* | *SSI=20* | *SSI=50* |
| | 105 | 91.47 | 91.47 | 91.47 | 0.00 | 0.00 | 0.00 | 48.75 | 49.35 | 49.75 |
| 80 | 89 | 77.95 | 77.95 | 77.95 | 0.00 | 0.00 | 0.00 | 48.71 | 49.35 | 49.71 |
| | 79 | 68.99 | 68.99 | 68.99 | 0.00 | 0.00 | 0.00 | 48.42 | 49.32 | 49.60 |
| | 105 | 57.49 | 57.49 | 57.49 | 0.00 | 0.00 | 0.00 | 30.25 | 28.75 | 29.75 |
| 50 | 89 | 49.23 | 49.23 | 49.23 | 0.00 | 0.00 | 0.00 | 30.25 | 28.75 | 29.71 |
| | 79 | 43.50 | 43.50 | 43.50 | 0.00 | 0.00 | 0.00 | 30.25 | 28.75 | 29.67 |
| | 105 | 33.97 | 33.97 | 33.97 | 0.00 | 0.00 | 0.00 | 20.07 | 19.17 | 19.14 |
| 30 | 89 | 28.92 | 28.92 | 28.92 | 0.00 | 0.00 | 0.00 | 20.07 | 19.17 | 19.14 |
| | 79 | 25.78 | 25.78 | 25.78 | 0.00 | 0.00 | 0.00 | 20.07 | 19.17 | 19.14 |
| | 105 | 10.45 | 10.45 | 10.45 | 0.00 | 0.00 | 0.00 | 12.46 | 12.89 | 11.89 |
| 10 | 89 | 8.96 | 8.96 | 8.96 | 0.00 | 0.00 | 0.00 | 12.46 | 12.89 | 11.89 |
| | 79 | 7.96 | 7.96 | 7.96 | 0.00 | 0.00 | 0.00 | 12.46 | 12.89 | 11.89 |
| | 105 | 5.22 | 5.22 | 5.22 | 0.00 | 0.00 | 0.00 | 5.75 | 6.89 | 5.67 |
| 1 | 89 | 4.48 | 4.48 | 4.48 | 0.00 | 0.00 | 0.00 | 5.75 | 6.89 | 5.67 |
| | 79 | 3.88 | 3.88 | 3.88 | 0.00 | 0.00 | 0.00 | 5.64 | 6.82 | 5.53 |

Table A34: Performance scores for ResNet50 with ImageNet-21k weights on UltraMNIST and its Effective data per epoch (EDPE), method=Random

| EDPE (%) | Epoch | Total Time (Mins) | | | Total Selection Time (Mins) | | | Quadratic $\kappa$ | | |
|---|---|---|---|---|---|---|---|---|---|---|
| 100 | 105 | | 16.59 | | | - | | | 0.88 | |
| | 89 | | 14.22 | | | - | | | 0.88 | |
| | 79 | | 12.64 | | | - | | | 0.88 | |
| | | SSI=10 | SSI=20 | SSI=50 | SSI=10 | SSI=20 | SSI=50 | SSI=10 | SSI=20 | SSI=50 |
| 80 | 105 | 15.91 | 14.88 | 14.25 | 2.08 | 1.05 | 0.42 | 0.87 | 0.89 | 0.90 |
| | 89 | 13.52 | 12.69 | 12.06 | 1.67 | 0.84 | 0.21 | 0.87 | 0.89 | 0.90 |
| | 79 | 11.99 | 11.16 | 10.74 | 1.46 | 0.63 | 0.21 | 0.87 | 0.89 | 0.90 |
| 50 | 105 | 10.38 | 9.35 | 8.72 | 2.08 | 1.05 | 0.42 | 0.87 | 0.86 | 0.88 |
| | 89 | 8.78 | 7.95 | 7.32 | 1.67 | 0.84 | 0.21 | 0.87 | 0.86 | 0.88 |
| | 79 | 7.78 | 6.95 | 6.53 | 1.46 | 0.63 | 0.21 | 0.87 | 0.86 | 0.88 |
| 30 | 105 | 7.61 | 6.58 | 5.95 | 2.08 | 1.05 | 0.42 | 0.86 | 0.86 | 0.87 |
| | 89 | 6.41 | 5.58 | 4.95 | 1.67 | 0.84 | 0.21 | 0.86 | 0.86 | 0.87 |
| | 79 | 5.67 | 4.84 | 4.42 | 1.46 | 0.63 | 0.21 | 0.86 | 0.86 | 0.87 |
| 10 | 105 | 4.16 | 3.12 | 2.50 | 2.08 | 1.05 | 0.42 | 0.86 | 0.84 | 0.85 |
| | 89 | 3.44 | 2.62 | 1.99 | 1.67 | 0.84 | 0.21 | 0.85 | 0.83 | 0.85 |
| | 79 | 3.04 | 2.21 | 1.79 | 1.46 | 0.63 | 0.21 | 0.85 | 0.83 | 0.85 |
| 1 | 105 | 2.26 | 1.22 | 0.60 | 2.08 | 1.05 | 0.42 | 0.73 | 0.72 | 0.66 |
| | 89 | 1.81 | 0.99 | 0.36 | 1.67 | 0.84 | 0.21 | 0.73 | 0.72 | 0.66 |
| | 79 | 1.59 | 0.76 | 0.34 | 1.46 | 0.63 | 0.21 | 0.71 | 0.72 | 0.66 |

Table A35: Performance scores for ViT_B16 with ImageNet-21k weights on Medical Dataset APTOS-2019 and its Effective data per epoch (EDPE), method=CRAIG

| EDPE (%) | Epoch | Total Time (Mins) | | | Total Selection Time (Mins) | | | Quadratic $\kappa$ | | |
|---|---|---|---|---|---|---|---|---|---|---|
| 100 | 105 | | 16.59 | | | - | | | 0.88 | |
| | 89 | | 14.22 | | | - | | | 0.88 | |
| | 79 | | 12.64 | | | - | | | 0.88 | |
| | | SSI=10 | SSI=20 | SSI=50 | SSI=10 | SSI=20 | SSI=50 | SSI=10 | SSI=20 | SSI=50 |
| 80 | 105 | 15.61 | 14.83 | 14.20 | 1.79 | 1.00 | 0.37 | 0.88 | 0.88 | 0.88 |
| | 89 | 13.28 | 12.65 | 12.04 | 1.43 | 0.80 | 0.18 | 0.88 | 0.88 | 0.88 |
| | 79 | 11.78 | 11.13 | 10.72 | 1.25 | 0.60 | 0.18 | 0.88 | 0.88 | 0.88 |
| 50 | 105 | 10.08 | 9.30 | 8.67 | 1.79 | 1.00 | 0.37 | 0.89 | 0.88 | 0.88 |
| | 89 | 8.54 | 7.91 | 7.30 | 1.43 | 0.80 | 0.18 | 0.89 | 0.88 | 0.88 |
| | 79 | 7.57 | 6.92 | 6.51 | 1.25 | 0.60 | 0.18 | 0.89 | 0.88 | 0.88 |
| 30 | 105 | 7.32 | 6.53 | 5.90 | 1.79 | 1.00 | 0.37 | 0.88 | 0.87 | 0.88 |
| | 89 | 6.17 | 5.54 | 4.93 | 1.43 | 0.80 | 0.18 | 0.88 | 0.87 | 0.88 |
| | 79 | 5.46 | 4.81 | 4.40 | 1.25 | 0.60 | 0.18 | 0.88 | 0.87 | 0.88 |
| 10 | 105 | 3.86 | 3.08 | 2.45 | 1.79 | 1.00 | 0.37 | 0.85 | 0.84 | 0.85 |
| | 89 | 3.21 | 2.58 | 1.96 | 1.43 | 0.80 | 0.18 | 0.85 | 0.84 | 0.85 |
| | 79 | 2.83 | 2.18 | 1.76 | 1.25 | 0.60 | 0.18 | 0.85 | 0.84 | 0.85 |
| 1 | 105 | 1.96 | 1.17 | 0.55 | 1.79 | 1.00 | 0.37 | 0.66 | 0.74 | 0.70 |
| | 89 | 1.58 | 0.95 | 0.33 | 1.43 | 0.80 | 0.18 | 0.66 | 0.74 | 0.70 |
| | 79 | 1.38 | 0.73 | 0.32 | 1.25 | 0.60 | 0.18 | 0.66 | 0.74 | 0.70 |

Table A36: Performance scores for ViT_B16 with ImageNet-21k weights on Medical Dataset APTOS-2019 and its Effective data per epoch (EDPE), method=GradMatch

| EDPE (%) | Epoch | Total Time (Mins) | | | Total Selection Time (Mins) | | | Quadratic $\kappa$ | | |
|---|---|---|---|---|---|---|---|---|---|---|
| | 105 | | 16.59 | | | - | | | 0.88 | |
| 100 | 89 | | 14.22 | | | - | | | 0.88 | |
| | 79 | | 12.64 | | | - | | | 0.88 | |
| | | SSI=10 | SSI=20 | SSI=50 | SSI=10 | SSI=20 | SSI=50 | SSI=10 | SSI=20 | SSI=50 |
| | 105 | 16.46 | 15.77 | 14.51 | 3.41 | 1.94 | 0.68 | 0.88 | 0.88 | 0.88 |
| 80 | 89 | 13.95 | 13.40 | 12.19 | 2.73 | 1.55 | 0.34 | 0.88 | 0.88 | 0.88 |
| | 79 | 12.37 | 11.70 | 10.87 | 2.38 | 1.16 | 0.34 | 0.88 | 0.88 | 0.88 |
| | 105 | 11.71 | 10.24 | 8.98 | 3.41 | 1.94 | 0.68 | 0.87 | 0.88 | 0.87 |
| 50 | 89 | 9.84 | 8.66 | 7.45 | 2.73 | 1.55 | 0.34 | 0.87 | 0.88 | 0.87 |
| | 79 | 8.71 | 7.48 | 6.66 | 2.38 | 1.16 | 0.34 | 0.87 | 0.88 | 0.87 |
| | 105 | 8.94 | 7.47 | 6.22 | 3.41 | 1.94 | 0.68 | 0.86 | 0.87 | 0.87 |
| 30 | 89 | 7.47 | 6.29 | 5.08 | 2.73 | 1.55 | 0.34 | 0.86 | 0.87 | 0.87 |
| | 79 | 6.60 | 5.38 | 4.55 | 2.38 | 1.16 | 0.34 | 0.86 | 0.87 | 0.87 |
| | 105 | 5.48 | 4.01 | 2.76 | 3.41 | 1.94 | 0.68 | 0.75 | 0.76 | 0.81 |
| 10 | 89 | 4.50 | 3.33 | 2.12 | 2.73 | 1.55 | 0.34 | 0.75 | 0.76 | 0.81 |
| | 79 | 3.96 | 2.74 | 1.92 | 2.38 | 1.16 | 0.34 | 0.75 | 0.76 | 0.81 |
| | 105 | 3.58 | 2.11 | 0.86 | 3.41 | 1.94 | 0.68 | 0.70 | 0.67 | 0.54 |
| 1 | 89 | 2.87 | 1.70 | 0.49 | 2.73 | 1.55 | 0.34 | 0.58 | 0.67 | 0.54 |
| | 79 | 2.52 | 1.29 | 0.47 | 2.38 | 1.16 | 0.34 | 0.58 | 0.67 | 0.54 |

Table A37: Performance scores for ViT_B16 with ImageNet-21k weights on Medical Dataset APTOS-2019 and its Effective data per epoch (EDPE), method=GLISTER

| EDPE (%) | Epoch | Total Time (Mins) | | | Total Selection Time (Mins) | | | Quadratic $\kappa$ | | |
|---|---|---|---|---|---|---|---|---|---|---|
| | 105 | | 16.59 | | | - | | | 0.88 | |
| 100 | 89 | | 14.22 | | | - | | | 0.88 | |
| | 79 | | 12.64 | | | - | | | 0.88 | |
| | | SSI=10 | SSI=20 | SSI=50 | SSI=10 | SSI=20 | SSI=50 | SSI=10 | SSI=20 | SSI=50 |
| | 105 | 13.82 | 13.82 | 13.82 | 0.0 | 0.0 | 0.0 | 0.88 | 0.88 | 0.89 |
| 80 | 89 | 11.85 | 11.85 | 11.85 | 0.0 | 0.0 | 0.0 | 0.88 | 0.88 | 0.89 |
| | 79 | 10.53 | 10.53 | 10.53 | 0.0 | 0.0 | 0.0 | 0.88 | 0.88 | 0.89 |
| | 105 | 8.29 | 8.29 | 8.29 | 0.0 | 0.0 | 0.0 | 0.88 | 0.88 | 0.87 |
| 50 | 89 | 7.11 | 7.11 | 7.11 | 0.0 | 0.0 | 0.0 | 0.88 | 0.88 | 0.87 |
| | 79 | 6.32 | 6.32 | 6.32 | 0.0 | 0.0 | 0.0 | 0.88 | 0.88 | 0.87 |
| | 105 | 5.53 | 5.53 | 5.53 | 0.0 | 0.0 | 0.0 | 0.87 | 0.87 | 0.87 |
| 30 | 89 | 4.74 | 4.74 | 4.74 | 0.0 | 0.0 | 0.0 | 0.87 | 0.87 | 0.87 |
| | 79 | 4.21 | 4.21 | 4.21 | 0.0 | 0.0 | 0.0 | 0.87 | 0.87 | 0.87 |
| | 105 | 2.07 | 2.07 | 2.07 | 0.0 | 0.0 | 0.0 | 0.83 | 0.83 | 0.86 |
| 10 | 89 | 1.77 | 1.77 | 1.77 | 0.0 | 0.0 | 0.0 | 0.83 | 0.83 | 0.86 |
| | 79 | 1.58 | 1.58 | 1.58 | 0.0 | 0.0 | 0.0 | 0.83 | 0.83 | 0.85 |
| | 105 | 0.17 | 0.17 | 0.17 | 0.0 | 0.0 | 0.0 | 0.69 | 0.66 | 0.75 |
| 1 | 89 | 0.14 | 0.14 | 0.14 | 0.0 | 0.0 | 0.0 | 0.69 | 0.66 | 0.75 |
| | 79 | 0.13 | 0.13 | 0.13 | 0.0 | 0.0 | 0.0 | 0.69 | 0.66 | 0.75 |

Table A38: Performance scores for ViT_B16 with ImageNet-21k weights on Medical Dataset APTOS-2019 and its Effective data per epoch (EDPE), method=Random

| EDPE (%) | Epoch | Total Time (Mins) | | | Total Selection Time (Mins) | | | Quadratic $\kappa$ | | |
|---|---|---|---|---|---|---|---|---|---|---|
| 100 | 105 | | 10.73 | | | - | | | 0.88 | |
| | 89 | | 9.20 | | | - | | | 0.88 | |
| | 79 | | 8.18 | | | - | | | 0.88 | |
| | | *SSI=10* | *SSI=20* | *SSI=50* | *SSI=10* | *SSI=20* | *SSI=50* | *SSI=10* | *SSI=20* | *SSI=50* |
| 80 | 105 | 10.76 | 9.86 | 9.32 | 1.81 | 0.91 | 0.37 | 0.90 | 0.90 | 0.90 |
| | 89 | 9.12 | 8.40 | 7.85 | 1.45 | 0.73 | 0.18 | 0.90 | 0.89 | 0.90 |
| | 79 | 8.08 | 7.37 | 7.00 | 1.26 | 0.55 | 0.18 | 0.89 | 0.89 | 0.90 |
| 50 | 105 | 7.18 | 6.28 | 5.74 | 1.81 | 0.91 | 0.37 | 0.88 | 0.88 | 0.88 |
| | 89 | 6.05 | 5.33 | 4.79 | 1.45 | 0.73 | 0.18 | 0.88 | 0.88 | 0.88 |
| | 79 | 5.36 | 4.64 | 4.27 | 1.26 | 0.55 | 0.18 | 0.87 | 0.88 | 0.88 |
| 30 | 105 | 5.39 | 4.49 | 3.95 | 1.81 | 0.91 | 0.37 | 0.87 | 0.87 | 0.88 |
| | 89 | 4.51 | 3.80 | 3.25 | 1.45 | 0.73 | 0.18 | 0.83 | 0.85 | 0.88 |
| | 79 | 3.99 | 3.27 | 2.91 | 1.26 | 0.55 | 0.18 | 0.83 | 0.84 | 0.87 |
| 10 | 105 | 3.15 | 2.26 | 1.71 | 1.81 | 0.91 | 0.37 | 0.86 | 0.82 | 0.84 |
| | 89 | 2.60 | 1.88 | 1.33 | 1.45 | 0.73 | 0.18 | 0.85 | 0.82 | 0.84 |
| | 79 | 2.29 | 1.57 | 1.21 | 1.26 | 0.55 | 0.18 | 0.85 | 0.82 | 0.84 |
| 1 | 105 | 1.92 | 1.03 | 0.48 | 1.81 | 0.91 | 0.37 | 0.81 | 0.74 | 0.84 |
| | 89 | 1.54 | 0.83 | 0.28 | 1.45 | 0.73 | 0.18 | 0.81 | 0.71 | 0.83 |
| | 79 | 1.35 | 0.63 | 0.27 | 1.26 | 0.55 | 0.18 | 0.79 | 0.69 | 0.83 |

Table A39: Performance scores for ResNet50 with ImageNet-21k weights on Medical Dataset APTOS-2019 and its Effective data per epoch (EDPE), method=CRAIG

| EDPE (%) | Epoch | Total Time (Mins) | | | Total Selection Time (Mins) | | | Quadratic $\kappa$ | | |
|---|---|---|---|---|---|---|---|---|---|---|
| 100 | 105 | | 10.73 | | | - | | | 0.88 | |
| | 89 | | 9.20 | | | - | | | 0.88 | |
| | 79 | | 8.18 | | | - | | | 0.88 | |
| | | *SSI=10* | *SSI=20* | *SSI=50* | *SSI=10* | *SSI=20* | *SSI=50* | *SSI=10* | *SSI=20* | *SSI=50* |
| 80 | 105 | 10.46 | 9.71 | 9.26 | 1.51 | 0.76 | 0.31 | 0.90 | 0.89 | 0.89 |
| | 89 | 8.88 | 8.28 | 7.82 | 1.21 | 0.61 | 0.15 | 0.90 | 0.89 | 0.89 |
| | 79 | 7.88 | 7.27 | 6.97 | 1.06 | 0.45 | 0.15 | 0.90 | 0.89 | 0.89 |
| 50 | 105 | 6.88 | 6.13 | 5.68 | 1.51 | 0.76 | 0.31 | 0.88 | 0.89 | 0.89 |
| | 89 | 5.81 | 5.21 | 4.75 | 1.21 | 0.61 | 0.15 | 0.88 | 0.89 | 0.89 |
| | 79 | 5.15 | 4.54 | 4.24 | 1.06 | 0.45 | 0.15 | 0.88 | 0.89 | 0.89 |
| 30 | 105 | 5.09 | 4.34 | 3.89 | 1.51 | 0.76 | 0.31 | 0.88 | 0.88 | 0.88 |
| | 89 | 4.28 | 3.67 | 3.22 | 1.21 | 0.61 | 0.15 | 0.88 | 0.88 | 0.88 |
| | 79 | 3.79 | 3.18 | 2.88 | 1.06 | 0.45 | 0.15 | 0.87 | 0.88 | 0.88 |
| 10 | 105 | 2.86 | 2.10 | 1.65 | 1.51 | 0.76 | 0.31 | 0.86 | 0.87 | 0.85 |
| | 89 | 2.36 | 1.76 | 1.30 | 1.21 | 0.61 | 0.15 | 0.86 | 0.87 | 0.85 |
| | 79 | 2.08 | 1.48 | 1.17 | 1.06 | 0.45 | 0.15 | 0.86 | 0.86 | 0.85 |
| 1 | 105 | 1.62 | 0.87 | 0.42 | 1.51 | 0.76 | 0.31 | 0.79 | 0.80 | 0.73 |
| | 89 | 1.31 | 0.70 | 0.25 | 1.21 | 0.61 | 0.15 | 0.79 | 0.80 | 0.73 |
| | 79 | 1.14 | 0.54 | 0.24 | 1.06 | 0.45 | 0.15 | 0.79 | 0.80 | 0.73 |

Table A40: Performance scores for ResNet50 with ImageNet-21k weights on Medical Dataset APTOS-2019 and its Effective data per epoch (EDPE), method=GradMatch

| EDPE (%) | Epoch | Total Time (Mins) | | | Total Selection Time (Mins) | | | Quadratic $\kappa$ | | |
|---|---|---|---|---|---|---|---|---|---|---|
| | | | 10.73 | | | - | | | 0.88 | |
| 100 | 105 | | | | | | | | | |
| | 89 | | 9.20 | | | - | | | 0.88 | |
| | 79 | | 8.18 | | | - | | | 0.88 | |
| | | SSI=10 | SSI=20 | SSI=50 | SSI=10 | SSI=20 | SSI=50 | SSI=10 | SSI=20 | SSI=50 |
| | 105 | 12.46 | 10.70 | 9.65 | 3.51 | 1.76 | 0.70 | 0.90 | 0.90 | 0.89 |
| 80 | 89 | 10.48 | 9.07 | 8.02 | 2.81 | 1.40 | 0.35 | 0.90 | 0.90 | 0.89 |
| | 79 | 9.28 | 7.87 | 7.17 | 2.46 | 1.05 | 0.35 | 0.89 | 0.89 | 0.89 |
| | 105 | 8.88 | 7.13 | 6.07 | 3.51 | 1.76 | 0.70 | 0.88 | 0.88 | 0.87 |
| 50 | 89 | 7.41 | 6.01 | 4.95 | 2.81 | 1.40 | 0.35 | 0.86 | 0.88 | 0.87 |
| | 79 | 6.55 | 5.14 | 4.44 | 2.46 | 1.05 | 0.35 | 0.86 | 0.85 | 0.87 |
| | 105 | 7.09 | 5.34 | 4.28 | 3.51 | 1.76 | 0.70 | 0.73 | 0.85 | 0.88 |
| 30 | 89 | 5.88 | 4.47 | 3.42 | 2.81 | 1.40 | 0.35 | 0.73 | 0.85 | 0.88 |
| | 79 | 5.18 | 3.78 | 3.08 | 2.46 | 1.05 | 0.35 | 0.73 | 0.85 | 0.88 |
| | 105 | 4.85 | 3.10 | 2.05 | 3.51 | 1.76 | 0.70 | 0.78 | 0.81 | 0.85 |
| 10 | 89 | 3.96 | 2.55 | 1.50 | 2.81 | 1.40 | 0.35 | 0.78 | 0.81 | 0.83 |
| | 79 | 3.48 | 2.07 | 1.37 | 2.46 | 1.05 | 0.35 | 0.78 | 0.80 | 0.83 |
| | 105 | 3.62 | 1.87 | 0.82 | 3.51 | 1.76 | 0.70 | 0.64 | 0.68 | 0.74 |
| 1 | 89 | 2.90 | 1.50 | 0.45 | 2.81 | 1.40 | 0.35 | 0.64 | 0.62 | 0.74 |
| | 79 | 2.54 | 1.14 | 0.43 | 2.46 | 1.05 | 0.35 | 0.64 | 0.62 | 0.74 |

Table A41: Performance scores for ResNet50 with ImageNet-21k weights on Medical Dataset APTOS-2019 and its Effective data per epoch (EDPE), method=GLISTER

| EDPE (%) | Epoch | Total Time (Mins) | | | Total Selection Time (Mins) | | | Quadratic $\kappa$ | | |
|---|---|---|---|---|---|---|---|---|---|---|
| | 105 | | 10.73 | | | - | | | 0.88 | |
| 100 | 89 | | 9.20 | | | - | | | 0.88 | |
| | 79 | | 8.18 | | | - | | | 0.88 | |
| | | SSI=10 | SSI=20 | SSI=50 | SSI=10 | SSI=20 | SSI=50 | SSI=10 | SSI=20 | SSI=50 |
| | 105 | 8.94 | 8.94 | 8.94 | 0.0 | 0.0 | 0.0 | 0.89 | 0.89 | 0.89 |
| 80 | 89 | 7.67 | 7.67 | 7.67 | 0.0 | 0.0 | 0.0 | 0.89 | 0.89 | 0.89 |
| | 79 | 6.81 | 6.81 | 6.81 | 0.0 | 0.0 | 0.0 | 0.89 | 0.89 | 0.89 |
| | 105 | 5.36 | 5.36 | 5.36 | 0.0 | 0.0 | 0.0 | 0.88 | 0.89 | 0.88 |
| 50 | 89 | 4.60 | 4.60 | 4.60 | 0.0 | 0.0 | 0.0 | 0.88 | 0.89 | 0.88 |
| | 79 | 4.09 | 4.09 | 4.09 | 0.0 | 0.0 | 0.0 | 0.88 | 0.89 | 0.88 |
| | 105 | 3.57 | 3.57 | 3.57 | 0.0 | 0.0 | 0.0 | 0.87 | 0.87 | 0.87 |
| 30 | 89 | 3.06 | 3.06 | 3.06 | 0.0 | 0.0 | 0.0 | 0.87 | 0.87 | 0.87 |
| | 79 | 2.72 | 2.72 | 2.72 | 0.0 | 0.0 | 0.0 | 0.87 | 0.87 | 0.87 |
| | 105 | 1.34 | 1.34 | 1.34 | 0.0 | 0.0 | 0.0 | 0.84 | 0.86 | 0.86 |
| 10 | 89 | 1.15 | 1.15 | 1.15 | 0.0 | 0.0 | 0.0 | 0.84 | 0.85 | 0.86 |
| | 79 | 1.02 | 1.02 | 1.02 | 0.0 | 0.0 | 0.0 | 0.84 | 0.85 | 0.86 |
| | 105 | 0.11 | 0.11 | 0.11 | 0.0 | 0.0 | 0.0 | 0.82 | 0.79 | 0.78 |
| 1 | 89 | 0.09 | 0.09 | 0.09 | 0.0 | 0.0 | 0.0 | 0.82 | 0.79 | 0.78 |
| | 79 | 0.08 | 0.08 | 0.08 | 0.0 | 0.0 | 0.0 | 0.82 | 0.79 | 0.78 |

Table A42: Performance scores for ResNet50 with ImageNet-21k weights on Medical Dataset APTOS-2019 and its Effective data per epoch (EDPE), method=Random

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
