# OpenReview forum: "Data-Efficient Training of CNNs and Transformers with Coresets: A Stability Perspective"
_TMLR — Rejected by TMLR_

### Review · Reviewer_qjey · 2023-03-23

**Summary Of Contributions:**

This paper presents a study measuring the effectiveness of coreset selection in CNN and Transformer image classification models in multiple settings, evaluating three coreset methods against random sampling for cases where models are pretrained, not pretrained, and where training data is taken from datasets very different from the pretraining data (medical images or ultra-mnist).


**Audience:**

No

**Claims And Evidence:**

No

**Requested Changes:**

See above.  All changes are necessary, but in particular:

* All claims should be evaluated on multiple datasets.  Effect of pretraining data, for example, is only compared on CIFAR-10.

* Claims around "natural" vs "non-natural" images should also use "natural" datasets outside of ImageNet.  Conclusions drawn from TinyImageNet may not apply to other datasets from sources outside the pretraining data, and CIFAR-10 is very small to use for measuring these higher res models.  Additional datasets that might make sense could be ones like oxford pets or places365 --- but those are just two currently at the top of my head, there are likely better ones to work with for this.

* Descriptions of the coreset selection methods should be checked and fixed.


Additionally, including some larger datasets would strengthen the work significantly.  Since this is a study comparing coreset methods on transformers, it would make sense to evaluate them in a large-data context where both transformers and coresets make most sense.

**Strengths And Weaknesses:**

While I appreciate the study on coreset sampling for more models and dataset settings, there are many significant issues in the current set of experiments.

There are several several major issues broadly applicable:

  * Too few datasets are used in each experiment section.  The effect of pretraining data, for example, is only compared on CIFAR-10.  Why not compare this on all datasets?

  * The only "natural image" datasets used are TinyImageNet, which is a subset of the pretraining data, and CIFAR-10, which is small.  Additional datasets drawn from sources similar to (but not exactly matching) ImageNet would be better for evaluating the effects of pretraining and "natural" vs "non-natural" images.

  * Results should be compared to the (upper) accuracy baseline of training the models without coreset sampling, for long enough that they converge to close to their best achievable values.  This would put the results in context, and determine whether any differences observed are from changing the coreset size or algorithm parameters, or whether merely training with a different sampling method by enabling subset selection algorithms has an impact.  It looks like this may be in the appendix, but should be included in the main text comparisons.


There are also several issues particular to several of the descriptions and experiments:


Sec 3.2:  Of the 3 methods summarized, only GradMatch made sense to me (after looking up and reading the cited reference).  CRAIG and GLISTER look incorrect.  The summaries for all three should be double-checked and rewritten if necessary.  For example, I see GLISTER is documented at [ https://cords.readthedocs.io/en/latest/strategies/cords.selection_strategies.SL.html#module-cords.selectionstrategies.SL.glisterstrategy ].  But eq.5 looks like it mixed together the equations from the doc/paper.

Sec 4.2:  Initial discussion of Fig 2:  The observation that random selection is better in this case, is limited by the fact that TinyImageNet is a subset of the pretraining dataset.  The networks only have to recover top-layer classifier weights on features already trained on this data, which tends to be fast and not need large data samples.  Why not measure the other datasets in this setting as well?

Sec 4.2 (p.12):  "the takeaway is that the complexity of the distribution per class should be taken into consideration" and p.2 claim 2 "... rather samples should be adaptively sampled based on ... complexity" --- This is a good hypothesis, but there isn't enough evidence to conclude this is the case without testing it:  How could you incorporate this effect into sampling?




Additional minor writing issues and typos:


p.4 broken ref for imagenet21k

p.5 just before eq. 2, I think shoudl say L*~ approx. = L*  (not = L as written now)

p.5 -> p.6: abrupt transition, goes from end of paragraph on p.5, to eq3 with no introduction for what it is defining

p.6 "EDPE":  why say "effective" data?  it actually _is_ the data at each epoch, so could more simply be called "DPE".

---

### Review · Reviewer_4r5Y · 2023-03-23

**Summary Of Contributions:**

The paper benchmarks existing solutions (Random selection, CRAIG, GradMatch and GLISTER) to coreset selection. The paper considers both CNN as well as transformer architectures. The coreset selection approaches are validated on four datasets CIFAR10, TinyImageNet, APTOS-2019 and UltraMNIST. Overall, the paper highlights the strengths of random selection over sota solutions, advocates for non-uniform (across classes) sampling, show that pretrained transformers outperform CNNs and in the lack of good pretraining CNNs perform better than transformers. The paper promises to release the code to reproduce the reported results.

**Audience:**

Yes

**Broader Impact Concerns:**

It would be interesting to discuss the potential impacts of coreset selection on the biases of the learned representations.

**Claims And Evidence:**

Yes

**Requested Changes:**

- Coreset definition: In the introduction of the manuscript the authors write that for the coreset “the model converges approximately to the same solution” as for the full dataset. However, definition in the section 3.1 just ensures the same loss value and not the same solution (as many solutions could have the same loss value). Could the authors clarify this point?

- Positioning the paper w.r.t. data pruning literature would strengthen the paper, e.g. see https://arxiv.org/abs/2206.14486.

- Adding standard deviation to plots would make the paper stronger.

- Plots should be bigger – currently they are hard to read.

- Adding ImageNet dataset would make the analysis stronger.




**Strengths And Weaknesses:**

Strengths:
- The problem of coreset selection is interesting.
- The paper studies a wide range of setups and datasets.
- The benchmarking looks solid.
- The overall paper presentation is rather clear.


Weaknesses:
- The conclusion of the analysis is not too interesting.
- The standard deviation in the comparisons is not reported.
- There are some inconsistencies around coreset definition
- The evaluation is performed on small scale datasets

---

### Review · Reviewer_KyV3 · 2023-04-12

**Summary Of Contributions:**

This paper presents a systematic benchmarking scheme that allows a fair comparison of different coreset selection methods for data efficient training. Specifically, a thorough comparison of different coreset selection methods with respect to model performance at different coreset sizes is discussed. CNN-based and Transformer-based networks are used for the models, and CRAIG, GradMatch, GLISTER, and random selection are compared for the coreset selection. Many validations are performed in experiments, including non-natural images such as medical images, and coreset selection methods and subset selection intervals are discussed.

**Audience:**

Yes

**Broader Impact Concerns:**

There is no concern.

**Claims And Evidence:**

Yes

**Requested Changes:**

Regarding the interpretation of experimental results

- In subsection 4.2, the authors state "For ResNet and MobileNet, except for random selection, GradMatch consistently outperforms other methods. Whereas, for longer training time, CRAIG is only marginally better." However, the performance of CRAIG and GradMatch appears to be identical from Figure 2. Is there a test for significant differences in the performance comparison of the coreset selections? Also, what is the variance of the performance?
- In subsection 4.2, the authors state "For very small coresets, GLISTER is outperformed by even the random selection method." However, how can we read from fig.2 the situation of very small coresets?
- In subsection 4.2, the authors state "As for the total time, except for GradMatch, all methods get outperformed by random selection (especially, small time budgets), which is counterintuitive and implies that the coreset selection methods are not stable for CNNs at low training time budgets."
However, how can this be said from Figure 2? For example, GRAIG is better than random selection at around 240 mins.
- The authors state "For this study, we conducted experiments on the CIFAR10 dataset, using ResNet50 and VIT-B16 for CRAIG and GradMatch methods. Related results are shown in Fig. 3. It can be seen that CNN architecture outperforms transformers significantly without pretraining. We argue that it is because transformers are data-hungry and thus are affected severely in the absence of pretraining."
It would be difficult to discuss the architectural differences without keeping the number of parameters in the model the same. It is also better not to expand the discussion to a general comparison of CNNs and transformers by comparing only one of the CNN and transformer implementations, such as resnet and ViT.
- In Figure 4, only CRAIG is used as the coreset selection, but what happens in the case of other selection methods?

Regarding the equations
- The reviewer would like the authors to explain what \mathcal{L}^i refers to in equations 3, 4 and 5. Is it the loss for one data i?
- In Equation 3, is \mathcal{L}^i \bm{w}?
- In Equation 5, argmin is used, while Equations 3 and 4 return values related to the loss. Equations 3 and 4 should also be expressions that return a coreset.
- In Equation 5, it is \nabla_{\theta}, but shouldn't it be \nabla_{w}?
- Is the notation for (\mu) in Equation 5 correct?
- Equation 5 is \mathcal{S} \subseteq \mathcal{D}, but is it \mathcal{S} \subset \mathcal{D} as stated in the text?


**Strengths And Weaknesses:**

Strengths

- Experiments are conducted to compare different coreset selection methods with respect to model performance at different coreset sizes.
- This paper conducts rigorous experiments comparing model performance and training time for different coreset selection methods for the Transformer model.
- When using pre-trained models, this paper tests whether the Transformer is as stable as the CNN model when a small coreset is selected.
- In the experiments, this paper has performed many validations including non-natural images such as medical images.

Weaknesses
- Discussions and conclusions are stated without verification of significance in comparative experiments.
- Discussions that develop from a small number of limited observations to generalizations are unclear.
- Mathematical expressions are ambiguous or incorrect.

---

### Author Response · Authors · 2023-04-26
**Response to all Reviewers and Action Editor**

Dear Action Editor and Reviewers,

We are thankful to all of you for putting your time and efforts into providing constructive feedback on the work. Based on the remarks, we are currently preparing the following set of experiments in response to the feedback received.
- To demonstrate the stability of the results in terms of variance for the four chosen coreset methods, we will be running 5 repetitions on one of our experiments using the Oxford-Pets dataset for 5 different EDPE values for the case of random as well as pretrained initializations.
- For the study aimed at studying the difference of random and pertained initializations, we restricted our study earlier to only two coreset selection methods. We will be extending the study to include the other two coreset methods as well on the CIFAR10 dataset. We will be using the same EDPE as for the other two methods.
- For experiments on larger dataset, we will be adding the results on ImageNET. Due to resource constraints, we will limit the experiments to one CNN and one transformer architecture for all the coreset methods and 3 EDPE values.

Each experiment reported above takes significant time and computational effort and requires quite some time. In this regard, we would like to request an additional time of 4-6 weeks to complete these and improve the overall draft. Hope the review team agrees with our request and acts accordingly.

Best regards
Authors

---

> ### Comment · Reviewer_qjey · 2023-04-28
> **reply**
>
> Thanks for the update.  I have no problem allowing additional time myself, though a decision on that would be up to the AE.
>
> The experiments you propose look like they would strengthen the analysis in the paper.  However, for the third point, I don't think ImageNet would be an appropriate dataset if any of your comparisons use models pretrained on ImageNet21k, as then the downstream data would be included in pretraining.  I'm not sure if you were planning to use imnet21k pretraining or random init in this experiment.  If only random init is used, then ImageNet is OK, but if any use pretrained weights, I think a different dataset would be needed.

---

> > ### Author Response · Authors · 2023-05-05
> >
> > Thank you for pointing this out. Indeed the pretrained models used in this study are already trained on Imagenet21K. Hence demonstrating the results on ImageNet might not be the best choice. Further, from a practical perspective, we believe it is more reasonable to conduct the experiments with the pretrained models. Hence, we consider an additional study on a medical dataset, Lung Adenocarcinoma (LUAD) versus Lung Squamous Cell Carcinoma (LUSC) in Non-Small Cell Lung Carcinoma (NSCLC). We will run 5 trials and report the variance of the results as well.

---

### Decision · Action_Editors · 2023-06-29

**Recommendation:** Reject

**Comment:**

This paper examines the problem of coreset selection, which involves the selection of relevant subsets of training data to reduce training time while maintaining predictive performance. The paper evaluates three coreset methods (GradMatch, CRAIG, and GLISTER) against random sampling in different learning contexts (pre-training, semantic distance between pre-trained and fine-tuned data), and for ConvNets and transformers.
The paper initially received mixed reviews: although the reviewers appreciated the focus on the work, they raised several concerns regarding paper presentation, methods' formalization (e.g. presentation of the baselines), and about the quality of the experiments. They especially pointed out the necessity of reporting statistical significance tests and extending the use of larger datasets. The authors provided a revised version, including new experiments on the Oxford Pets dataset. Although some reviewers were satisfied with the improvement in paper presentation, others considered that the definition of the coreset was still not aligned with the paper's equation. All reviewers still believed that the link between the paper's claims and the experimental results was insufficient. Additionally, one reviewer was not convinced by the authors' discussion on the relationship between coreset and pruning methods. After the revision, there was a consensus among the reviewers to reject the paper.

The AE carefully reads the submission and discussions. The AE considers that the focus of this work is relevant. The main claim of the paper is that random selection overall gives similar performances to more advanced coreset selection methods in the literature. Although this negative result might be interesting for the TMLR audience, it should be consolidated at several levels. The authors did not respond to the different concerns of the reviewers on the OpenReview console, and the revised manuscript only partially resolved the issues pointed out by them. The AC considers that the authors should better justify the level of generality reached in the experiments and thoroughly analyze the poor performances of coreset selection methods compared to the random selection baseline, e.g., regarding the EDPE metrics or classes' imbalance. Therefore, the AE recommends rejection.


**Audience:**

The submission overall outputs negative results. This might be interesting for the TMLR audience, but the results are premature for publication, and additional analyses should be provided to better explain this behavior.


**Claims And Evidence:**

The main claims should be consolidated and are not always fully validated by evidence.


**Resubmission Of Major Revision:**

The authors may consider submitting a major revision at a later time.